# Block-Diagonal LoRA for Eliminating Communication Overhead in Tensor Parallel LoRA Serving

**Xinyu Wang** *
University of Warwick
Coventry, UK

**Jonas M. Kübler**
Amazon Web Services
Tübingen, Germany

**Kailash Budhathoki**
Amazon Web Services
Tübingen, Germany

**Yida Wang**
Amazon Web Services
Santa Clara, USA

**Matthäus Kleindessner** [†]
Amazon Web Services
Tübingen, Germany

## Abstract

When serving a single base LLM with several different LoRA adapters simultaneously, the adapters cannot simply be merged with the base model's weights as the adapter swapping would create overhead and requests using different adapters could not be batched. Rather, the LoRA computations have to be separated from the base LLM computations, and in a multi-device setup the LoRA adapters can be sharded in a way that is well aligned with the base model's tensor parallel execution, as proposed in S-LoRA. However, the S-LoRA sharding strategy encounters some communication overhead, which may be small in theory, but can be large in practice. In this paper, we propose to constrain certain LoRA factors to be block-diagonal, which allows for an alternative way of sharding LoRA adapters that does not require any additional communication for the LoRA computations. We demonstrate in extensive experiments that our block-diagonal LoRA approach is similarly parameter efficient as standard LoRA (i.e., for a similar number of parameters it achieves similar downstream performance) and that it leads to significant end-to-end speed-up over S-LoRA. For example, when serving on eight A100 GPUs, we observe up to 1.79x (1.23x) end-to-end speed-up with 0.87x (1.74x) the number of adapter parameters for Llama-3.1-70B, and up to 1.63x (1.3x) end-to-end speed-up with 0.86x (1.73x) the number of adapter parameters for Llama-3.1-8B.

## 1 Introduction

Low-Rank Adaptation (LoRA) [10] is one of the most prominent methods for parameter-efficient fine-tuning: rather than fine-tuning all of a model's weights, we only tune a small number of parameters in low-rank factorizations to be added to the model's weight matrices. Concretely, if $W \in \mathbb{R}^{d_{in} \times d_{out}}$ denotes a weight matrix in the model to be fine-tuned, we replace $W$ by $W + AB$, where $A \in \mathbb{R}^{d_{in} \times r}$, $B \in \mathbb{R}^{r \times d_{out}}$ with $r \ll \min\{d_{in}, d_{out}\}$ are the low-rank factors that are fine-tuned, while the parameters in $W$ are frozen. If there is only one LoRA adapter (i.e., one $A$ and $B$ per weight matrix $W$) trained for one task, after fine-tuning $A$ and $B$ we can actually *replace* $W$ by $W + AB$ and thus merge the adapter with the base model's weights. This has the nice effect that the LoRA adapter does not introduce any additional inference latency compared to the base model (or any fully fine-tuned variant of it). However, if we want to serve the same base model with various different

39th Conference on Neural Information Processing Systems (NeurIPS 2025).

---

*Work done during an internship at AWS Tübingen.

†Correspondence to matkle@amazon.com.

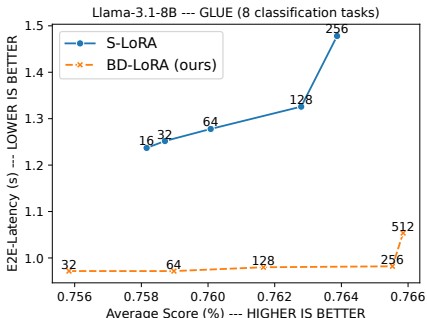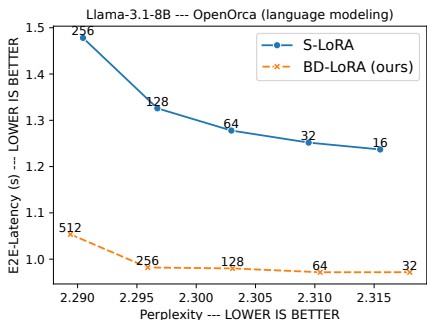

Figure 1: Downstream performance versus run-time trade-off plots for BD-LoRA (our proposed approach) and S-LoRA [28] when serving Llama-3.1-8B with LoRA adapters of varying rank (numbers in the plots). Downstream performance is perplexity on OpenOrca (top plot; lower is better) or the mean performance over eight classification / regression tasks in the GLUE benchmark (bottom plot; higher is better). Run-time is end-to-end latency when serving on eight A100 GPUs and querying with requests of batch size 1 and 1024 input tokens to generate 128 output tokens. For a certain downstream performance, BD-LoRA is significantly faster than S-LoRA (since it requires similar compute and memory movement but no communication), that is BD-LoRA strictly Pareto-dominates S-LoRA in these plots.

LoRA adapters (e.g., to serve different users on different tasks), adapter merging may be inefficient as every time we changed to a new adapter, we would have to subtract the old LoRA weights and add the new ones. Furthermore, it does not allow for batched inference unless all requests within a batch use the same LoRA adapter. Hence, in multi-LoRA serving we do not aim to replace the matrix $W$, but instead replace the computation of $XW$, for some input $X$, by $XW + XAB$. The S-LoRA paper [28] introduces techniques to do so efficiently for large language models (LLMs) based on the transformer architecture [34] and provides a system for the scalable serving of up to thousands of LoRA adapters. In particular, S-LoRA introduces a tensor parallelism (TP) strategy to shard $A$ and $B$ over multiple devices (e.g., GPUs, ML accelerators) and parallelize the computation of $XAB$.

The S-LoRA TP strategy is well aligned with the Megatron-LM [29] TP strategy used to shard the base LLM. However, it requires additional communication on top of the communication required for the base model. In theory, this communication overhead is small if the rank of the LoRA adapters is small. But while Hu et al. [10] originally show that LoRA adapters with larger ranks are not more accurate than the ones with small rank, the more recent rsLoRA paper [13] demonstrates that LoRA with the right scaling factor *does* benefit from larger ranks. More importantly, as we will see in our evaluations of the S-LoRA implementation in the popular vLLM serving framework [14], in practice, the communication overhead of S-LoRA is significant even when the rank is small.

We propose to modify LoRA's architecture and to constrain certain LoRA factors to be block-diagonal, which allows for an alternative to the S-LoRA tensor parallelism strategy. With our TP strategy we can parallelize the computation of $XAB$ without any additional communication on top of the communication required for the base model. Our approach can be interpreted as adding independent LoRA adapters to every shard of the base model's weights. Of course, our block-diagonality constraint reduces the LoRA adapter's expressiveness (in particular, our modified architecture is not mathematically equivalent to standard LoRA), but we show in extensive experiments that for a similar number of effective (i.e., non-zero) parameters, both standard LoRA / S-LoRA and our block-diagonal variant (termed BD-LoRA in the following) achieve very similar downstream performance (often BD-LoRA actually outperforms standard LoRA). At the same time, for a similar number of effective parameters, both S-LoRA and BD-LoRA require similar compute and memory resources, but since BD-LoRA does not require any communication, overall it is significantly faster than S-LoRA. Hence, BD-LoRA strictly Pareto-dominates S-LoRA in downstream performance versus run-time trade-off plots as shown in Figure 1. In these plots, when comparing end-to-end latency at a certain downstream performance (and favoring S-LoRA by requiring BD-LoRA to have better or equal downstream performance for a pair of comparison), we observe end-to-end speed-ups of 1.27x - 1.51x for Llama-3.1-8B served on eight A100 GPUs. In our other experiments, where we compare run-time at a similar number of parameters, we observe end-to-end speed-ups of up to 1.79x (1.23x) with

0.87x (1.74x) number of adapter parameters for Llama-3.1-70B on eight A100 GPUs, 1.63x (1.3x) with 0.86x (1.73x) number of adapter parameters for Llama-3.1-8B on eight A100 GPUs, 1.27x with 1.03x number of adapter parameters for Llama-3.1-8B on four A100 GPUs, and 1.36x (1.27x) with 0.86x (1.73x) number of adapter parameters for Llama-3.1-8B on eight less powerful A10G GPUs.

To measure run-time we implemented BD-LoRA inference in the popular open-source vLLM serving framework [14], which provides an implementation of the S-LoRA tensor parallelism strategy. **A slightly updated version of our BD-LoRA serving code is available as a draft pull request at** `https://github.com/vllm-project/vllm/pull/28136`. **Our BD-LoRA training code has been merged into Huggingface PEFT (see** `https://github.com/huggingface/peft/tree/main/examples/bdlora_finetuning`**).**

## 2 Background on LoRA, rsLoRA and S-LoRA

**Low-Rank Adaptation (LoRA)** [10] makes model fine-tuning much more efficient by freezing model weights and instead only tuning a small number of parameters in low-rank factors to be added to the pre-trained weight matrices as described in Section 1. Hu et al. [10] only focus on transformer language models and add LoRA adapters only to the attention weight matrices, but the technique can be applied to any linear layer in arbitrary neural networks. In our main experiments we add LoRA adapters to both attention and MLP weight matrices similarly to other papers on LoRA [e.g., 5, 13, 44], but we also run some experiments with attention or MLP weight matrices only.

**Rank-stabilized LoRA (rsLoRA)** During training, low-rank adapters are parameterized as $\gamma_r AB$ for $\gamma_r \in \mathbb{R}^+$, where Hu et al. [10] suggest to set $\gamma_r = \frac{\alpha}{r}$ for some hyperparameter $\alpha$ that does not depend on the rank $r$, to reduce the effect of $r$ on the product $AB$ and help with hyperparameter transfer across different values of $r$. Kalajdzievski [13] shows theoretically that the scaling factor should rather be $\frac{\alpha}{\sqrt{r}}$ as this is the only scaling factor that ensures stable adapter learning as $r \rightarrow \infty$. Furthermore, Kalajdzievski [13] demonstrates empirically that with this scaling factor fine-tuning results improve as $r$ increases (up to $r = 2048$). This is in contrast to Hu et al. [10], who report similar results for $r = 8$ and $r = 64$ and conclude that often a very small rank would be sufficient. We verified that with rsLoRA scaling downstream performance monotonically improves with $r$ whereas with standard scaling it does not (cf. Appendix E.3) and hence use rsLoRA scaling.

**S-LoRA** [28] is a system designed for the scalable serving of up to thousands of different LoRA adapters with the same base LLM. This is achieved by exploiting several techniques. Crucially, in contrast to the original LoRA paper, in S-LoRA the adapter weights are not merged, but the computational graph is adapted to compute $XW + XAB$ instead of $XW$. S-LoRA stores adapter weights in the main memory and only loads the LoRA adapters required for the next batch into the device memory, where a unified paging mechanism jointly manages both KV cache and adapter weights. Optimized kernels align with the paging mechanism to gather adapter weights and perform LoRA computations, allowing to batch LoRA computations with different ranks and sequence lengths. Finally, S-LoRA introduces a tensor parallelism strategy to parallelize LoRA computations over multiple devices. In this paper, we propose to alter S-LoRA's TP strategy after imposing a constraint on the LoRA adapters, allowing us to completely eliminate the TP strategy's communication overhead. The other techniques used by S-LoRA are not affected by our alteration to the sharding strategy.

We use the basic version of the MLP module in the transformer architecture to present the S-LoRA sharding strategy. It is a single hidden-layer MLP comprising two weight matrices $W_1 \in \mathbb{R}^{d_H \times d_I}$ and $W_2 \in \mathbb{R}^{d_I \times d_H}$ interleaved by an element-wise non-linearity $\sigma$, that is input $X \in \mathbb{R}^{S \times d_H}$ is transformed to $\sigma(XW_1)W_2$, where $d_H$ is the LLM's hidden dimension, $d_I$ is the LLM's intermediate dimension and $S$ is the number of tokens (i.e., batch size $\times$ sequence length). As illustrated in the top part of Figure 2, the Megatron-LM TP strategy shards $W_1$ column-wise and $W_2$ row-wise (different shards go to different devices). Assuming a copy of $X$ resides on every device, after processing weight shards independently on every device, a single *all-reduce* operation is sufficient to obtain a copy of the MLP's output on every device. The bottom part of Figure 2 illustrates the S-LoRA sharding strategy, which builds on top of the Megatron-LM TP strategy. It shards both LoRA factors in the adapter for $W_1$, denoted by $A_1 \in \mathbb{R}^{d_H \times r}$ and $B_1 \in \mathbb{R}^{r \times d_I}$, column-wise, and shards the LoRA factors in the adapter for $W_2$, denoted by $A_2 \in \mathbb{R}^{d_I \times r}$ and $B_2 \in \mathbb{R}^{r \times d_H}$, row-wise and column-wise, respectively. Here and in the following $r$ represents the rank of the LoRA adapters. The S-LoRA sharding strategy requires an *all-gather* operation after the multiplication of the shards

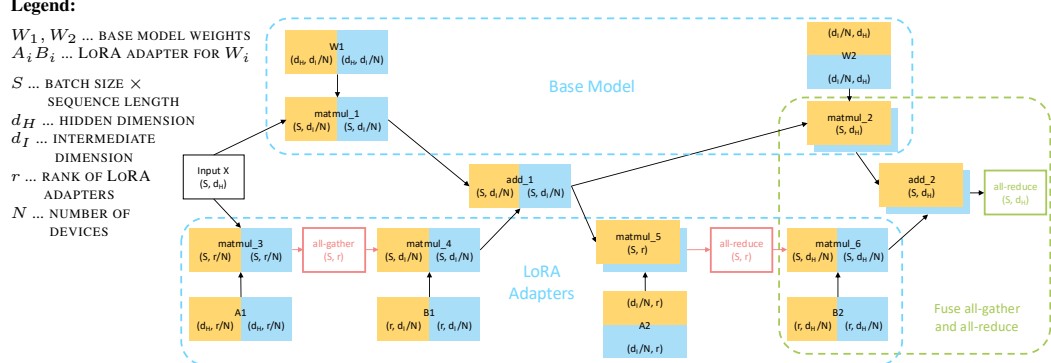

Figure 2: The TP strategy proposed in S-LoRA [28, Figure 4] to parallelize the LoRA computations (lower block). It aligns well with the Megatron-LM TP strategy for the base model (upper block), but requires additional *all-gather* and *all-reduce* operations (red boxes). Different colors represent tensor shards residing on different devices. The figure applies to both attention and MLP modules and does not show operations not relevant for the S-LoRA sharding strategy (e.g., element-wise operations).

of $A_1$ with $X$ (i.e., after *matmul_3* in Figure 2) and an *all-reduce* operation after the multiplication of the shards of $A_2$ with the respective input $X'$ (i.e., after *matmul_5*). A final *all-gather* operation (after *matmul_6*) is fused with the *all-reduce* operation of the Megatron-LM TP strategy for the base model by adding outputs of the LoRA computations and the base model computations on each device before the base model's *all-reduce*. This fused *all-gather* operation does not create any additional communication cost. Hence, for the basic MLP module considered here the S-LoRA TP strategy adds two communication operations on top of the communication required for the base LLM.

For GLU variants of the MLP module [27] that involve three weight matrices and transform $X$ to $(\sigma(XW_1) \otimes \widehat{W_1})W_2$, with $\otimes$ denoting element-wise multiplication, as used in Llama [32], the S-LoRA sharding strategy is adapted in the obvious way with $\widehat{W_1}$ and its LoRA factors going through the same computations as $W_1$, $A_1$ and $B_1$. In that case the communication overhead of S-LoRA is two *all-gather* and one *all-reduce*. Similarly, when sharding the attention module, the query / key / value projections are treated as $W_1$ and the output projection is treated as $W_2$, and the communication overhead of S-LoRA for the attention module is three *all-gather*[3] and one *all-reduce* operations.

According to Sheng et al. [28], for one attention module, the additional communication cost is $\frac{5(N-1)rS}{N}$, where $N$ is the number of devices. Compared to the cost of $\frac{2(N-1)d_H S}{N}$ required for the base LLM, this additional cost is negligible if $r \ll d_H$, but can be significant if $r$ is somewhat large (e.g., for $r = 256$ as users requested in issue #2847 in vLLM and $d_H = 4096$ as in Llama-3.1-8B, the communication overhead is $> 15\%$). More importantly, the cost estimate of Sheng et al. [28] does not take into account the start time for initiating a communication operation [2]. As a result, the communication overhead of S-LoRA is significant even when the rank is small (as we will see in our experiments in Section 5).

## 3 Block-diagonal LoRA

With block-diagonal LoRA (BD-LoRA), we propose a variant of LoRA and an alternative to the S-LoRA tensor parallelism strategy that completely eliminates S-LoRA's communication overhead. The key principle of our approach is that a column-sharded input $X$ multiplied by a block-diagonal matrix $W$[4], where every block sits on one device, yields a column-sharded output without requiring

---

[3]In practice, like it is done in vLLM, the computations for $W_1$ and $\widehat{W_1}$ or query / key / value projections can be merged, and then one does not need two or three *all-gather* operations, but only one *all-gather* operation of twice or thrice the size.

[4]We use the term "block-diagonal" as defined in [7], which does not require $W$ to be square or symmetric.

Figure 3: The TP strategy in our proposed BD-LoRA method. Unlike S-LoRA depicted in Figure 2, in BD-LoRA the LoRA factors $B_1$ and $A_2$ are constrained to be block-diagonal matrices. Our design eliminates the *all-gather* and *all-reduce* operations, marked in red, that S-LoRA requires.

any communication. That is, for $X = [X^1 | X^2 | \cdots | X^N] \in \mathbb{R}^{S \times d_{in}}$ and

$$
W = \begin{bmatrix} W^1 & 0 & \cdots & 0 \\ 0 & W^2 & 0 & \cdots \\ 0 & 0 & \ddots & 0 \\ 0 & \cdots & 0 & W^N \end{bmatrix} \in \mathbb{R}^{d_{in} \times d_{out}}
$$

with $X^i \in \mathbb{R}^{S \times \frac{d_{in}}{N}}$ and $W^i \in \mathbb{R}^{\frac{d_{in}}{N} \times \frac{d_{out}}{N}}$ hosted on device $i$, we can compute $XW = [X^1 W^1 | X^2 W^2 | \cdots | X^N W^N] \in \mathbb{R}^{S \times d_{out}}$ with $X^i W^i \in \mathbb{R}^{S \times \frac{d_{out}}{N}}$ hosted on device $i$ in parallel on the $N$ devices and without any communication. Note that the zeros in $W$ are not touched when computing $XW$ and therefore also do not need to be loaded or stored in any way.

This principle allows us to introduce block-diagonality constraints on some of the LoRA factors $A_1, B_1, A_2, B_2$ (using the notation of Section 2) and subsequently shard them in a way that is aligned with the Megatron-LM sharding of the base model's weights and that does not introduce any additional communication operations. Concretely, if we shard $A_1$ column-wise and require $B_1$ to be block-diagonal, the intermediate LoRA result $XA_1B_1$ is column-sharded and aligned with the intermediate and column-sharded base model result $XW_1$. Hence, we can compute the intermediate result $Y = XW_1 + XA_1B_1$ on every device without requiring any communication. Next, if we require $A_2$ to be block-diagonal and shard $B_2$ row-wise, we can compute $YA_2B_2$ on every device without requiring any communication and obtain distributed results that can be added to the distributed $YW_2$ from the base model computation before the base model's *all-reduce* operation yields the final result. Figure 3 provides an illustration of our BD-LoRA sharding strategy. We discuss how to enforce the block-diagonality constraints during adapter training below.

Constraining $B_1$ and $A_2$ to be block-diagonal reduces their expressiveness. However, we can compensate for the reduced expressiveness by increasing the rank $r$ such that our BD-LoRA adapters have a similar number of effective (i.e., non-zero) parameters as standard LoRA adapters. As we show in our experiments, for a similar number of effective parameters standard LoRA and BD-LoRA adapters achieve very similar downstream performance (cf. Section 5.1). At the same time, for a similar number of effective parameters both S-LoRA and BD-LoRA have similar compute and memory costs (cf. Appendix A), but since BD-LoRA does not require any communication, overall it is significantly faster than S-LoRA (cf. Section 5.2).

**Alternative interpretation and BD-LoRA adapter training** From Figure 3 it is not hard to see that BD-LoRA is equivalent to adding independent LoRA adapters of rank $r/N$ to every shard of the base model's weights (it is also easy to see from the algorithmic description of BD-LoRA that we provide in Appendix B). This interpretation removes the block-diagonality constraints from the picture and provides a straightforward way to implement BD-LoRA fine-tuning in libraries supporting standard LoRA fine-tuning, e.g., Hugging Face Transformers [40] and PEFT [21]: we simply rewrite the model architecture such that weight shards become separate model weights that we can attach

LoRA adapters to. It also resolves the question how to combine BD-LoRA with rsLoRA, i.e., how to set the scaling factor in training: since we train independent adapters of rank $\frac{r}{N}$, we set it as $\frac{\alpha\sqrt{N}}{\sqrt{r}}$.

**Knowledge of $N$ at training time** One limitation of BD-LoRA is that we need to know the number of devices $N$ (aka TP degree) already when training the BD-LoRA adapters. Note that this is not a big restriction since for each LLM and hardware configuration there is usually a typical TP degree that is used for serving the LLM (e.g., TP $= 8$ when serving Llama-X-70B on NVIDIA DGX A100 servers). Furthermore, conceptually it is easy to adapt BD-LoRA to be "downward compatible" (i.e., allowing adapters trained for a larger TP degree to be served with smaller TP degree): putting memory constraints aside, we can always run the workloads of $N_h$ many devices on only $N_l < N_h$ many devices. For BD-LoRA serving, which only involves matrix multiplications and additions, this could be done by stacking the computations of different devices along a new tensor dimension. However, implementing such variant would require to adjust the base model sharding accordingly and presumably to write new efficient kernels, and hence we leave such variant to future work.

## 4  Additional related work

**Punica** [4] is concurrent work with S-LoRA [28] and like S-LoRA a system for multi-LoRA serving. Its key innovation is the design of a new CUDA kernel to batch computations for different LoRA adapters, which some of the kernels introduced in S-LoRA are based on. Punica does not discuss tensor parallelism for the LoRA computations, which is the focus of our paper.

**Reducing communication overhead** Numerous papers have found communication to be a bottleneck in LLM training or inference and propose techniques to overlap communication with computation [e.g., 37, 3, 12, 25, 36]. Overlapping communication with computation is hard whenever the communication is *blocking*, and these papers rely on fine-grained decompositions and low-level interventions to break dependencies. In contrast, Zhang et al. [43] propose a simple architectural change to the transformer block that allows to overlap communication with computation. They also introduce an alternative modification termed Desynced Residual that completely eliminates some communication operations. The latter is closely related to our paper in that we modify the architecture of LoRA adapters attached to transformer modules to completely eliminate S-LoRA's blocking communication.

**LoRA variants** Numerous variants of LoRA [e.g., 42, 44, 33, 15, 17] and refinements of the LoRA training [e.g., 45, 9, 22, 41] have been developed, most of which are compatible with our proposed BD-LoRA approach in the sense that we can replace LoRA with BD-LoRA in these variants and eliminate the communication overhead one would encounter if using S-LoRA sharding at inference time. In particular, BD-LoRA is fully compatible with Q-LoRA [5] both for training and inference.

**SpartanServe** [31] is a system for efficiently serving many different butterfly orthogonal fine-tuning (BOFT) adapters [18] with the same base LLM. BOFT is a method for parameter-efficient fine-tuning that updates model weights by multiplying with a product of sparse orthogonal matrices (in contrast to adding low-rank factorizations as in LoRA). While the SpartanServe paper reports speed-ups over S-LoRA for Llama-2-7B served on a single GPU, it does not discuss multi-GPU serving and hence is incomparable to BD-LoRA.

## 5  Experiments

We first fine-tune numerous LoRA and BD-LoRA adapters on various downstream tasks to investigate whether the block-diagonality constraint in BD-LoRA negatively impacts downstream performance. We then study the run-time of BD-LoRA in comparison to S-LoRA in vLLM [14].

For our fine-tuning experiments, we used Llama-3.2-1B and Llama-3.1-8B [8] as base LLMs. We performed the fine-tuning for numerous ranks, methods, datasets, TP degrees and random seeds, and with a larger model this would have been infeasible. For our run-time experiments we used Llama-3.1-8B and Llama-3.1-70B. We did not study the run-time for smaller models since those are usually served without tensor parallelism.

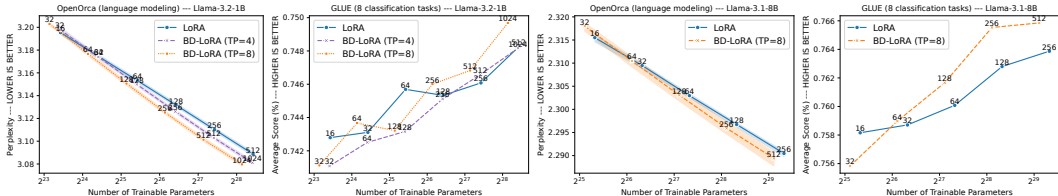

Figure 4: Downstream performance for standard LoRA and BD-LoRA adapters (both with rsLoRA scaling) on Llama-3.2-1B (1st & 2nd plot) and Llama-3.1-8B (3rd & 4th plot) as a function of the number of trainable parameters. For a similar number of parameters, the performance is very similar between the two variants. The plots for OpenOrca (1st & 3rd plot) provide confidence intervals obtained from running experiments for three or five different random seeds. The plots for GLUE (2nd & 4th plot) provide average results over different tasks and different seeds. Plots for the individual tasks with confidence intervals are provided in Figure 6 in the appendix.

**Takeaways:** Across datasets, we find that the block-diagonality constraint does not degrade downstream performance—for a similar number of effective parameters both **standard LoRA and BD-LoRA achieve very similar downstream performance** (cf. Section 5.1). In almost all settings (with varying adapter ranks, batch sizes, input and output lengths, TP degrees, and whether we add adapters to both attention and MLP weight matrices or to only either of them), we find that **BD-LoRA provides significant speed-ups over S-LoRA** (cf. Section 5.2).

## 5.1 Fine-tuning performance

**Setup** We fine-tuned LoRA and BD-LoRA adapters with various ranks on the tasks in the GLUE benchmark [35] and on a language modeling task based on the OpenOrca dataset [23].

The GLUE benchmark is a widely used collection of natural language understanding tasks that span various domains. It includes MNLI [39] for inference tasks, SST-2 [30] for sentiment analysis, QNLI [26] for question-answering inference, QQP [11], for identifying duplicate questions, MRPC [6] for paraphrase identification, CoLA [38] for evaluating linguistic acceptability, RTE for inference, and STS-B [1] for measuring textual similarity. All tasks are classification tasks, except for STS-B, which is a regression task. The GLUE benchmark also contains WNLI, but most papers using GLUE [e.g., 10, 44, 9] exclude that task due to its small size, and so do we. GLUE provides an evaluation metric for each task: Matthews correlation for CoLA, Pearson correlation for STS-B and accuracy for all other datasets. OpenOrca [16] comprises language model instructions from the FLAN collection [19] and corresponding GPT-3.5 / GPT-4 completions [24]. The task is language modeling (i.e., next token prediction), and we evaluate performance with perplexity.

We fine-tuned the adapters using Huggingface Transformers [40] and PEFT [21], where we implemented BD-LoRA as described in Section 3. Some more implementation details and hyperparameter choices are provided in Appendix C.1. We ran experiments on AWS g5 instances equipped with NVIDIA A10G GPUs. For the smaller datasets, including MRPC, CoLA, RTE, and STS-B, we ran each experiment ten times with different random seeds (affecting the initialization of adapter weights and data shuffling). For Llama-3.2-1B and the larger datasets, including MNLI, SST-2, QNLI, QQP, and OpenOrca, we ran each experiment five times. For Llama-3.1-8B with the larger datasets, we ran each experiment three times. Recall from Section 3 that the number of devices $N$ used at inference time determines the shape of the block-diagonal LoRA adapters, and hence $N$ needs to be known at training time. For Llama-3.2-1B we consider BD-LoRA adapters for $N = 4$ and $N = 8$ devices. For Llama-3.1-8B we consider BD-LoRA adapters for $N = 8$ devices. In the following, we refer to the number of devices also as the TP degree (and write, e.g., TP = 4 when $N = 4$). Note that the TP degree has nothing to do with how many devices we use for training BD-LoRA adapters (for training we use data parallelism rather than tensor parallelism).

**Results** Figure 4 provides results for OpenOrca and average results for the GLUE benchmark. Results for the individual tasks in GLUE and when applying LoRA / BD-LoRA adapters only to the attention or MLP weight matrices are provided in Figures 6 to 8 and Table 2 in Appendix E.

For the same rank, BD-LoRA has roughly half the number of trainable parameters as standard LoRA, because some adapter matrices are block-diagonal. For both Llama-3.2-1B and Llama-3.1-8B and across OpenOrca and the eight datasets in the GLUE benchmark, for a similar number of trainable parameters, BD-LoRA achieves performance comparable to standard LoRA. In many cases, BD-LoRA even seems to outperform LoRA, and in the experiments with Llama-3.2-1B the results with $TP = 8$ seem to be slightly better than those with $TP = 4$. However, confidence intervals heavily overlap (cf. Figure 6), and we cannot consider any method superior to another one. These conclusions also hold when applying LoRA / BD-LoRA only to the weight matrices in the attention or MLP module (cf. Appendix E.2).

In these experiments, we used rsLoRA scaling (cf. Section 2), with which the performance of both LoRA and BD-LoRA increases with the rank. This is in contrast to standard scaling, with which both LoRA and BD-LoRA achieve nearly identical results across all ranks (cf. Appendix E.3).

## 5.2 Run-time performance

**Setup**  For our run-time evaluation, we implemented BD-LoRA inference in vLLM [14]. We provide some information about our implementation in Appendix C.2.

We used LLMPerf[5] to send requests to a vLLM server hosting a base LLM and one or more LoRA adapters and measure latency and throughput. We considered different inference settings corresponding to different use cases with different numbers of input token (IT) length, output token (OT) length, and batch size (BS). In each iteration, LLMPerf sends a batch of BS many requests to the vLLM server and waits for all requests to be completed before sending the next batch, where we sent a total of 128 requests in each experiment. For each request, vLLM is restricted to generate a fixed number of OT many output tokens given a prompt of input token length IT. We mainly considered the case where all requests use the same LoRA / BD-LoRA adapter, and we ran most experiments using eight devices (i.e., $TP = 8$) and applying adapters in both attention and MLP modules. When there is only a single adapter, both within a batch and across batches, this adapter gets cached in the GPUs and running time is not affected by adapter (un-)loading. This setting is somewhat artificial as a single adapter would best be served via weight merging. However, it best demonstrates BD-LoRA's innovation, which is the elimination of the communication overhead in S-LoRA's sharding strategy, and its results are also valid in the scenario where we have a small number of different adapters (whether within or across batches), all of which get cached. We also considered the case where every request uses a different adapter (thus completely preventing adapter caching—the extreme real-world setting), applied adapters to attention or MLP weight matrices only, and ran an ablation with different TP degrees.

We ran experiments on an AWS p4d instance equipped with eight NVIDIA A100 GPUs (each with 40 GB HBM2) and NVIDIA NVLink Switch that provides fast all-to-all GPU communication at 600 GB/s bidirectional throughput. For an ablation with different hardware we ran some experiments on a less powerful AWS g5 instance equipped with eight NVIDIA A10G GPUs (each with 24 GB HBM2) and PCIe interconnect with 100 GB/s networking throughput.

**Baseline**  To provide the full picture, we included *Not-Fully-Sharded-LoRA (NFS-LoRA)* as baseline in our run-time evaluation. NFS-LoRA is the default LoRA serving option in vLLM and was the only serving option before S-LoRA was integrated. In NFS-LoRA, adapters $A_1$ and $B_2$ are not sharded accross devices, but replicated so that each device maintains a full copy. The number of communication operations in NFS-LoRA is the same as in BD-LoRA since the full copies of $A_1$ and $B_2$ avoid the inter-device communication introduced by S-LoRA. However, NFS-LoRA consumes $N$ times more memory and computation for $A_1$ and $B_2$ compared to fully shared LoRA (i.e., S-LoRA or BD-LoRA), which can become a significant drawback when serving LoRA adapters with large rank or a large number of adapters or when handling large batch sizes or sequence lengths.

**Results with a single adapter and $TP = 8$**  Figure 5 provides results for Llama-3.1-70B with all requests using the same adapter, which gets cached in the GPUs, and when $IT = 1024$, $OT = 128$ and $BS = 1$ (1st row) or $BS = 64$ (2nd row). Here, we applied adapters to both attention and MLP weight matrices. The results for both Llama-3.1-8B and Llama-3.1-70B and numerous other settings are provided in Appendix F.

---

[5]https://github.com/ray-project/llmperf

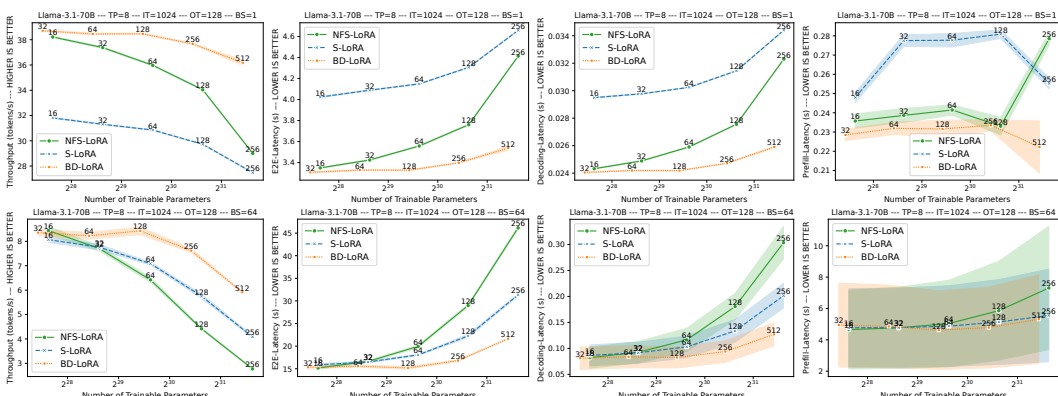

Figure 5: Throughput (1st column—higher is better), end-to-end (E2E) latency (2nd column—lower is better), decoding latency (3rd column—lower is better), and prefill latency (4th column—lower is better) of Llama-3.1-70B served with various LoRA adapters (Not-Fully-Sharded LoRA in green, S-LoRA in blue, BD-LoRA in orange) as a function of the number of trainable parameters. In these plots we have an input token (IT) length of 1024, output token (OT) length of 128, and batch size (BS) of 1 (1st row) or 64 (2nd) row, and all requests use the same adapter. We provide additional plots for several other settings in Appendix F, but the results are largely consistent across settings: BD-LoRA significantly outperforms S-LoRA for small batch size or medium to large ranks. It outperforms the baseline of not-fully-sharded LoRA (NFS-LoRA) whenever the rank is not very small. The speed-up of BD-LoRA mainly comes from a speed-up in decoding whereas in prefill we often do not see any significant discrepancy between the three methods (the confidence intervals, which show a large variation across different requests despite all requests having the same input length, heavily overlap).

The plots show the throughput (# output tokens / total time), end-to-end latency (E2E latency; total time), decoding latency (time per output token), and prefill latency (time to first output token) as a function of the number of trainable parameters and averaged over 128 requests. The parameter range corresponds to ranks from 16 to 256 for standard LoRA adapters (S-LoRA and NFS-LoRA) and 32 to 512 for BD-LoRA adapters. Note that a rank of 256 for standard LoRA is not particularly large and much smaller than the ranks of up to 2048 considered by Kalajdzievski [13] and that users of vLLM have requested the support of ranks up to 256 when only ranks up to 64 were supported (cf. issue #2847 in vLLM).

In almost all settings and for both Llama-3.1-8B and Llama-3.1-70B, BD-LoRA demonstrates the highest throughput and the lowest E2E and decoding latency compared to NFS-LoRA and S-LoRA. In particular, BD-LoRA significantly outperforms S-LoRA for small batch size or medium to large ranks, and it outperforms NFS-LoRA whenever the rank is not very small. The speed-up of BD-LoRA over S-LoRA mainly comes from a speed-up in decoding (and hence is largest in generation-heavy tasks with small IT and large OT; cf. Appendix F). This is because during decoding the data size to be communicated is very small, hence, the communication kernels' latency is essentially a fixed overhead. This overhead plays a large role during decoding. During prefill, the communication data sizes are larger, but so are the compute latencies of the matrix multiplications. Overall, during prefill the communication overhead is not as significant as during decoding. Empirically, BD-LoRA usually has the lowest prefill latency, but for the prefill latency the confidence intervals are typically very large and heavily overlap so that we cannot consider any method to be superior. As the rank increases, both NFS-LoRA and S-LoRA show a strong increase in latency. In contrast, BD-LoRA's latency grows slower with the rank, showcasing its higher efficiency at larger ranks, which yield better downstream performance when using rsLoRA scaling [13]. The performance of NFS-LoRA heavily depends on the batch size. It outperforms S-LoRA for smaller batch sizes, where S-LoRA's communication overhead mainly comes from the communication startup time and sufficient resources are available to handle NFS-LoRA's increased memory and computation requirements, but is inferior to S-LoRA for larger batch sizes.

These findings hold true when applying adapters in only attention or MLP modules, where we see a larger speed-up of BD-LoRA over S-LoRA in the first case (cf. Appendix F.3) as there S-LoRA's communication overhead is larger (cf. Section 2).

**Results with different adapters for every request** When every request uses a different adapter and adapters cannot be cached in memory, adapters have to be loaded from disk before being used. This adds a constant amount of time to a request's prefill and E2E latency, which depends on the adapter's number of parameters, but is the same for NFS-LoRA, S-LoRA and BD-LoRA. Hence the curves look similar as the ones discussed above but are shifted, and the overall speed-up numbers of BD-LoRA are reduced, as it shares the same additional overhead of adapter loading as NFS-LoRA and S-LoRA. Nevertheless, BD-LoRA still achieves a speedup of up to 1.78x. (cf. Appendix F.4).

**Results for different TP degrees** We investigated the effect of the TP degree for Llama-3.1-8B. Plots can be found in Appendix F.2. We see that for smaller TP degrees the performance differences between NFS-LoRA, S-LoRA, and BD-LoRA are less significant. This is expected since for smaller TP degrees NFS-LoRA copies LoRA adapters to fewer devices and S-LoRA communicates between fewer devices. Still, BD-LoRA consistently outperforms NFS-LoRA and S-LoRA. Note that for all methods run-time performance goes down as the TP degree gets smaller, making TP = 8 the fastest option for serving Llama-3.1-8B on one AWS p4d instance. These results suggest that BD-LoRA's advantage is likely to widen at larger TP degrees (e.g., 16 or 32), highlighting its scalability potential.

**Results on different hardware** We ran some experiments with Llama-3.1-8B on an AWS g5 instance equipped with eight NVIDIA A10G GPUs with PCIe interconnect. Note that compared to A100 GPUs, A10G GPUs are not only slower in terms of communication, but also in terms of compute and have less memory. Results are provided in Appendix F.5. We observe BD-LoRA speed-ups over S-LoRA of up to 1.36x (1.27x) with 0.86x (1.73x) number of adapter parameters, demonstrating BD-LoRA's effectiveness also on less powerful hardware.

**Results with quantized base model** BD-LoRA and our vLLM implementation also work out of the box with quantized backbone models as required for Q-LoRA [5]. To illustrate this we include performance results for one setting with a quantized Llama-3.1-8B model in Appendix F.6. We find that at better accuracy, serving BD-LoRA on top of the quantized model gives E2E speedups of 1.26x over S-LoRA and 1.17x over NFS-LoRA (both also with quantized model).

## 6    Discussion

We proposed BD-LoRA as a variant / architectural change of LoRA that allows to exploit tensor parallelism in multi-LoRA serving without any communication overhead for the LoRA computations. BD-LoRA provides an alternative to the S-LoRA [28] sharding strategy, for which the communication overhead can be significant. We demonstrated that for a similar number of parameters, S-LoRA and BD-LoRA achieve very similar downstream performance, but BD-LoRA runs significantly faster. We expect the run-time savings of BD-LoRA to be even more pronounced when combined with further optimizations of the base model, in which case communication operations likely make up an even bigger fraction of the overall run-time, but leave an investigation to future work.

One limitation of BD-LoRA is that we need to know the number of devices already at training time. Another limitation is that we need to train BD-LoRA adapters from scratch even when standard LoRA adapters already exist. While we discussed that the first limitation could be mitigated by a modified implementation, we consider it an interesting question for future work how to efficiently train BD-LoRA adapters starting from existing standard LoRA adapters.

## Acknowledgments

We thank Alexander Conzelmann for valuable comments on the camera-ready version of the paper, as well as for upgrading and open-sourcing our code.

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

Table 1: The number of parameters per LoRA factor and per device and the associated compute cost (number of floating-point operations) and memory cost (number of floating-point elements to be moved) for S-LoRA and BD-LoRA. We use $r$ to denote the rank for S-LoRA and $r'$ to denote the rank for BD-LoRA. S-LoRA and BD-LoRA have the same number of parameters when $r' = r(d_H + d_I)/(d_H + \frac{d_I}{N})$.

| | LoRA factor | # parameters | # operations | Cost of writing and loading output |
|---|---|---|---|---|
| | $A_1$ | $d_H \cdot \frac{r}{N}$ | $2 \cdot S \cdot d_H \cdot \frac{r}{N}$ | $2 \cdot S \cdot \frac{r}{N}$ |
| | $B_1$ | $d_I \cdot \frac{r}{N}$ | $2 \cdot S \cdot d_I \cdot \frac{r}{N}$ | $2 \cdot S \cdot \frac{d_I}{N}$ |
| S-LoRA | $A_2$ | $d_I \cdot \frac{r}{N}$ | $2 \cdot S \cdot d_I \cdot \frac{r}{N}$ | $2 \cdot S \cdot r$ |
| | $B_2$ | $d_H \cdot \frac{r}{N}$ | $2 \cdot S \cdot d_H \cdot \frac{r}{N}$ | $2 \cdot S \cdot \frac{d_H}{N}$ |
| | SUM | $2 \cdot (d_H + d_I) \cdot \frac{r}{N}$ | $4 \cdot S \cdot (d_H + d_I) \cdot \frac{r}{N}$ | $2 \cdot S \cdot (\frac{r}{N} + \frac{d_I}{N} + \frac{d_H}{N} + r)$ |
| | $A_1$ | $d_H \cdot \frac{r'}{N}$ | $2 \cdot S \cdot d_H \cdot \frac{r'}{N}$ | $2 \cdot S \cdot \frac{r'}{N}$ |
| | $B_1$ | $\frac{d_I}{N} \cdot \frac{r'}{N}$ | $2 \cdot S \cdot \frac{d_I}{N} \cdot \frac{r'}{N}$ | $2 \cdot S \cdot \frac{d_I}{N}$ |
| BD-LoRA | $A_2$ | $\frac{d_I}{N} \cdot \frac{r'}{N}$ | $2 \cdot S \cdot \frac{d_I}{N} \cdot \frac{r'}{N}$ | $2 \cdot S \cdot \frac{r'}{N}$ |
| | $B_2$ | $d_H \cdot \frac{r'}{N}$ | $2 \cdot S \cdot d_H \cdot \frac{r'}{N}$ | $2 \cdot S \cdot d_H$ |
| | SUM | $2 \cdot (d_H + \frac{d_I}{N}) \cdot \frac{r'}{N}$ | $4 \cdot S \cdot (d_H + \frac{d_I}{N}) \cdot \frac{r'}{N}$ | $2 \cdot S \cdot (\frac{2r'}{N} + \frac{d_I}{N} + d_H)$ |

# A  Compute and Memory Costs of S-LoRA and BD-LoRA

We analyze the compute and memory costs of S-LoRA and BD-LoRA for one basic MLP module as considered in Section 2 and Section 3, respectively. For following our analysis, we recommend to refer to the illustrations of S-LoRA and BD-LoRA in the main part of the paper: the S-LoRA sharding strategy for a basic MLP module is illustrated in Figure 2, and the BD-LoRA sharding strategy for a basic MLP module is illustrated in Figure 3.

For both S-LoRA and BD-LoRA the compute cost is the cost of *matmul_3*, *matmul_4*, *matmul_5* and *matmul_6* (corresponding to matrix multiplications with LoRA factors $A_1$, $B_1$, $A_2$, $B_2$) and of *add_1* and *add_2* (corresponding to adding outputs of the LoRA computations and the base model computations) in Figure 2 and Figure 3, respectively. We can see from Table 1 that for both S-LoRA and BD-LoRA the cost of the matrix multiplications is $2 \cdot S \cdot$ # parameters. Hence, for the same number of effective parameters S-LoRA and BD-LoRA incur the same compute cost for the matrix multiplications. The cost of *add_1* is $S\frac{d_I}{N}$ floating-point operations per device for both S-LoRA and BD-LoRA. The cost of *add_2* is $S\frac{d_H}{N}$ floating-point operations per device for S-LoRA and $Sd_H$ for BD-LoRA. The costs of these additions are negligible compared to the cost of the matrix multiplications as stated in Table 1 (note that $r \geq N$ and typically $d_I > d_H$, e.g., $d_H = 4096$ and $d_I = 14336$ in Llama-3.1-8B), and S-LoRA and BD-LoRA incur similar compute cost overall.

The memory cost is the cost for loading model weights and inputs and for loading and writing intermediate results. The cost for loading the input (input $X$ of size $S \times d_H$ in both Figure 2 and Figure 3) is the same for S-LoRA and BD-LoRA. For the same number of effective parameters, the cost for loading model weights is also the same. For S-LoRA the cost for loading and writing intermediate results is

$$\underbrace{2 \cdot S \cdot \left( \frac{r}{N} + \frac{d_I}{N} + \frac{d_H}{N} + r \right)}_{\text{intermediate results from } matmul \text{ as in Table 1}} + \underbrace{2 \cdot S \cdot r}_{\text{all-gather}} + \underbrace{2 \cdot S \cdot \frac{d_I}{N}}_{add\_1} + \underbrace{2 \cdot S \cdot r}_{\text{all-reduce}} + \underbrace{2 \cdot S \cdot d_H}_{add\_2} =$$

$$2 \cdot S \cdot \left( \frac{r}{N} + \frac{d_H}{N} \right) + 6 \cdot S \cdot r + 4 \cdot S \cdot \frac{d_I}{N} + 2 \cdot S \cdot d_H.$$

For BD-LoRA the cost for loading and writing intermediate results is

$$\underbrace{2 \cdot S \cdot \left( \frac{2r'}{N} + \frac{d_I}{N} + d_H \right)}_{\text{intermediate results from } \textit{matmul} \text{ as in Table 1}} + \underbrace{2 \cdot S \cdot \frac{d_I}{N}}_{add\_1} + \underbrace{2 \cdot S \cdot d_H}_{add\_2} = 4 \cdot S \cdot \frac{r'}{N} + 4 \cdot S \cdot \frac{d_I}{N} + 4 \cdot S \cdot d_H,$$

and the absolute difference between these two is

$$\left| 4 \cdot S \cdot \frac{r'}{N} + 2 \cdot S \cdot d_H - 2 \cdot S \cdot \frac{r}{N} - 2 \cdot S \cdot \frac{d_H}{N} - 6 \cdot S \cdot r \right| \leq$$

$$2 \cdot \frac{S}{N} \cdot |2r' - r| + 2 \cdot S \cdot \left| d_H - \frac{d_H}{N} - 3r \right|.$$

For the number of parameters to be the same, we have

$$r' = r \frac{d_H + d_I}{d_H + \frac{d_I}{N}}$$

and hence

$$1 < r' < r \left( 1 + \frac{d_I}{d_H} \right).$$

In a typical setting we have $d_I = c \cdot d_H$ for some rather small $c$ (e.g., $c = 3.5$ in Llama-3.1-8B), $N \geq 2$ being of similar size or larger and $r \ll d_H$ such that $d_H/N + 3r \ll d_H$. It follows that the absolute difference in the cost is not greater than $2 \cdot S \cdot d_H$, which is as small as the memory cost incurred for loading the input and writing the final result in the base model computation.

## B    Algorithms for BD-LoRA

In the main paper, we described BD-LoRA and illustrated it in Figure 3. In this section, we further include an algorithmic description in Algorithm 1. From the algorithm it is clear that lines 6-16 allow for full parallelization without the need to communicate. Furthermore, the algorithm facilitates the interpretation that we have presented in Section 3. In the main **for** loop, each device processes a standard (unsharded) LoRA-adapted MLP of intermediate size $d_I/N$ and with rank $r/N$. We can thus interpret BD-LoRA as adding independent LoRA adapters for each GPU.

Our main paper focuses on the full MLP computation as it is prevalent to LLMs, which couples to matrix multiplications. However, in general, the idea of BD-LoRA can be applied to the adaptation of any linear layer in deep neural networks. We emphasize that the structure of the adapters should always be based on the parallelism design of the serving configuration. For a general matrix multiplication, for example, let us assume the setting that full inputs and outputs should be replicated across devices and that the weights are column-sharded (sharded along the output dimension). Then the backbone model's distributed matrix multiplication would require an all-gather after the sharded matrix multiplication. In this case BD-LoRA would have the $B$ matrix block diagonal, i.e., the constraint is the same as for the first matrix multiplication in the Megatron style approach. The $A$ matrix is dense and regularly column-sharded. This approach would hence not introduce any additional communication beyond the all-gather required by the backbone. We illustrate this for $N$ shards in Algorithm 2.

## C    Implementation and Hyperparameter Details

### C.1    Fine-tuning Experiments

**Implementation**    We implemented the training of BD-LoRA adapters within Huggingface Transformers[6] & PEFT[7]. We modified the Huggingface Transformers source code that implements the Llama architecture to split each projection layer into $N$ shards and then applied standard LoRA to these shards using PEFT. PEFT allows the addition of a LoRA adapter to any linear layer. For each shard of the linear layer in the Attention module and / or MLP module, we added a LoRA adapter with a rank of $r' = \frac{r}{N}$ (cf. Section 3).

---

[6]https://huggingface.co/docs/transformers Version: 4.46.2.
[7]https://huggingface.co/docs/peft Version: 0.12.0.

**Algorithm 1** BD-LoRA Forward Pass for Megatron-Style MLP with Tensor Parallelism (Figure 3)

**Require:**
1: Input $X \in \mathbb{R}^{B \times d_H}$ (replicated across $N$ devices)
2: Base weights $W_1^{(i)} \in \mathbb{R}^{d_H \times d_I/N}$, $W_2^{(i)} \in \mathbb{R}^{d_I/N \times d_H}$ (sharded column-and row-parallel)
3: BD-LoRA factors $A_1^{(i)} \in \mathbb{R}^{d_H \times r/N}$, $B_1^{(i)} \in \mathbb{R}^{r/N \times d_I/N}$ (with $B_1$ block-diagonal)
4: BD-LoRA factors $A_2^{(i)} \in \mathbb{R}^{d_I/N \times r/N}$ (with $A_2^{(i)}$ block-diagonal), $B_2^{(i)} \in \mathbb{R}^{r/N \times d_H}$
5: **for** each device $i \in \{1, \ldots, N\}$ **in parallel do**
6:  *// First linear projection (base + BD-LoRA)*
7:  $Y_1^{(i)} \leftarrow X W_1^{(i)}$           ▷ Base model contribution
8:  $Z_1^{(i)} \leftarrow X A_1^{(i)}$           ▷ Local adapter projection
9:  $Y_{1,\text{adapter}}^{(i)} \leftarrow Z_1^{(i)} B_1^{(i)}$      ▷ Block-diagonal $B_1$ ensures no communication
10:  $Y_1^{(i)} \leftarrow Y_1^{(i)} + Y_{1,\text{adapter}}^{(i)}$
11:  *// Second linear projection (base + BD-LoRA)*
12:  $Y_2^{(i)} \leftarrow Y_1^{(i)} W_2^{(i)}$
13:  $Z_2^{(i)} \leftarrow Y_1^{(i)} A_2^{(i)}$       ▷ Block-diagonal $A_2$ ensures no communication
14:  $Y_{2,\text{adapter}}^{(i)} \leftarrow Z_2^{(i)} B_2^{(i)}$
15:  $Y_2^{(i)} \leftarrow Y_2^{(i)} + Y_{2,\text{adapter}}^{(i)}$
16: **end for**
17: *// Final aggregation (only standard Megatron all-reduce)*
18: $Y \leftarrow \text{AllReduce}(\{Y_2^{(1)}, \ldots, Y_2^{(N)}\})$
19: **return** $Y$ (replicated across $N$ devices)

---

**Algorithm 2** BD-LoRA Forward Pass for a Single Column-Parallel Linear Layer

**Require:**
1: Input $X \in \mathbb{R}^{B \times d_{in}}$ (replicated across $N$ devices)
2: Base weights $W^{(i)} \in \mathbb{R}^{d_{in} \times d_{out}/N}$ (sharded column-parallel)
3: BD-LoRA factors $A^{(i)} \in \mathbb{R}^{d_{in} \times r/N}$, $B^{(i)} \in \mathbb{R}^{r/N \times d_{out}/N}$ (with $B^{(i)}$ block-diagonal)
4: **for** each device $i \in \{1, \ldots, N\}$ **in parallel do**
5:  $Y_{base}^{(i)} \leftarrow X W^{(i)}$          ▷ Base model contribution
6:  $Z^{(i)} \leftarrow X A^{(i)}$           ▷ Local adapter projection
7:  $Y_{adapter}^{(i)} \leftarrow Z^{(i)} B^{(i)}$      ▷ Block-diagonal $B$ ensures no communication
8:  $Y_{local}^{(i)} \leftarrow Y_{base}^{(i)} + Y_{adapter}^{(i)}$
9: **end for**
10: $Y \leftarrow \text{AllGather}(\{Y_{local}^{(1)}, \ldots, Y_{local}^{(N)}\})$      ▷ Standard all-gather operation
11: **return** $Y$ (replicated across $N$ devices)

---

**Hyperparameters** We set LoRA's $\alpha$ parameter to 16 and used the AdamW optimizer [20] with a linear learning rate schedule and a warmup ratio of 0.05. For the GLUE benchmark, we set the learning rate to $10^{-5}$, enabled early stopping, and restricted the maximum sequence length to 128. For OpenOrca, we followed the settings described in the rsLoRA-paper [13] and fine-tuned on 20,000 examples with a learning rate of $5 \cdot 10^{-5}$ and evaluated on another 20,000 examples. In all experiments, LoRA [10] and BD-LoRA were applied with the same settings.

## C.2 Run-time Experiments

**Implementation** For our run-time evaluation, we integrated BD-LoRA in vLLM[8]. We store the block-diagonal matrices $B_1$ and $A_2$ in our BD-LoRA adapters (using the notation of Figure 3) as matrices of shape $\frac{r}{N} \times d_I$ and $d_I \times \frac{r}{N}$, respectively, by putting blocks next to each other or on top of

---

[8]`https://docs.vllm.ai/` Version: commit c11f172 (between releases v0.6.4.post1 and v0.6.5).

each other. This way we do not store or touch any zeros in the block-diagonal matrices. We modified the slicing code of vLLM to correctly distribute the block-diagonal LoRA adapters across multiple GPUs. After loading the BD-LoRA adapters into GPU memory, all we have to do is to remove the communication operations from the LoRA computations in vLLM to fully implement BD-LoRA. All other components, including LoRA memory management, scheduling, and request handling, are kept unchanged and handled by vLLM in the same way as for S-LoRA.

# D    Licenses

The primary GLUE tasks are built on and derived from existing datasets. Please refer to the original licenses accompanying each dataset. The original GLUE website[9] refers users to the original licenses accompanying each dataset. OpenOrca is licensed under the MIT License[10]. The Llama models are licensed under the LLAMA 3.1 COMMUNITY LICENSE AGREEMENT[11] and LLAMA 3.2 COMMUNITY LICENSE AGREEMENT[12].

# E    Additional Fine-tuning Experiments

## E.1    BD-LoRA Fine-tuning

Table 2 and Figure 6 provides a detailed comparison of Llama-3.2-1B and Llama-3.1-8B [8] on the OpenOrca [23] and GLUE benchmarks [35], as represented in Figure 4. Perplexity is used as the evaluation metric for OpenOrca, where lower is better, while for GLUE, we use Matthew's correlation for CoLA, Pearson correlation for STS-B, and accuracy for other tasks, where higher is better. The confidence intervals overlap significantly, suggesting that BD-LoRA and LoRA perform at a similar level overall. BD-LoRA occasionally surpasses LoRA, particularly on OpenOrca and the average GLUE score. BD-LoRA trained with $TP = 8$ appears to slightly outperform BD-LoRA trained with $TP = 4$. For datasets such as MNLI and SST-2, increasing the rank does not significantly impact performance, even when using rsLoRA. However, for the majority of datasets, increasing the rank continues to enhance performance. For Llama-3.2-1B, the confidence intervals largely overlap across the eight datasets, indicating that BD-LoRA and LoRA achieve similar performance with the same number of trainable parameters. For Llama-3.1-8B, BD-LoRA outperforms LoRA on SST-2 but underperforms on MRPC. SST-2 is a sentiment analysis task with a larger dataset, while MRPC focuses on paraphrase detection and involves a smaller dataset. The task type and dataset size may influence the relative performance of LoRA and BD-LoRA. Nevertheless, for most datasets, the confidence intervals overlap significantly, indicating that BD-LoRA and LoRA perform similarly overall.

## E.2    BD-LoRA with Attention-only and MLP-only adapters Fine-tuning

Figure 7 and Figure 8 show the performance of Llama-3.2-1B using Attention-only and MLP-only adapters for both $TP = 4$ and $TP = 8$ on eight GLUE datasets.

The confidence intervals largely overlap across the evaluated datasets. Figure 7 indicates that, with Attention-only adapters, LoRA achieves better performance on MNLI, while BD-LoRA performs better on QQP. For the remaining datasets, the confidence intervals of the methods exhibit significant overlap, suggesting comparable performance. Similarly, Figure 8 shows that, with MLP-only adapters, LoRA performs better on CoLA and at lower ranks on QNLI.

Although BD-LoRA shows a slightly lower average score, less than 0.4%, at lower ranks in both Attention-only and MLP-only adapters, it demonstrates similar performance to LoRA when both Attention and MLP modules are equipped with LoRA adapters, as shown in Figure 4 and Figure 6. This slight difference in performance at lower ranks could be influenced by the reduced number of trainable parameters, as applying LoRA to only part of a module effectively decreases the parameter count, leading to marginally worse performance for BD-LoRA at lower ranks. However,

---

[9]https://gluebenchmark.com/faq
[10]https://huggingface.co/datasets/Open-Orca/OpenOrca
[11]https://www.llama.com/llama3_1/license/
[12]https://www.llama.com/llama3_2/license/

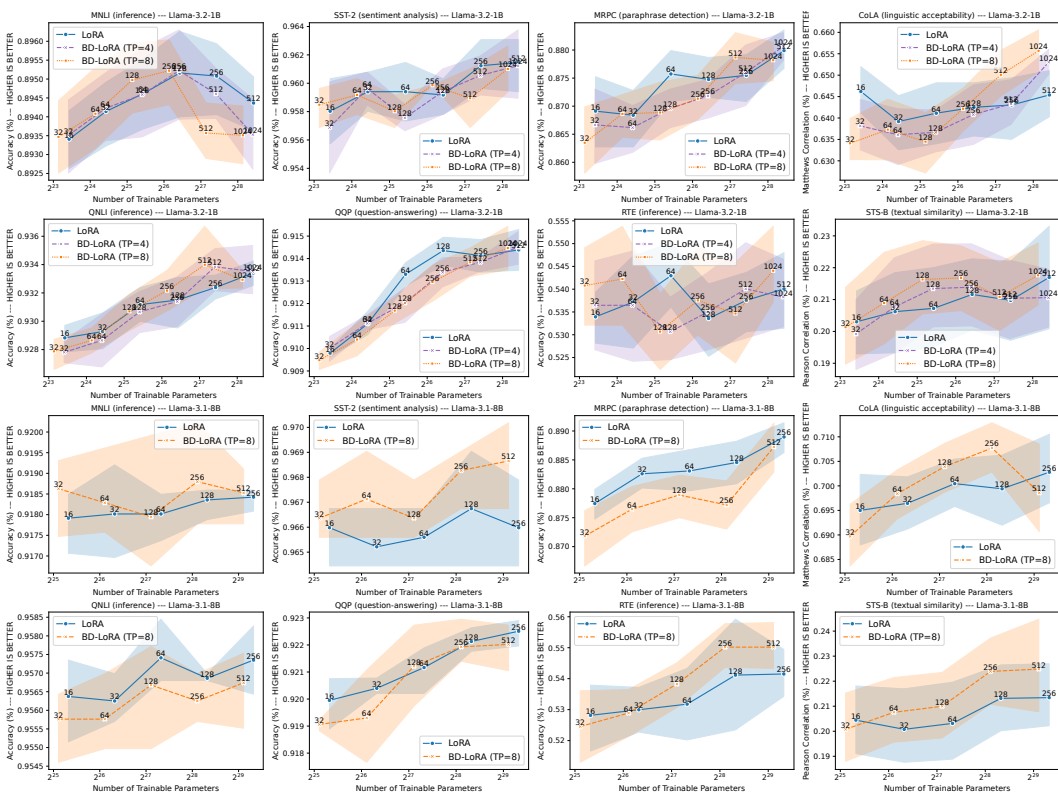

Figure 6: **Fine-tuning results** with confidence intervals obtained from running experiments three or five times with different random seeds for **Llama-3.2-1B** (1st and 2nd row) and **Llama-3.1-8B** (3rd and 4th row) on eight **GLUE datasets**. For Llama-3.2-1B we consider BD-LoRA adapters for TP = 4 and TP = 8 while for Llama-3.1-8B we only consider BD-LoRA adapters for TP = 8.

the confidence intervals for the methods often overlap significantly, as seen with most datasets. The high overlap of confidence intervals across most datasets suggests comparable performance overall, with only a few datasets contributing to the slightly lower average score.

### E.3    rsLoRA Fine-tuning

In this section, we evaluate LoRA and BD-LoRA with standard scaling and rsLoRA scaling [13] using Llama-3.2-1B on OpenOrca as well as four small datasets from GLUE: MRPC, CoLA, RTE, and STS-B, as shown in Figure 9.

With standard scaling, both LoRA and BD-LoRA achieve nearly identical results across all ranks in both OpenOrca and GLUE. However, with rsLoRA scaling, both LoRA and BD-LoRA demonstrate performance improvements as the rank increases on both OpenOrca and GLUE. This indicates that with rsLoRA scaling, larger ranks enhance performance, making higher ranks more effective and capable of achieving better results. Meanwhile, our proposed BD-LoRA method significantly reduces the latency associated with large ranks.

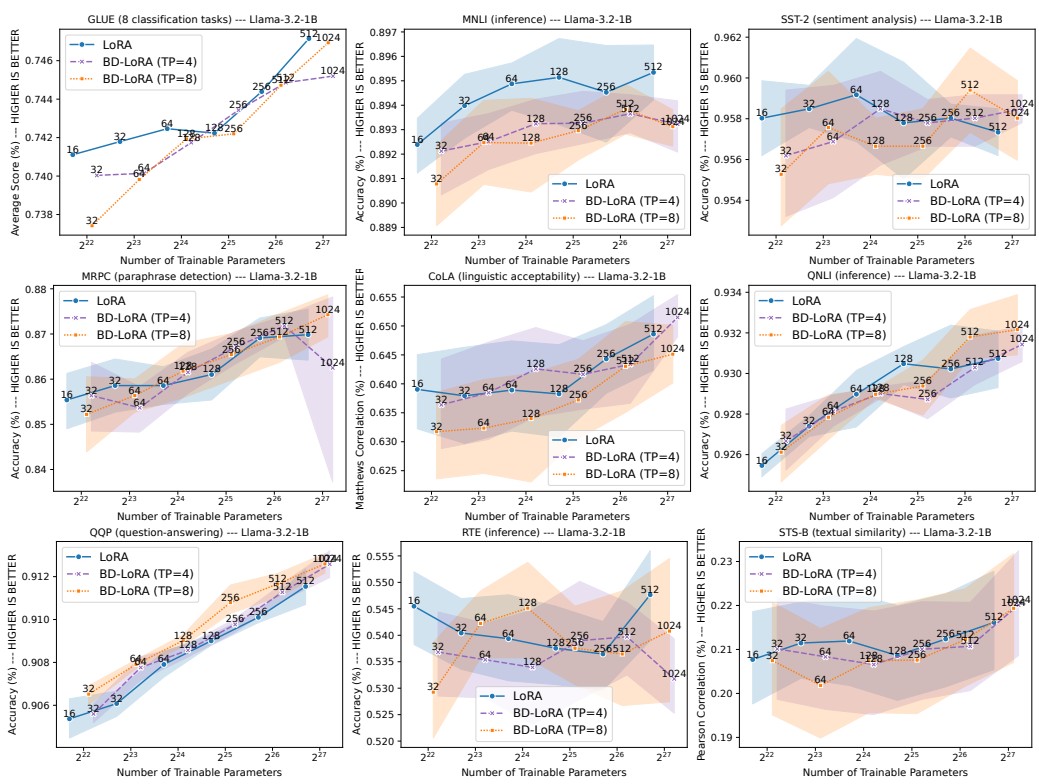

Figure 7: **Fine-tuning results** for **Llama-3.2-1B** on eight **GLUE datasets** when applying LoRA only to the **Attention weight matrices**. The top left plot shows the average performance.

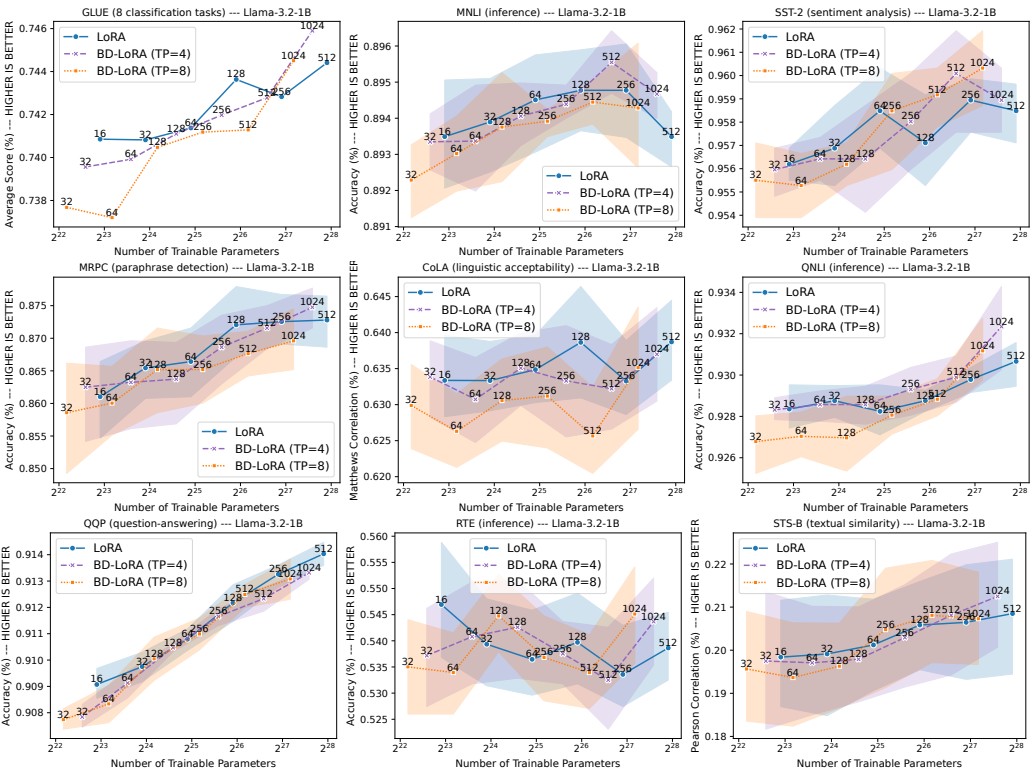

Figure 8: **Fine-tuning results** for **Llama-3.2-1B** on eight **GLUE datasets** when applying LoRA only to the **MLP weight matrices**. The top left plot shows the average performance.

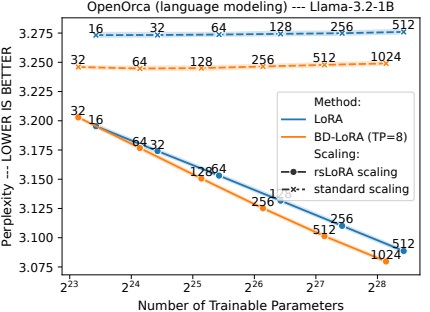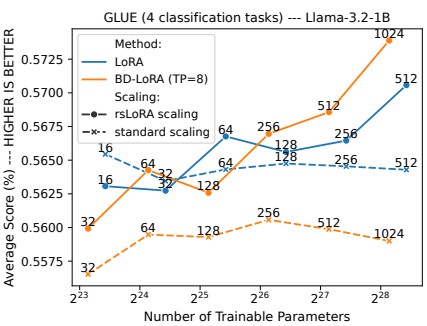

Figure 9: **Comparison of fine-tuning with standard scaling and rsLoRA scaling** on OpenOrca (left plot) and four GLUE tasks (MRPC, CoLA, RTE, and STS-B; right plot) using **Llama-3.2-1B** as base model. With standard scaling, both LoRA and BD-LoRA achieve similar results across all ranks. However, with rsLoRA scaling, both LoRA and BD-LoRA show performance improvements as the rank increases.

Table 2: **Downstream performance** after fine-tuning LoRA and BD-LoRA adapters for **Llama-3.2-1B** and **Llama-3.1-8B** on the OpenOrca dataset and eight GLUE benchmarks. Performance on OpenOrca is measured by perplexity (lower is better). For GLUE, performance on CoLA is measured by Matthew's correlation, on STS-B by Pearson correlation, and on other tasks by accuracy (higher is always better).

| Method | Rank | # Trainable Parameters | OpenOrca ↓ | GLUE ↑ | | | | | | | | |
|---|---|---|---|---|---|---|---|---|---|---|---|---|
| | | | | MNLI | SST-2 | MRPC | CoLA | QNLI | QQP | RTE | STS-B | Avg. |
| **Llama-3.2-1B** | | | | | | | | | | | | |
| LoRA | 16 | 11.3M | 3.20 | $89.3_{\pm0.1}$ | $95.8_{\pm0.3}$ | $86.9_{\pm1.0}$ | $64.6_{\pm0.9}$ | $92.9_{\pm0.1}$ | $91.0_{\pm0.0}$ | $53.4_{\pm0.9}$ | $20.3_{\pm1.7}$ | 74.3 |
| | 32 | 22.5M | 3.17 | $89.4_{\pm0.1}$ | $96.0_{\pm0.1}$ | $86.8_{\pm0.8}$ | $63.9_{\pm1.0}$ | $93.0_{\pm0.1}$ | $91.1_{\pm0.1}$ | $53.7_{\pm1.6}$ | $20.6_{\pm2.1}$ | 74.3 |
| | 64 | 45.1M | 3.15 | $89.5_{\pm0.1}$ | $96.1_{\pm0.2}$ | $87.6_{\pm0.7}$ | $64.1_{\pm1.1}$ | $93.0_{\pm0.1}$ | $91.3_{\pm0.1}$ | $54.3_{\pm1.2}$ | $20.7_{\pm2.0}$ | 74.6 |
| | 128 | 90.2M | 3.13 | $89.5_{\pm0.1}$ | $95.9_{\pm0.2}$ | $87.5_{\pm0.6}$ | $64.3_{\pm1.0}$ | $93.2_{\pm0.1}$ | $91.4_{\pm0.1}$ | $53.4_{\pm1.3}$ | $21.2_{\pm1.8}$ | 74.5 |
| | 256 | 180.4M | 3.11 | $89.5_{\pm0.1}$ | $96.1_{\pm0.2}$ | $87.5_{\pm0.7}$ | $64.3_{\pm1.2}$ | $93.3_{\pm0.1}$ | $91.4_{\pm0.1}$ | $53.8_{\pm1.1}$ | $21.0_{\pm2.1}$ | 74.6 |
| | 512 | 360.7M | 3.09 | $89.4_{\pm0.1}$ | $95.9_{\pm0.1}$ | $88.0_{\pm0.5}$ | $64.5_{\pm0.9}$ | $93.3_{\pm0.1}$ | $91.4_{\pm0.1}$ | $54.0_{\pm1.4}$ | $21.7_{\pm2.6}$ | 74.8 |
| BD-LoRA TP=4 | 32 | 11.1M | 3.20 | $89.4_{\pm0.1}$ | $95.7_{\pm0.4}$ | $86.7_{\pm1.1}$ | $63.8_{\pm0.9}$ | $92.8_{\pm0.1}$ | $91.0_{\pm0.1}$ | $53.6_{\pm1.6}$ | $19.9_{\pm1.8}$ | 74.1 |
| | 64 | 22.3M | 3.17 | $89.4_{\pm0.1}$ | $96.0_{\pm0.1}$ | $86.6_{\pm0.7}$ | $63.6_{\pm1.0}$ | $92.9_{\pm0.2}$ | $91.1_{\pm0.0}$ | $53.6_{\pm1.9}$ | $20.8_{\pm2.4}$ | 74.3 |
| | 128 | 44.6M | 3.15 | $89.5_{\pm0.1}$ | $95.8_{\pm0.1}$ | $87.0_{\pm0.5}$ | $63.7_{\pm0.8}$ | $93.1_{\pm0.2}$ | $91.2_{\pm0.1}$ | $53.1_{\pm1.1}$ | $21.3_{\pm2.1}$ | 74.3 |
| | 256 | 89.1M | 3.13 | $89.5_{\pm0.1}$ | $95.9_{\pm0.2}$ | $87.2_{\pm0.5}$ | $64.1_{\pm1.1}$ | $93.1_{\pm0.2}$ | $91.3_{\pm0.1}$ | $53.5_{\pm1.4}$ | $21.4_{\pm2.1}$ | 74.5 |
| | 512 | 178.3M | 3.10 | $89.5_{\pm0.1}$ | $96.1_{\pm0.2}$ | $87.6_{\pm0.8}$ | $64.3_{\pm0.8}$ | $93.4_{\pm0.2}$ | $91.4_{\pm0.0}$ | $54.0_{\pm1.6}$ | $21.0_{\pm2.1}$ | 74.7 |
| | 1024 | 356.5M | **3.08** | $89.4_{\pm0.1}$ | $96.1_{\pm0.3}$ | $88.1_{\pm0.4}$ | $65.3_{\pm0.5}$ | $93.4_{\pm0.2}$ | $91.5_{\pm0.1}$ | $53.8_{\pm1.2}$ | $21.1_{\pm1.4}$ | 74.8 |
| BD-LoRA TP=8 | 32 | 9.2M | 3.20 | $89.3_{\pm0.1}$ | $95.8_{\pm0.2}$ | $86.3_{\pm1.0}$ | $63.4_{\pm0.8}$ | $92.8_{\pm0.1}$ | $90.9_{\pm0.0}$ | $54.1_{\pm1.4}$ | $20.1_{\pm1.9}$ | 74.1 |
| | 64 | 18.5M | 3.18 | $89.4_{\pm0.1}$ | $95.9_{\pm0.2}$ | $86.9_{\pm0.6}$ | $63.7_{\pm1.1}$ | $92.9_{\pm0.1}$ | $91.0_{\pm0.1}$ | $54.2_{\pm1.7}$ | $20.9_{\pm2.3}$ | 74.4 |
| | 128 | 37.0M | 3.15 | $89.5_{\pm0.1}$ | $95.8_{\pm0.1}$ | $86.9_{\pm0.8}$ | $63.4_{\pm1.1}$ | $93.1_{\pm0.2}$ | $91.2_{\pm0.1}$ | $53.1_{\pm1.3}$ | $21.6_{\pm1.9}$ | 74.3 |
| | 256 | 73.9M | 3.13 | $89.5_{\pm0.1}$ | $96.0_{\pm0.0}$ | $87.1_{\pm0.6}$ | $64.2_{\pm0.8}$ | $93.2_{\pm0.2}$ | $91.3_{\pm0.1}$ | $53.8_{\pm1.7}$ | $21.7_{\pm1.7}$ | 74.6 |
| | 512 | 147.8M | 3.10 | $89.4_{\pm0.1}$ | $95.9_{\pm0.2}$ | $87.9_{\pm0.7}$ | $65.0_{\pm0.9}$ | $93.4_{\pm0.3}$ | $91.4_{\pm0.1}$ | $53.5_{\pm1.8}$ | $21.1_{\pm1.8}$ | 74.7 |
| | 1024 | 295.7M | **3.08** | $89.4_{\pm0.1}$ | $96.1_{\pm0.2}$ | $87.8_{\pm0.6}$ | $65.6_{\pm0.8}$ | $93.3_{\pm0.1}$ | $91.4_{\pm0.1}$ | $54.4_{\pm1.4}$ | $21.8_{\pm1.6}$ | **75.0** |
| **Llama-3.1-8B** | | | | | | | | | | | | |
| LoRA | 16 | 41.9M | 2.32 | $91.8_{\pm0.1}$ | $96.6_{\pm0.1}$ | $87.7_{\pm0.4}$ | $69.5_{\pm1.1}$ | $95.6_{\pm0.1}$ | $92.0_{\pm0.1}$ | $52.8_{\pm1.8}$ | $20.4_{\pm2.3}$ | 75.8 |
| | 32 | 83.9M | 2.31 | $91.8_{\pm0.1}$ | $96.4_{\pm0.1}$ | $88.3_{\pm0.4}$ | $69.6_{\pm0.9}$ | $95.6_{\pm0.1}$ | $92.1_{\pm0.0}$ | $53.0_{\pm1.2}$ | $20.1_{\pm2.4}$ | 75.9 |
| | 64 | 167.8M | 2.30 | $91.8_{\pm0.1}$ | $96.6_{\pm0.1}$ | $88.3_{\pm0.6}$ | $70.0_{\pm0.8}$ | $95.7_{\pm0.1}$ | $92.1_{\pm0.1}$ | $53.2_{\pm1.9}$ | $20.3_{\pm2.6}$ | 76.0 |
| | 128 | 335.5M | 2.30 | $91.8_{\pm0.0}$ | $96.7_{\pm0.2}$ | $88.5_{\pm0.6}$ | $69.9_{\pm1.2}$ | $95.7_{\pm0.0}$ | $92.2_{\pm0.0}$ | $54.1_{\pm3.0}$ | $21.3_{\pm2.2}$ | 76.3 |
| | 256 | 671.1M | **2.29** | $91.8_{\pm0.0}$ | $96.6_{\pm0.1}$ | $88.9_{\pm0.4}$ | $70.3_{\pm1.1}$ | $95.7_{\pm0.1}$ | $92.3_{\pm0.0}$ | $54.2_{\pm1.3}$ | $21.3_{\pm2.1}$ | 76.4 |
| BD-LoRA TP=8 | 32 | 36.2M | 2.32 | $91.8_{\pm0.1}$ | $96.7_{\pm0.1}$ | $87.2_{\pm0.7}$ | $69.0_{\pm1.0}$ | $95.5_{\pm0.1}$ | $91.9_{\pm0.0}$ | $52.5_{\pm1.9}$ | $20.1_{\pm2.2}$ | 75.6 |
| | 64 | 72.4M | 2.31 | $91.8_{\pm0.1}$ | $96.7_{\pm0.1}$ | $87.6_{\pm0.7}$ | $69.8_{\pm0.8}$ | $95.6_{\pm0.1}$ | $91.9_{\pm0.1}$ | $52.9_{\pm1.2}$ | $20.8_{\pm2.1}$ | 75.9 |
| | 128 | 144.7M | 2.30 | $91.8_{\pm0.1}$ | $96.6_{\pm0.1}$ | $87.9_{\pm0.6}$ | $70.4_{\pm0.8}$ | $95.7_{\pm0.1}$ | $92.1_{\pm0.1}$ | $53.8_{\pm1.2}$ | $21.0_{\pm2.1}$ | 76.2 |
| | 256 | 289.4M | 2.30 | $91.9_{\pm0.1}$ | $96.8_{\pm0.1}$ | $87.7_{\pm0.7}$ | $70.8_{\pm0.8}$ | $95.6_{\pm0.1}$ | $92.2_{\pm0.1}$ | $55.0_{\pm1.1}$ | $22.4_{\pm2.2}$ | 76.6 |
| | 512 | 578.8M | **2.29** | $91.9_{\pm0.1}$ | $96.9_{\pm0.1}$ | $88.7_{\pm0.7}$ | $69.9_{\pm1.0}$ | $95.7_{\pm0.1}$ | $92.2_{\pm0.1}$ | $55.0_{\pm1.2}$ | $22.5_{\pm2.6}$ | **76.6** |

# F   Additional Run-time Experiments

## F.1   Main Results

In this section, we present the throughput, end-to-end (E2E) latency, decoding latency, and prefill latency of Llama-3.1-8B and Llama-3.1-70B with batch sizes (BS) of 1 / 16 / 32 / 64. Figure 10 and 11 show the results for an input token length of 1024 and an output token length of 128, while Figure 12 and 13 present the results for an input token length of 4096 and an output token length of 256.

BD-LoRA demonstrates the best E2E latency, decoding latency, and throughput compared to NFS-LoRA and S-LoRA across all settings and for both Llama-3.1-8B and Llama-3.1-70B. BD-LoRA outperforms NFS-LoRA particularly on larger model sizes, while surpassing S-LoRA especially on smaller model sizes, demonstrating stable and superior performance overall. As the ranks increase, both NFS-LoRA and S-LoRA experience significant increases in latency and decreases in throughput. As the batch size increases, NFS-LoRA exhibits higher latency and lower throughput, particularly at larger batch sizes on the 70B model. S-LoRA, on the other hand, shows higher latency and lower throughput even with BS = 1, primarily due to the additional communication it introduces. From the results shown in Figure 10, 11, 12 and 13, the input token length and output token length do not significantly affect the relative runtime performance differences among these three methods. However, batch size has a substantial impact on NFS-LoRA. BD-LoRA consistently demonstrates stable and superior results across different settings.

We also present the results in Figure 14 and 15 for an input token length of 128 and an output token length of 1024, focusing on generation-heavy tasks. The results and trends are similar between Figure 10, 11 and Figure 14, 15. BD-LoRA shows even better results in generation-heavy tasks, as it performs better in decoding latency. These results highlight the stability and superior runtime performance of BD-LoRA in generation-intensive tasks, demonstrating its effectiveness in both short and long generation scenarios.

**Speedup with respect to fine-tuning performance**    To quantify BD-LoRA's speedup with respect to fine-tuning performance, we adopt the following procedure. For each rank of BD-LoRA, we identify a rank of NFS-LoRA and S-LoRA that produces the most comparable but slightly worse results compared to the corresponding rank of BD-LoRA. We then compare their speedup. This analysis is conducted for both OpenOrca [16] and GLUE [35], and the results are presented in Table 3 to 8. For example, in Table 3, for BS=1 and rank=128 of BD-LoRA, we found that rank=64 of S-LoRA produces worse but most similar results to rank=128 of BD-LoRA. We then compute the speedup based on the performance of rank=128 for BD-LoRA and rank=64 for S-LoRA.

This approach ensures that, when computing speedup between BD-LoRA and S-LoRA, BD-LoRA always achieves better results than S-LoRA, making the comparison fair. If a rank of BD-LoRA does not outperform any rank of S-LoRA, the speedup is marked as "N/A."

We observe that BD-LoRA achieves up to **2.68x** speedup in E2E latency compared to S-LoRA, which means that, even with slightly better downstream performance, BD-LoRA achieves up to a 2.68x speedup. With a similar number of trainable parameters, BD-LoRA demonstrates slightly better downstream results, as shown in Appendix E.1. This makes the speedup more significant, as BD-LoRA performs better in both downstream performance and efficiency.

**Speedup with respect to the number of trainable parameters**    To compare speedup with respect to the number of trainable parameters, for each rank of BD-LoRA, we identify two ranks of S-LoRA for comparison: one with fewer trainable parameters but most similar to BD-LoRA, and one with more trainable parameters but closest to BD-LoRA. In this way, there are two speedup values for each rank of BD-LoRA. If no corresponding rank can be found in S-LoRA, the speedup is marked as "N/A."

Tables 9 to 20 present the detailed results for throughput, end-to-end (E2E) latency, decoding latency, and prefill latency of Llama-3.1-8B and Llama-3.1-70B for the generation task with input token lengths (IT) of 1024, 4096, and 128, and output token lengths (OT) of 128, 256, and 1024, respectively. Tables 21 and 22 present the detailed results for Llama-3.1-8B with TP=4.

For Llama-3.1-8B, BD-LoRA achieves up to a **1.63x** speedup with 0.86x the number of trainable parameters. BD-LoRA achieves up to a **1.30x** speedup with 1.73x the number of trainable parameters.

For Llama-3.1-70B, BD-LoRA achieves up to a **1.79x** speedup with 0.87x the number of trainable parameters. BD-LoRA achieves up to a **1.23x** speedup with 1.74x the number of trainable parameters. When TP=4, BD-LoRA achieves up to a **1.27x** speedup with 1.03x the number of trainable parameters.

## F.2 Runtime Analysis with Varying TP Configurations

We present experiments with TP degree TP = 2/4/8 and batch size BS = 1/16/32 in Figure 16, 17, and 18, respectively, using an input length of 1024 and an output length of 128 with Llama-3.1-8B.

We see that for smaller TP degrees the performance differences between NFS-LoRA, S-LoRA, and BD-LoRA are less significant. This is expected since for smaller TP degrees NFS-LoRA copies LoRA adapters to fewer devices and S-LoRA communicates between fewer devices. Still, BD-LoRA consistently outperforms NFS-LoRA and S-LoRA. Note that for all methods performance goes down as the TP degree gets smaller, making TP = 8 the fastest option for serving Llama-3.1-8B on one AWS p4d instance. This trend indicates that BD-LoRA's performance advantage is likely to widen as the TP size grows, suggesting strong scalability to larger configurations (e.g., TP-16 and TP-32).

## F.3 Attention-only and MLP-only Runtime

Figure 19, 20, 21, and 22 show the throughput, E2E latency, decoding latency, and prefill latency of Llama-3.1-8B and Llama-3.1-70B using Attention-only and MLP-only adapters.

For the Attention-only adapter, compared to Figure 10 and 11, the absolute latency of all three methods decreases slightly. Removing LoRA from the MLP module reduces latency and computation for all methods. However, BD-LoRA demonstrates relatively greater advantages compared to the other baselines in the Attention module, with lower latency and higher throughput. Additionally, the Attention module performs computations within each device, which increases intermediate computation before each communication step. This means devices are more likely to wait for communication to complete. The Attention module includes Q, K, V, and O projections, resulting in four projection layers and four additional LoRA communication steps, which are more than the MLP module. BD-LoRA eliminates communication overhead when sharding LoRA adapters, resulting in greater efficiency in the Attention module.

For the MLP-only adapters, BD-LoRA has fewer trainable parameters compared to the Attention adapters. The intermediate size in the MLP module is typically very large, and using block-diagonal matrices in $B_1$ and $A_2$ significantly reduces the number of trainable parameters compared to the Attention module. The speedup for MLP-only adapters is also more limited compared to Attention adapters, as the adapters involve four communication steps in the Attention module, whereas the MLP module has only two communication steps for standard transformers [34] and three for GLU variants [27]. Nonetheless, BD-LoRA still demonstrates lower latency compared to other baselines.

## F.4 Multi-LoRA Runtime

The adapters are loaded from the disk into GPUs via the following pipeline in vLLM: (1) LoRA weights are first loaded into CPU memory and cached using the CPU's LRU cache. (2) The LoRA weights are then sliced within the CPU, transferred to the GPU, and cached in GPU memory.

The loading process from the CPU to the GPU is fast and has minimal impact on prefill latency. However, loading from the disk to the CPU memory is time-consuming. Our implementation, based on vLLM, excludes zeros for BD-LoRA in both loading and computation. In this section, we analyze the runtime of BD-LoRA when each request uses a different adapter, preventing all adapters from being cached in memory and requiring them to be loaded from disk. This evaluation aims to assess the performance of BD-LoRA under such conditions and verify whether our implementation, which excludes zeros, is efficient during the loading process.

Figure 23 shows the results of Llama-3.1-8B and Llama-3.1-70B using multi-adapters loaded from the disk with an input token length (IT) of 1024, an output token length (OT) of 128, and a batch size (BS) of 1. Compared with Figure 10 and 11, all methods with multi-LoRA exhibit a similar increase in prefill latency. This increase also influences the E2E latency and throughput, as these two metrics depend on the prefill latency. However, the decoding latency is largely unaffected, as it

does not include loading time, and BD-LoRA still demonstrates superior performance. The loading time is proportional to the number of trainable parameters. The prefill latency of the three methods aligns with the number of trainable parameters, demonstrating that our implementation successfully excludes zeros during loading.

Table 23, 24, and 25 show the speedup of Llama-3.1-8B and Llama-3.1-70B using multi-adapters loaded from the disk with an input token length (IT) of 1024, an output token length (OT) of 128, and a batch size (BS) of 1. With respect to fine-tuning performance, BD-LoRA achieves up to **1.78x** speedup in E2E latency compared to S-LoRA. With respect to the number of trainable parameters, for Llama-3.1-8B, BD-LoRA achieves up to a **1.31x** speedup with 0.86x the number of trainable parameters and up to a **1.20x** speedup with 1.73x the number of trainable parameters. For Llama-3.1-70B, BD-LoRA achieves up to a **1.23x** speedup with 0.87x the number of trainable parameters and up to a **1.13x** speedup with 1.74x the number of trainable parameters. As both BD-LoRA and S-LoRA experience a constant increase in prefill latency, the speedup is reduced. However, BD-LoRA still demonstrates relatively higher efficiency compared to S-LoRA.

Nevertheless, caching adapters in CPU or GPU memory does not significantly impact latency. The use of loading adapters from disk is typically considered only when the number of adapters is large and exceeds the maximum memory capacity.

### F.5   Runtime Analysis with Varying Devices

We present experiments with Llama-3.1-8B on AWS g5 instance equipped with eight NVIDIA A10G GPUs (each with 24 GB HBM2) and PCIe interconnect in Figure 24-25 and Table 26-27.

With respect to the number of trainable parameters, for Llama-3.1-8B, BD-LoRA achieves up to a **1.36x** speedup with 0.86x the number of trainable parameters and up to a **1.27x** speedup with 1.73x the number of trainable parameters. These results demonstrate that BD-LoRA is effective across different devices.

### F.6   Runtime Analysis for Quantized Base Model

Table 28 provides some performance results for a quantized Llama-3.1-8B model (weight-only quantization to INT4 data type) as base model.

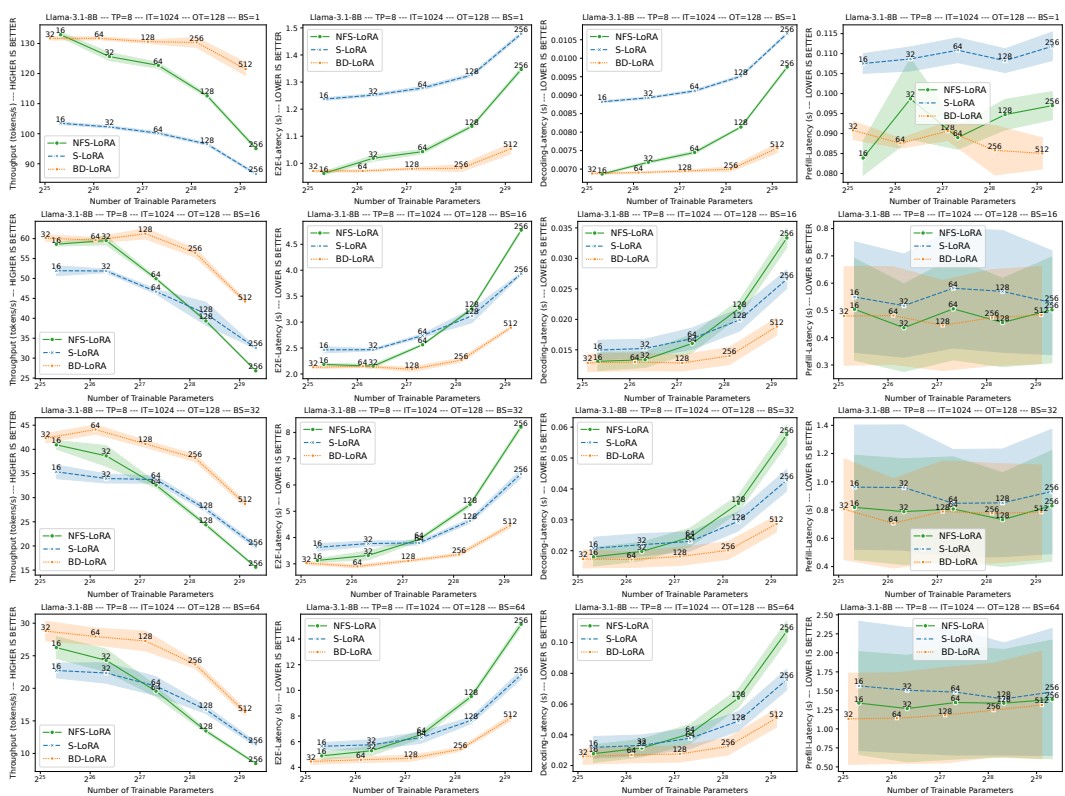

Figure 10: Throughput (1st column—higher is better), end-to-end (E2E) latency (2nd column—lower is better), decoding latency (3rd column—lower is better), and prefill latency (4th column—lower is better) of **Llama-3.1-8B** with an input token (**IT**) length of **1024**, an output token (**OT**) length of **128**, and batch sizes (**BS**) of **1, 16, 32 and 64** (1st to 4th row).

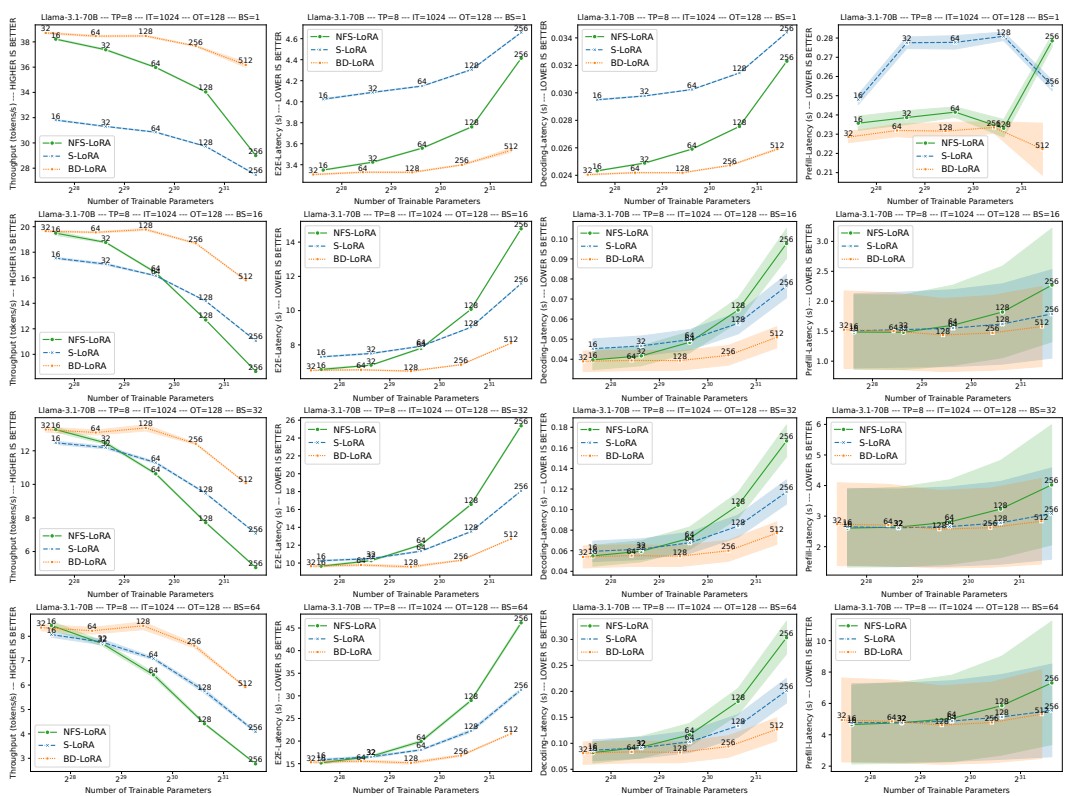

Figure 11: Throughput (1st column—higher is better), end-to-end (E2E) latency (2nd column—lower is better), decoding latency (3rd column—lower is better), and prefill latency (4th column—lower is better) of **Llama-3.1-70B** with an Input Token (**IT**) length of **1024**, an output token (**OT**) length of **128**, and batch sizes (**BS**) of **1, 16, 32 and 64** (1st to 4th row).

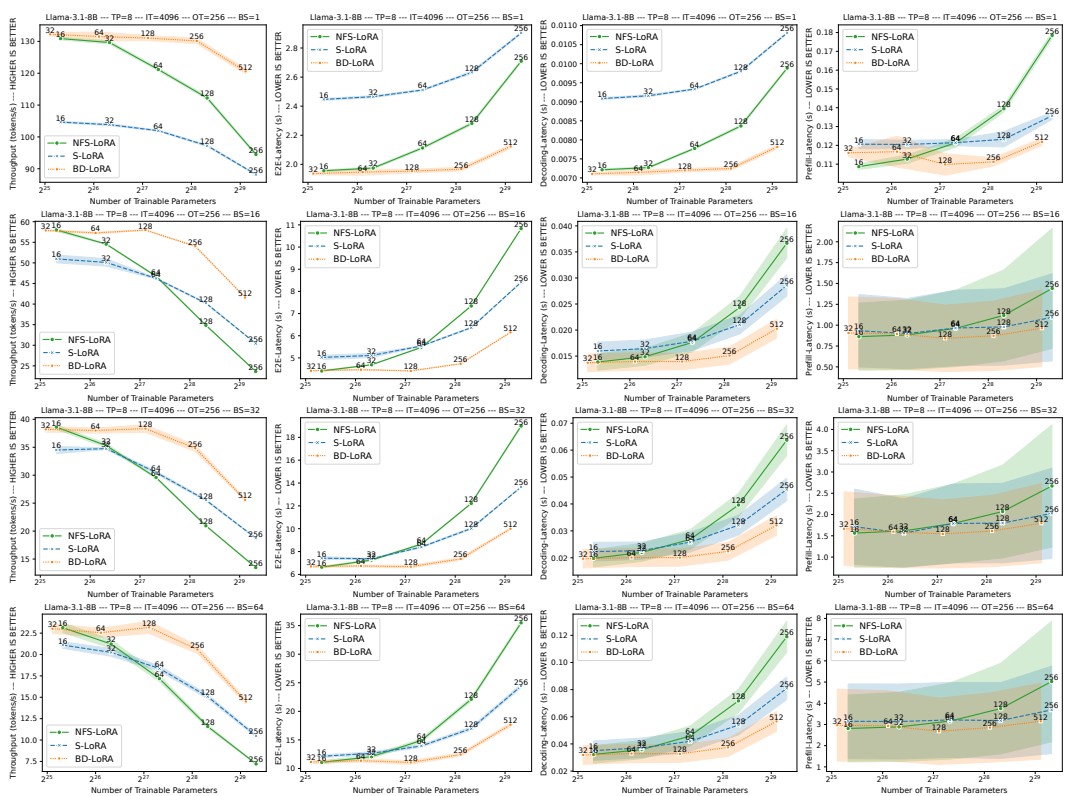

Figure 12: Throughput (1st column—higher is better), end-to-end (E2E) latency (2nd column—lower is better), decoding latency (3rd column—lower is better), and prefill latency (4th column—lower is better) of **Llama-3.1-8B** with an Input Token (**IT**) length of **4096**, an output token (**OT**) length of **256**, and batch sizes (**BS**) of **1, 16, 32 and 64** (1st to 4th row).

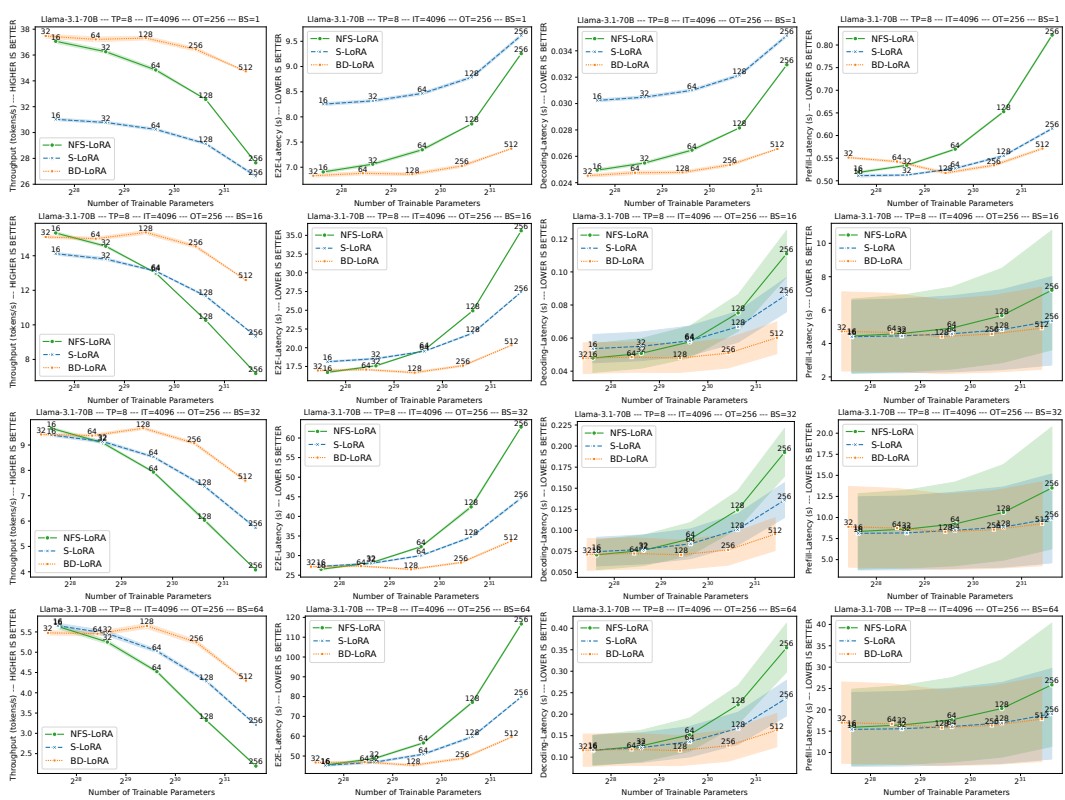

Figure 13: Throughput (1st column—higher is better), end-to-end (E2E) latency (2nd column—lower is better), decoding latency (3rd column—lower is better), and prefill latency (4th column—lower is better) of **Llama-3.1-70B** with an input Token (**IT**) length of **4096**, an output token (**OT**) length of **256**, and batch sizes (**BS**) of **1, 16, 32 and 64** (1st to 4th row).

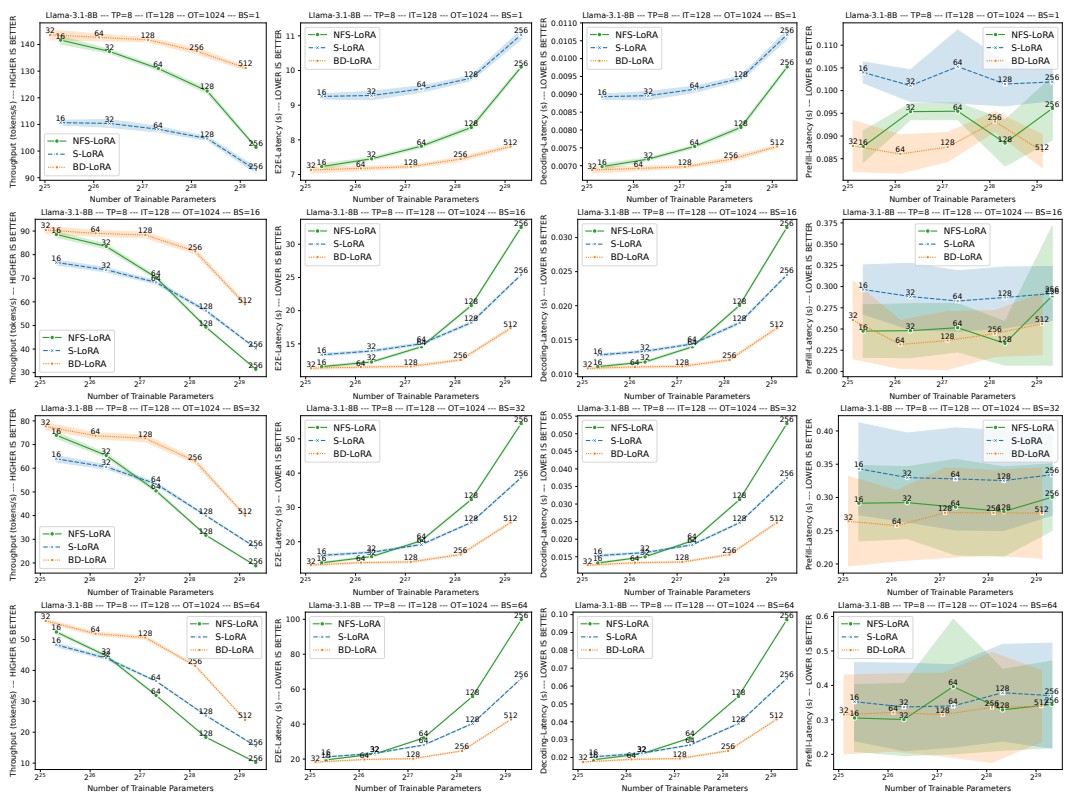

Figure 14: Throughput (1st column—higher is better), end-to-end (E2E) latency (2nd column—lower is better), decoding latency (3rd column—lower is better), and prefill latency (4th column—lower is better) of **Llama-3.1-8B** with an input token (**IT**) length of **128**, an output token (**OT**) length of **1024**, and batch sizes (**BS**) of **1, 16, 32 and 64** (1st to 4th row).

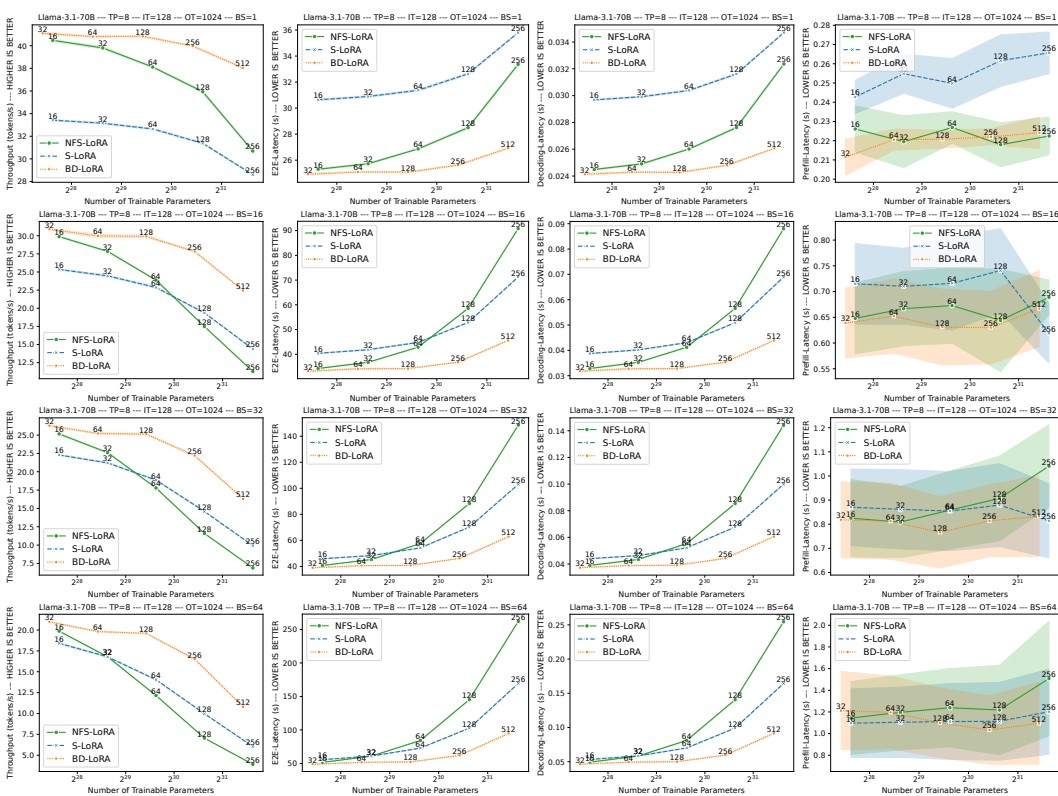

Figure 15: Throughput (1st column—higher is better), end-to-end (E2E) latency (2nd column—lower is better), decoding latency (3rd column—lower is better), and prefill latency (4th column—lower is better) of **Llama-3.1-70B** with an input token (**IT**) length of **128**, an output token (**OT**) length of **1024**, and batch sizes (**BS**) of **1, 16, 32 and 64** (1st to 4th row).

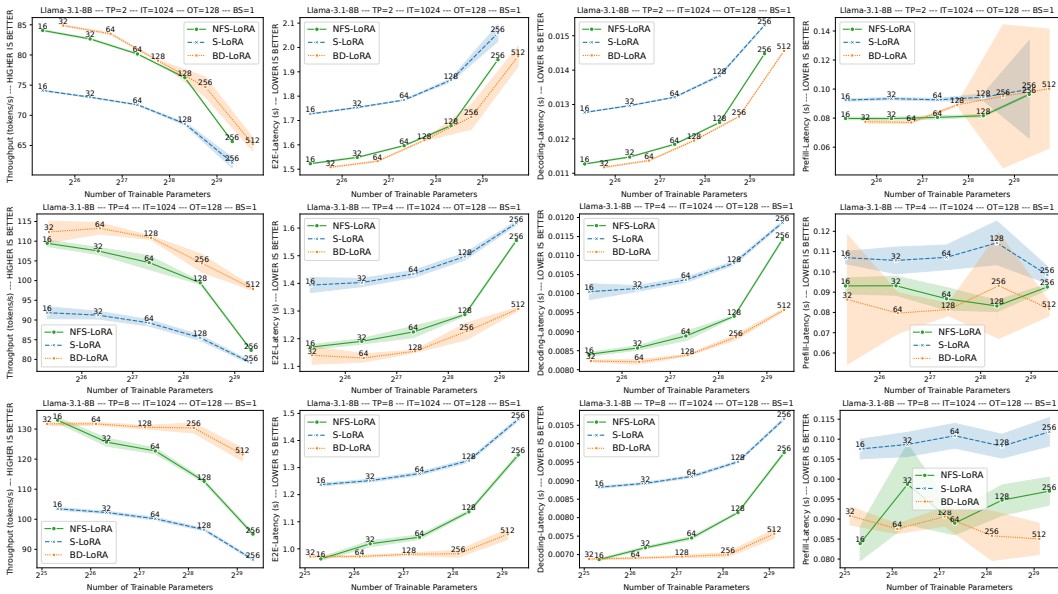

Figure 16: Throughput (1st column—higher is better), end-to-end (E2E) latency (2nd column—lower is better), decoding latency (3rd column—lower is better), and prefill latency (4th column—lower is better) of **Llama-3.1-8B** with **TP** degrees of **2, 4, and 8** (1st to 3rd row), an input token (**IT**) length of **1024**, an output token (**OT**) length of **128**, and Batch Size (**BS**) of **1**.

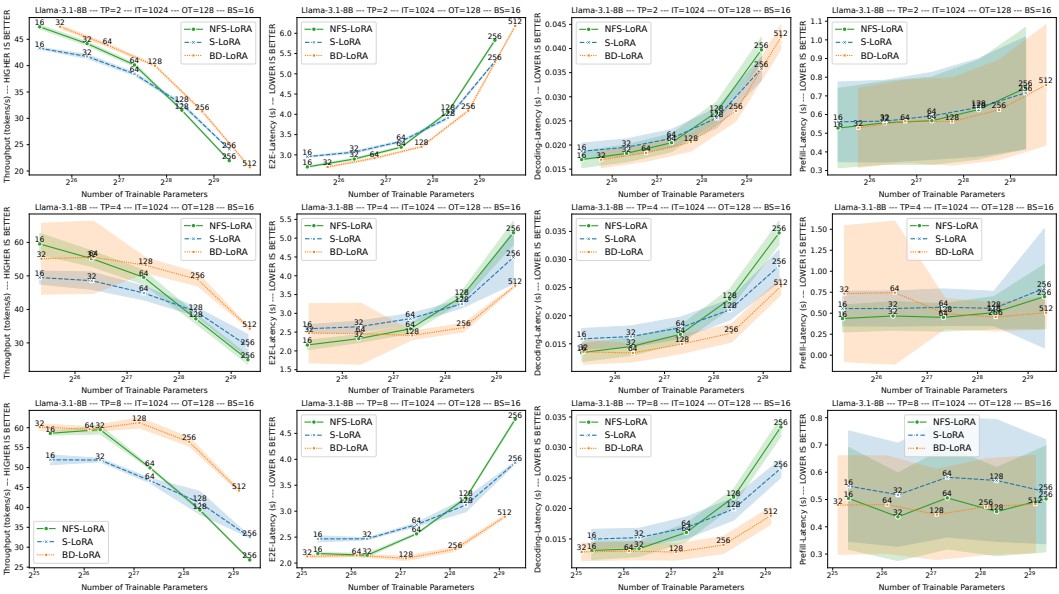

Figure 17: Throughput (1st column—higher is better), end-to-end (E2E) latency (2nd column—lower is better), decoding latency (3rd column—lower is better), and prefill latency (4th column—lower is better) of **Llama-3.1-8B** with **TP** degrees of **2, 4, and 8** (1st to 3rd row), an input token (**IT**) length of **1024**, an output token (**OT**) length of **128**, and Batch Size (**BS**) of **16**.

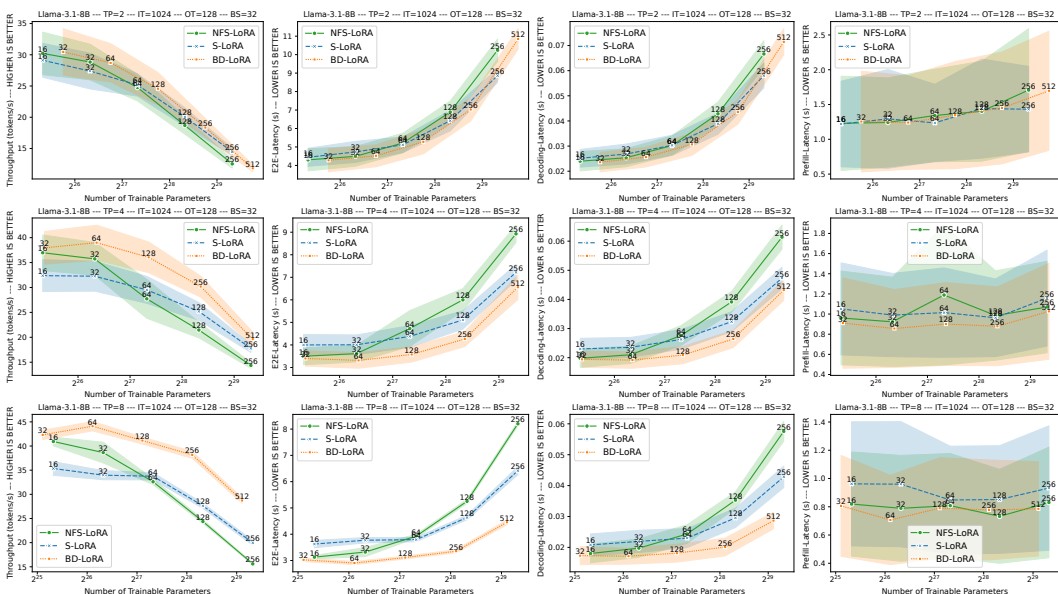

Figure 18: Throughput (1st column—higher is better), end-to-end (E2E) latency (2nd column—lower is better), decoding latency (3rd column—lower is better), and prefill latency (4th column—lower is better) of **Llama-3.1-8B** with **TP** degrees of **2, 4, and 8** (1st to 3rd row), an input token (**IT**) length of **1024**, an output token (**OT**) length of **128**, and Batch Size (**BS**) of **32**.

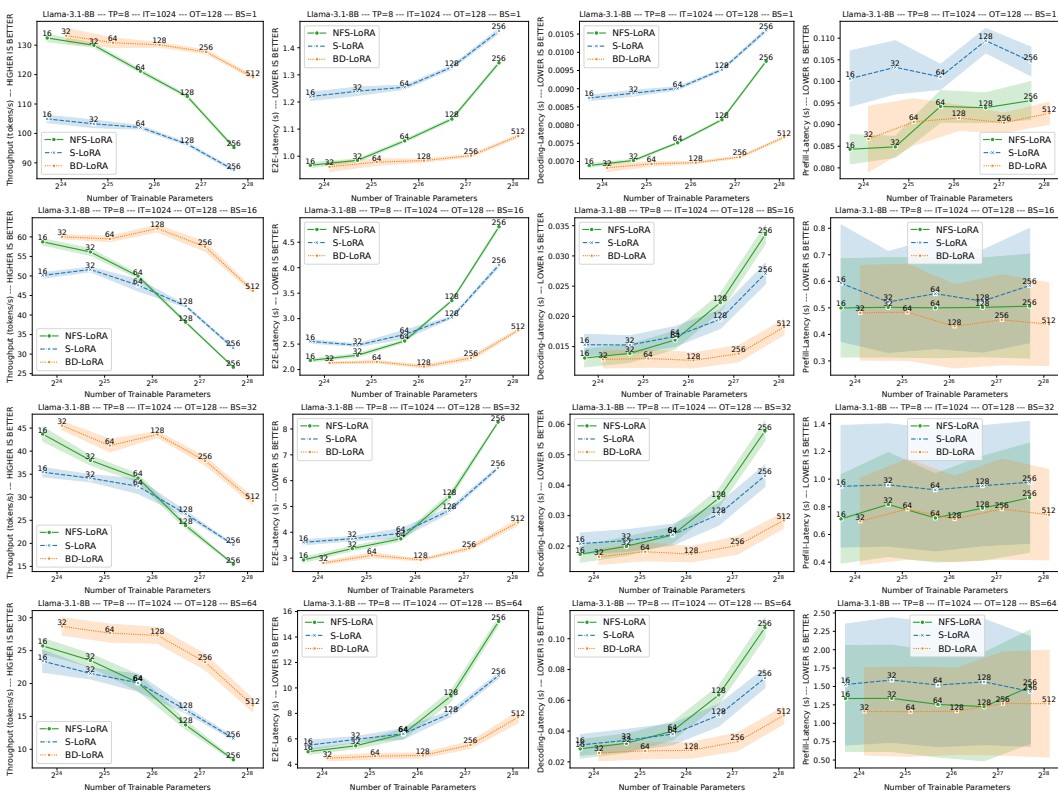

Figure 19: Throughput (1st column—higher is better), end-to-end (E2E) latency (2nd column—lower is better), decoding latency (3rd column—lower is better), and prefill latency (4th column—lower is better) of **Llama-3.1-8B** using **Attention-only adapters**. The experiments are conducted with an input token (**IT**) length of **1024**, an output token (**OT**) length of **128**, and batch sizes (**BS**) of **1, 16, 32, and 64**.

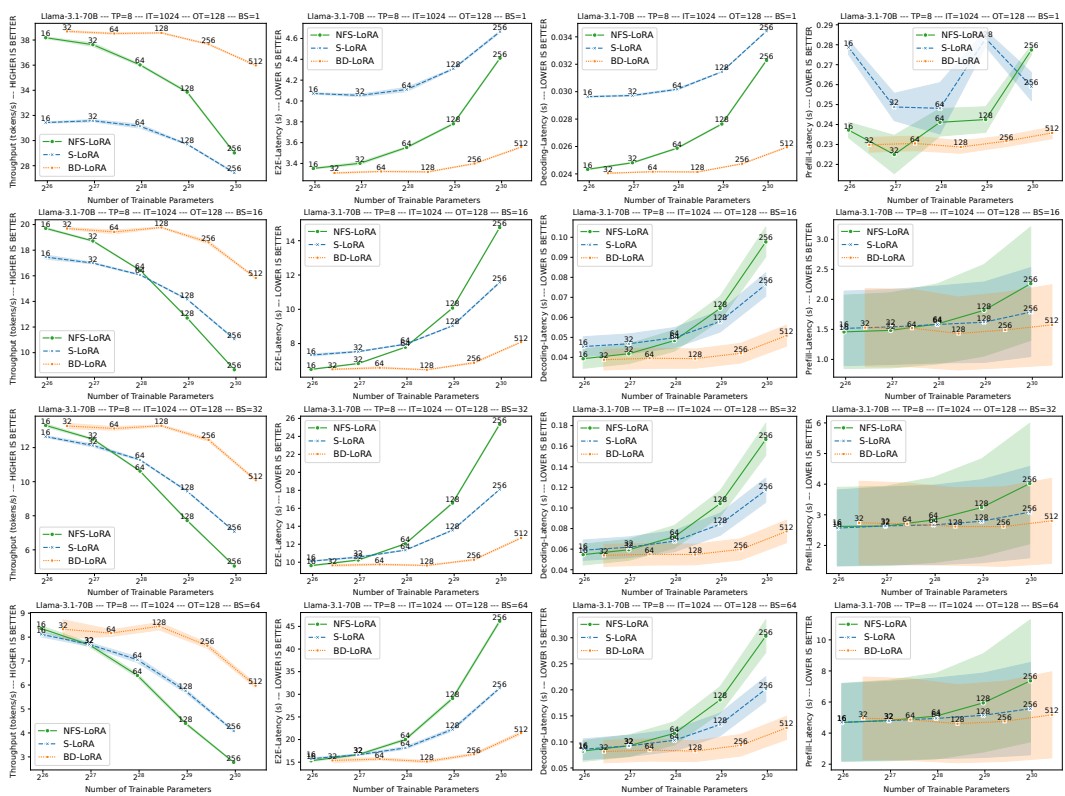

Figure 20: Throughput (1st column—higher is better), end-to-end (E2E) latency (2nd column—lower is better), decoding latency (3rd column—lower is better), and prefill latency (4th column—lower is better) of **Llama-3.1-70B** using **Attention-only adapters**. The experiments are conducted with an input token (**IT**) length of **1024**, an output token (**OT**) length of **128**, and batch sizes (**BS**) of **1, 16, 32, and 64**.

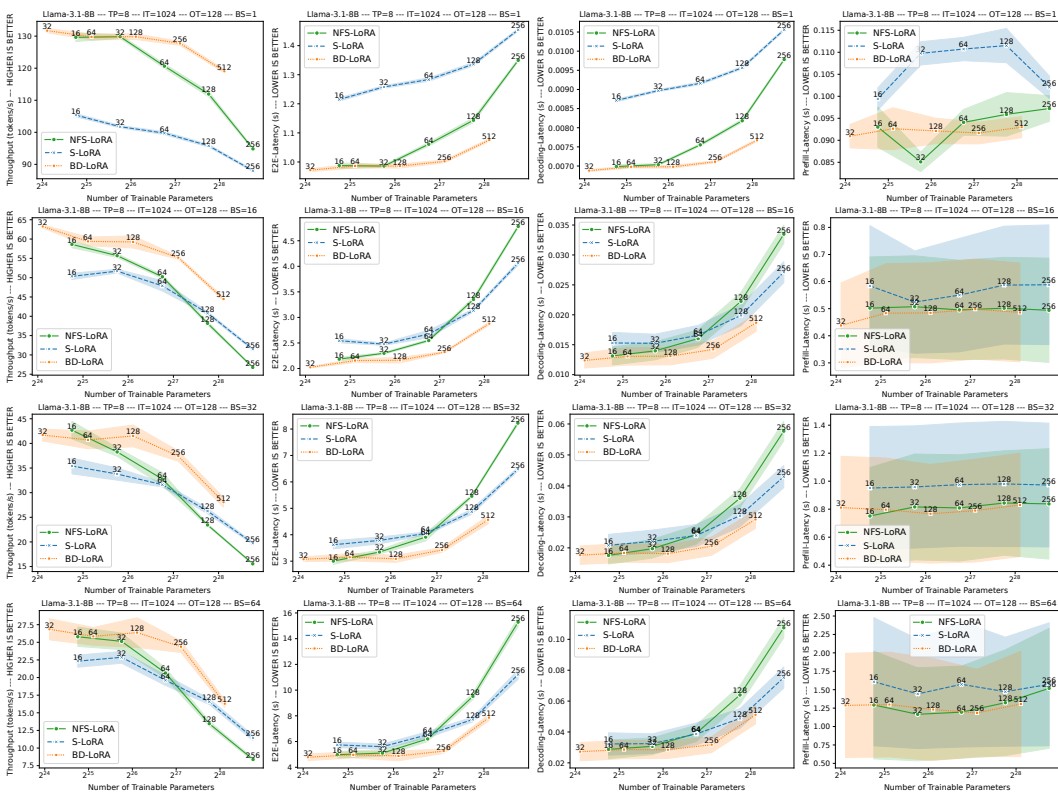

Figure 21: Throughput (1st column—higher is better), end-to-end (E2E) latency (2nd column—lower is better), decoding latency (3rd column—lower is better), and prefill latency (4th column—lower is better) of **Llama-3.1-8B** using **MLP-only adapters**. The experiments are conducted with an input token (**IT**) length of **1024**, an output token (**OT**) length of **128**, and batch sizes (**BS**) of **1, 16, 32, and 64**.

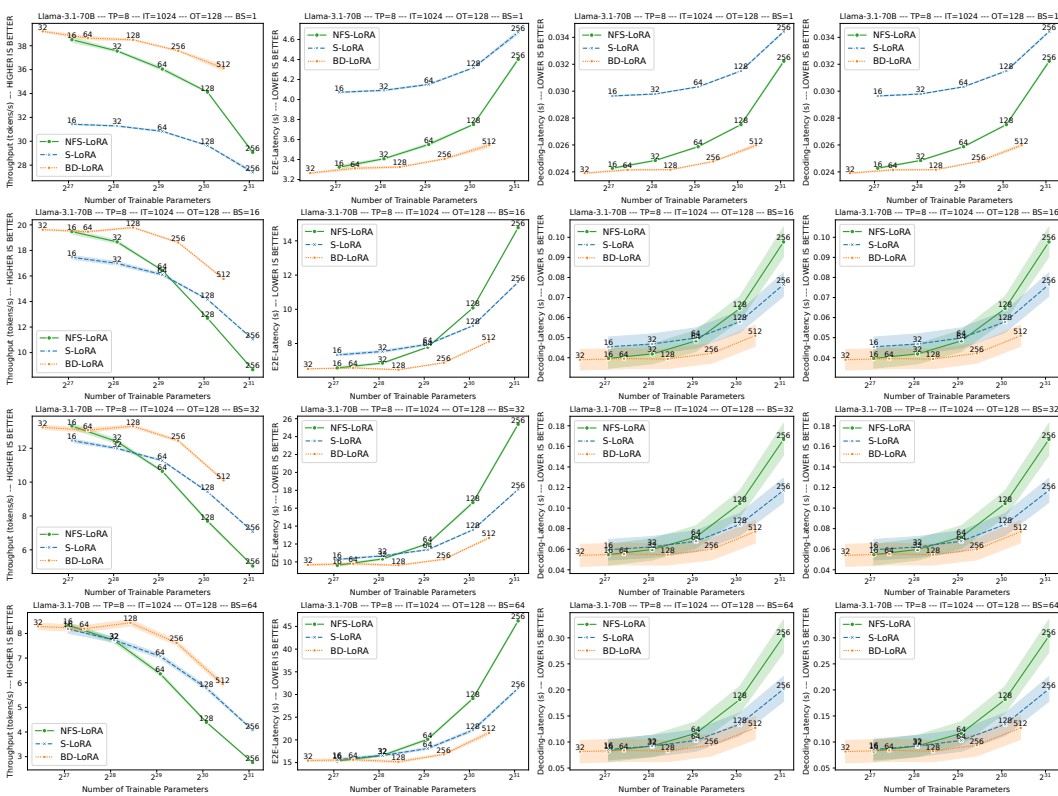

Figure 22: Throughput (1st column—higher is better), end-to-end (E2E) latency (2nd column—lower is better), decoding latency (3rd column—lower is better), and prefill latency (4th column—lower is better) of **Llama-3.1-70B** using **MLP-only adapters**. The experiments are conducted with an input token (**IT**) length of **1024**, an output token (**OT**) length of **128**, and batch sizes (**BS**) of **1, 16, 32, and 64**.

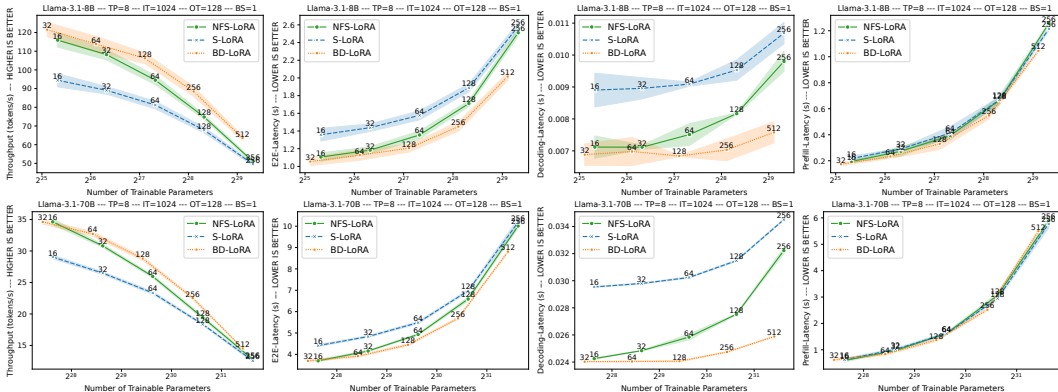

Figure 23: Throughput (1st column—higher is better), end-to-end (E2E) latency (2nd column—lower is better), decoding latency (3rd column—lower is better), and prefill latency (4th column—lower is better) of **Llama-3.1-8B** and **Llama-3.1-70B** using **multi-adapters** loaded from the **disk**. The evaluation is performed with an input token (**IT**) length of **1024** and an output token (**OT**) length of **128**, and batch sizes (**BS**) of **1**.

Table 3: Speedup of throughput, end-to-end (E2E) latency, decoding latency, and prefill latency of **Llama-3.1-8B** with an input token (**IT**) length of **1024**, an output token (**OT**) length of **128**, and batch sizes (**BS**) of **1, 16, 32, and 64**. **S.-OO.** and **S.-G.** denote the speedup with respect to OpenOrca and GLUE, respectively.

| Method | Rank | # Trainable Parameters | OpenOrca ↓ | GLUE ↑ | Throughput ↑ | | | E2E Latency ↓ | | | Decoding Latency ↓ | | | Prefill Latency ↓ | | |
|---|---|---|---|---|---|---|---|---|---|---|---|---|---|---|---|---|
| | | | | | Token/s | S.-OO. | S.-G. | Time | S.-OO. | S.-G. | Time | S.-OO. | S.-G. | Time | S.-OO. | S.-G. |
| | | | | | | | | **BS=1** | | | | | | | | |
| | 16 | 41.9M | 2.316 | 75.82 | 132.9 | 1.00x | 1.00x | 0.96 | 1.00x | 1.00x | 0.0069 | 1.00x | 1.00x | 0.084 | 1.00x | 1.00x |
| | 32 | 83.9M | 2.309 | 75.87 | 125.7 | 1.00x | 1.00x | 1.02 | 1.00x | 1.00x | 0.0072 | 1.00x | 1.00x | 0.099 | 1.00x | 1.00x |
| **NFS-LoRA** | 64 | 167.8M | 2.303 | 76.01 | 122.7 | 1.00x | 1.00x | 1.04 | 1.00x | 1.00x | 0.0075 | 1.00x | 1.00x | 0.089 | 1.00x | 1.00x |
| | 128 | 335.5M | 2.297 | 76.28 | 112.6 | 1.00x | 1.00x | 1.14 | 1.00x | 1.00x | 0.0081 | 1.00x | 1.00x | 0.095 | 1.00x | 1.00x |
| | 256 | 671.1M | 2.290 | 76.39 | 95.1 | 1.00x | 1.00x | 1.35 | 1.00x | 1.00x | 0.0098 | 1.00x | 1.00x | 0.097 | 1.00x | 1.00x |
| | 32 | 36.2M | 2.318 | 75.58 | 131.7 | N/A | N/A | 0.97 | N/A | N/A | 0.0069 | N/A | N/A | 0.091 | N/A | N/A |
| | 64 | 72.4M | 2.310 | 75.90 | 131.7 | 0.99x | 1.05x | 0.97 | 0.99x | 1.05x | 0.0069 | 0.99x | 1.04x | 0.088 | 0.96x | 1.13x |
| **BD-LoRA** | 128 | 144.7M | 2.303 | 76.17 | 130.6 | 1.04x | 1.06x | 0.98 | 1.04x | 1.06x | 0.0069 | 1.03x | 1.07x | 0.091 | 1.09x | 0.98x |
| | 256 | 289.4M | 2.296 | 76.55 | 130.4 | 1.16x | 1.37x | 0.98 | 1.16x | 1.37x | 0.0070 | 1.16x | 1.39x | 0.086 | 1.10x | 1.13x |
| | 512 | 578.8M | **2.289** | **76.59** | 121.5 | 1.28x | 1.28x | 1.05 | 1.28x | 1.28x | 0.0076 | 1.29x | 1.29x | 0.085 | 1.14x | **1.14x** |
| | | | | | | | | **BS=16** | | | | | | | | |
| | 16 | 41.9M | 2.316 | 75.82 | 58.6 | 1.00x | 1.00x | 2.18 | 1.00x | 1.00x | 0.0131 | 1.00x | 1.00x | 0.504 | 1.00x | 1.00x |
| | 32 | 83.9M | 2.309 | 75.87 | 59.5 | 1.00x | 1.00x | 2.15 | 1.00x | 1.00x | 0.0134 | 1.00x | 1.00x | 0.437 | 1.00x | 1.00x |
| **NFS-LoRA** | 64 | 167.8M | 2.303 | 76.01 | 49.9 | 1.00x | 1.00x | 2.56 | 1.00x | 1.00x | 0.0161 | 1.00x | 1.00x | 0.506 | 1.00x | 1.00x |
| | 128 | 335.5M | 2.297 | 76.28 | 39.3 | 1.00x | 1.00x | 3.25 | 1.00x | 1.00x | 0.0218 | 1.00x | 1.00x | 0.456 | 1.00x | 1.00x |
| | 256 | 671.1M | 2.290 | 76.39 | 26.8 | 1.00x | 1.00x | 4.77 | 1.00x | 1.00x | 0.0334 | 1.00x | 1.00x | 0.502 | 1.00x | 1.00x |
| | 32 | 36.2M | 2.318 | 75.58 | 60.1 | N/A | N/A | 2.13 | N/A | N/A | 0.0129 | N/A | N/A | 0.480 | N/A | N/A |
| | 64 | 72.4M | 2.310 | 75.90 | 59.6 | 1.02x | 1.00x | 2.15 | 1.02x | 1.00x | 0.0130 | 1.01x | 1.03x | 0.481 | 1.05x | 0.91x |
| **BD-LoRA** | 128 | 144.7M | 2.303 | 76.17 | 61.2 | 1.03x | 1.23x | 2.09 | 1.03x | 1.23x | 0.0128 | 1.04x | 1.25x | 0.445 | 0.98x | **1.14x** |
| | 256 | 289.4M | 2.296 | 76.55 | 56.4 | 1.44x | 2.10x | 2.27 | 1.43x | 2.10x | 0.0140 | 1.56x | 2.38x | 0.474 | 0.96x | 1.06x |
| | 512 | 578.8M | **2.289** | **76.59** | 44.2 | 1.65x | 1.65x | 2.89 | 1.65x | 1.65x | 0.0188 | 1.77x | 1.77x | 0.483 | 1.04x | 1.04x |
| | | | | | | | | **BS=32** | | | | | | | | |
| | 16 | 41.9M | 2.316 | 75.82 | 40.9 | 1.00x | 1.00x | 3.13 | 1.00x | 1.00x | 0.0180 | 1.00x | 1.00x | 0.819 | 1.00x | 1.00x |
| | 32 | 83.9M | 2.309 | 75.87 | 38.6 | 1.00x | 1.00x | 3.32 | 1.00x | 1.00x | 0.0198 | 1.00x | 1.00x | 0.789 | 1.00x | 1.00x |
| **NFS-LoRA** | 64 | 167.8M | 2.303 | 76.01 | 32.6 | 1.00x | 1.00x | 3.93 | 1.00x | 1.00x | 0.0243 | 1.00x | 1.00x | 0.808 | 1.00x | 1.00x |
| | 128 | 335.5M | 2.297 | 76.28 | 24.4 | 1.00x | 1.00x | 5.25 | 1.00x | 1.00x | 0.0353 | 1.00x | 1.00x | 0.732 | 1.00x | 1.00x |
| | 256 | 671.1M | 2.290 | 76.39 | 15.6 | 1.00x | 1.00x | 8.20 | 1.00x | 1.00x | 0.0576 | 1.00x | 1.00x | 0.830 | 1.00x | 1.00x |
| | 32 | 36.2M | 2.318 | 75.58 | 42.3 | N/A | N/A | 3.03 | N/A | N/A | 0.0173 | N/A | N/A | 0.806 | N/A | N/A |
| | 64 | 72.4M | 2.310 | 75.90 | 44.1 | 1.08x | 1.14x | 2.90 | 1.08x | 1.14x | 0.0171 | 1.05x | 1.15x | 0.706 | 1.16x | 1.12x |
| **BD-LoRA** | 128 | 144.7M | 2.303 | 76.17 | 41.1 | 1.06x | 1.26x | 3.11 | 1.07x | 1.26x | 0.0181 | 1.09x | 1.34x | 0.790 | 1.00x | 1.02x |
| | 256 | 289.4M | 2.296 | 76.55 | 38.2 | 1.57x | 2.45x | 3.35 | 1.57x | 2.45x | 0.0201 | 1.76x | 2.86x | 0.778 | 0.94x | 1.07x |
| | 512 | 578.8M | **2.289** | **76.59** | 28.7 | 1.84x | 1.84x | 4.46 | 1.84x | 1.84x | 0.0287 | 2.01x | 2.01x | 0.785 | 1.06x | 1.06x |
| | | | | | | | | **BS=64** | | | | | | | | |
| | 16 | 41.9M | 2.316 | 75.82 | 26.3 | 1.00x | 1.00x | 4.90 | 1.00x | 1.00x | 0.0277 | 1.00x | 1.00x | 1.339 | 1.00x | 1.00x |
| | 32 | 83.9M | 2.309 | 75.87 | 24.3 | 1.00x | 1.00x | 5.29 | 1.00x | 1.00x | 0.0314 | 1.00x | 1.00x | 1.267 | 1.00x | 1.00x |
| **NFS-LoRA** | 64 | 167.8M | 2.303 | 76.01 | 19.6 | 1.00x | 1.00x | 6.56 | 1.00x | 1.00x | 0.0407 | 1.00x | 1.00x | 1.346 | 1.00x | 1.00x |
| | 128 | 335.5M | 2.297 | 76.28 | 13.5 | 1.00x | 1.00x | 9.52 | 1.00x | 1.00x | 0.0639 | 1.00x | 1.00x | 1.338 | 1.00x | 1.00x |
| | 256 | 671.1M | 2.290 | 76.39 | 8.5 | 1.00x | 1.00x | 15.15 | 1.00x | 1.00x | 0.1074 | 1.00x | 1.00x | 1.388 | 1.00x | 1.00x |
| | 32 | 36.2M | 2.318 | 75.58 | 28.8 | N/A | N/A | 4.46 | N/A | N/A | 0.0259 | N/A | N/A | 1.132 | N/A | N/A |
| | 64 | 72.4M | 2.310 | 75.90 | 27.9 | 1.06x | 1.15x | 4.59 | 1.07x | 1.15x | 0.0275 | 1.03x | 1.17x | 1.141 | **1.17x** | 1.11x |
| **BD-LoRA** | 128 | 144.7M | 2.303 | 76.17 | 27.3 | 1.12x | 1.39x | 4.71 | 1.12x | 1.39x | 0.0275 | 1.14x | 1.48x | 1.180 | 1.07x | **1.14x** |
| | 256 | 289.4M | 2.296 | 76.55 | 23.7 | 1.76x | **2.80x** | 5.41 | 1.76x | **2.80x** | 0.0325 | 1.97x | **3.31x** | 1.247 | 1.07x | 1.11x |
| | 512 | 578.8M | **2.289** | **76.59** | 16.3 | **1.93x** | 1.93x | 7.84 | **1.93x** | 1.93x | 0.0509 | **2.11x** | 2.11x | 1.314 | 1.06x | ↓1.06x |

Table 4: Speedup of throughput, end-to-end (E2E) latency, decoding latency, and prefill latency of **Llama-3.1-8B** with an input token (**IT**) length of **1024**, an output token (**OT**) length of **128**, and batch sizes (**BS**) of **1, 16, 32, and 64**. **S.-OO.** and **S.-G.** denote the speedup with respect to OpenOrca and GLUE, respectively.

| Method | Rank | # Trainable Parameters | OpenOrca ↓ | GLUE ↑ | Throughput ↑ | | | E2E Latency ↓ | | | Decoding Latency ↓ | | | Prefill Latency ↓ | | |
|---|---|---|---|---|---|---|---|---|---|---|---|---|---|---|---|---|
| | | | | | Token/s | S.-OO. | S.-G. | Time | S.-OO. | S.-G. | Time | S.-OO. | S.-G. | Time | S.-OO. | S.-G. |
| | | | | | | | | BS=1 | | | | | | | | |
| | 16 | 41.9M | 2.316 | 75.82 | 103.5 | 1.00x | 1.00x | 1.24 | 1.00x | 1.00x | 0.0088 | 1.00x | 1.00x | 0.108 | 1.00x | 1.00x |
| | 32 | 83.9M | 2.309 | 75.87 | 102.3 | 1.00x | 1.00x | 1.25 | 1.00x | 1.00x | 0.0089 | 1.00x | 1.00x | 0.109 | 1.00x | 1.00x |
| **S-LoRA** | 64 | 167.8M | 2.303 | 76.01 | 100.2 | 1.00x | 1.00x | 1.28 | 1.00x | 1.00x | 0.0091 | 1.00x | 1.00x | 0.111 | 1.00x | 1.00x |
| | 128 | 335.5M | 2.297 | 76.28 | 96.5 | 1.00x | 1.00x | 1.33 | 1.00x | 1.00x | 0.0095 | 1.00x | 1.00x | 0.108 | 1.00x | 1.00x |
| | 256 | 671.1M | 2.290 | 76.39 | 86.6 | 1.00x | 1.00x | 1.48 | 1.00x | 1.00x | 0.0107 | 1.00x | 1.00x | 0.112 | 1.00x | 1.00x |
| | 32 | 36.2M | 2.318 | 75.58 | 131.7 | N/A | N/A | 0.97 | N/A | N/A | 0.0069 | N/A | N/A | 0.091 | N/A | N/A |
| | 64 | 72.4M | 2.310 | 75.90 | 131.7 | 1.27x | 1.29x | 0.97 | 1.27x | 1.29x | 0.0069 | 1.28x | 1.29x | 0.088 | 1.23x | 1.24x |
| **BD-LoRA** | 128 | 144.7M | 2.303 | 76.17 | 130.6 | 1.28x | 1.30x | 0.98 | 1.28x | 1.30x | 0.0069 | 1.29x | 1.31x | 0.091 | 1.20x | 1.22x |
| | 256 | 289.4M | 2.296 | 76.55 | 130.4 | 1.35x | 1.51x | 0.98 | 1.35x | 1.51x | 0.0070 | 1.36x | 1.52x | 0.086 | 1.26x | 1.30x |
| | 512 | 578.8M | **2.289** | **76.59** | 121.5 | 1.40x | 1.40x | 1.05 | 1.40x | 1.40x | 0.0076 | 1.41x | 1.41x | 0.085 | 1.32x | 1.32x |
| | | | | | | | | BS=16 | | | | | | | | |
| | 16 | 41.9M | 2.316 | 75.82 | 51.9 | 1.00x | 1.00x | 2.47 | 1.00x | 1.00x | 0.0150 | 1.00x | 1.00x | 0.549 | 1.00x | 1.00x |
| | 32 | 83.9M | 2.309 | 75.87 | 51.9 | 1.00x | 1.00x | 2.47 | 1.00x | 1.00x | 0.0152 | 1.00x | 1.00x | 0.518 | 1.00x | 1.00x |
| **S-LoRA** | 64 | 167.8M | 2.303 | 76.01 | 46.7 | 1.00x | 1.00x | 2.74 | 1.00x | 1.00x | 0.0169 | 1.00x | 1.00x | 0.582 | 1.00x | 1.00x |
| | 128 | 335.5M | 2.297 | 76.28 | 41.1 | 1.00x | 1.00x | 3.12 | 1.00x | 1.00x | 0.0199 | 1.00x | 1.00x | 0.570 | 1.00x | 1.00x |
| | 256 | 671.1M | 2.290 | 76.39 | 32.5 | 1.00x | 1.00x | 3.93 | 1.00x | 1.00x | 0.0266 | 1.00x | 1.00x | 0.528 | 1.00x | 1.00x |
| | 32 | 36.2M | 2.318 | 75.58 | 60.1 | N/A | N/A | 2.13 | N/A | N/A | 0.0129 | N/A | N/A | 0.480 | N/A | N/A |
| | 64 | 72.4M | 2.310 | 75.90 | 59.6 | 1.15x | 1.15x | 2.15 | 1.15x | 1.15x | 0.0130 | 1.15x | 1.17x | 0.481 | 1.14x | 1.08x |
| **BD-LoRA** | 128 | 144.7M | 2.303 | 76.17 | 61.2 | 1.18x | 1.31x | 2.09 | 1.18x | 1.31x | 0.0128 | 1.18x | 1.31x | 0.445 | 1.16x | 1.31x |
| | 256 | 289.4M | 2.296 | 76.55 | 56.4 | 1.37x | 1.73x | 2.27 | 1.38x | 1.73x | 0.0140 | 1.42x | 1.90x | 0.474 | 1.20x | 1.12x |
| | 512 | 578.8M | **2.289** | **76.59** | 44.2 | 1.36x | 1.36x | 2.89 | 1.36x | 1.36x | 0.0188 | 1.41x | 1.41x | 0.483 | 1.09x | 1.09x |
| | | | | | | | | BS=32 | | | | | | | | |
| | 16 | 41.9M | 2.316 | 75.82 | 35.3 | 1.00x | 1.00x | 3.63 | 1.00x | 1.00x | 0.0208 | 1.00x | 1.00x | 0.962 | 1.00x | 1.00x |
| | 32 | 83.9M | 2.309 | 75.87 | 34.0 | 1.00x | 1.00x | 3.77 | 1.00x | 1.00x | 0.0219 | 1.00x | 1.00x | 0.960 | 1.00x | 1.00x |
| **S-LoRA** | 64 | 167.8M | 2.303 | 76.01 | 33.7 | 1.00x | 1.00x | 3.80 | 1.00x | 1.00x | 0.0230 | 1.00x | 1.00x | 0.848 | 1.00x | 1.00x |
| | 128 | 335.5M | 2.297 | 76.28 | 27.6 | 1.00x | 1.00x | 4.64 | 1.00x | 1.00x | 0.0295 | 1.00x | 1.00x | 0.851 | 1.00x | 1.00x |
| | 256 | 671.1M | 2.290 | 76.39 | 20.0 | 1.00x | 1.00x | 6.42 | 1.00x | 1.00x | 0.0428 | 1.00x | 1.00x | 0.932 | 1.00x | 1.00x |
| | 32 | 36.2M | 2.318 | 75.58 | 42.3 | N/A | N/A | 3.03 | N/A | N/A | 0.0173 | N/A | N/A | 0.806 | N/A | N/A |
| | 64 | 72.4M | 2.310 | 75.90 | 44.1 | 1.25x | 1.30x | 2.90 | 1.25x | 1.30x | 0.0171 | 1.21x | 1.28x | 0.706 | 1.36x | **1.36x** |
| **BD-LoRA** | 128 | 144.7M | 2.303 | 76.17 | 41.1 | 1.21x | 1.22x | 3.11 | 1.21x | 1.22x | 0.0181 | 1.21x | 1.27x | 0.790 | 1.21x | 1.07x |
| | 256 | 289.4M | 2.296 | 76.55 | 38.2 | 1.38x | 1.91x | 3.35 | 1.38x | 1.91x | 0.0201 | 1.47x | 2.13x | 0.778 | 1.09x | 1.20x |
| | 512 | 578.8M | **2.289** | **76.59** | 28.7 | **1.44x** | 1.44x | 4.46 | **1.44x** | 1.44x | 0.0287 | 1.49x | 1.49x | 0.785 | 1.19x | 1.19x |
| | | | | | | | | BS=64 | | | | | | | | |
| | 16 | 41.9M | 2.316 | 75.82 | 22.7 | 1.00x | 1.00x | 5.64 | 1.00x | 1.00x | 0.0318 | 1.00x | 1.00x | 1.565 | 1.00x | 1.00x |
| | 32 | 83.9M | 2.309 | 75.87 | 22.3 | 1.00x | 1.00x | 5.76 | 1.00x | 1.00x | 0.0331 | 1.00x | 1.00x | 1.508 | 1.00x | 1.00x |
| **S-LoRA** | 64 | 167.8M | 2.303 | 76.01 | 20.3 | 1.00x | 1.00x | 6.32 | 1.00x | 1.00x | 0.0378 | 1.00x | 1.00x | 1.484 | 1.00x | 1.00x |
| | 128 | 335.5M | 2.297 | 76.28 | 16.7 | 1.00x | 1.00x | 7.66 | 1.00x | 1.00x | 0.0489 | 1.00x | 1.00x | 1.394 | 1.00x | 1.00x |
| | 256 | 671.1M | 2.290 | 76.39 | 11.4 | 1.00x | 1.00x | 11.21 | 1.00x | 1.00x | 0.0759 | 1.00x | 1.00x | 1.486 | 1.00x | 1.00x |
| | 32 | 36.2M | 2.318 | 75.58 | 28.8 | N/A | N/A | 4.46 | N/A | N/A | 0.0259 | N/A | N/A | 1.132 | N/A | N/A |
| | 64 | 72.4M | 2.310 | 75.90 | 27.9 | 1.23x | 1.25x | 4.59 | 1.23x | 1.25x | 0.0269 | 1.18x | 1.23x | 1.141 | **1.37x** | 1.32x |
| **BD-LoRA** | 128 | 144.7M | 2.303 | 76.17 | 27.3 | 1.22x | 1.34x | 4.71 | 1.22x | 1.34x | 0.0275 | 1.20x | 1.37x | 1.180 | 1.28x | 1.26x |
| | 256 | 289.4M | 2.296 | 76.55 | 23.7 | 1.41x | **2.07x** | 5.41 | 1.42x | **2.07x** | 0.0325 | **1.51x** | **2.34x** | 1.247 | 1.12x | 1.19x |
| | 512 | 578.8M | **2.289** | **76.59** | 16.3 | 1.43x | 1.43x | 7.84 | 1.43x | 1.43x | 0.0509 | 1.49x | 1.49x | 1.314 | 1.13x | 1.13x |

Table 5: Speedup of throughput, end-to-end (E2E) latency, decoding latency, and prefill latency of **Llama-3.1-8B** with an input token (**IT**) length of **4096**, an output token (**OT**) length of **256**, and batch sizes (**BS**) of **1, 16, 32, and 64**. **S.-OO.** and **S.-G.** denote the speedup with respect to OpenOrca and GLUE, respectively.

| Method | Rank | # Trainable Parameters | OpenOrca↓ | GLUE↑ | Throughput↑ | | | E2E Latency↓ | | | Decoding Latency↓ | | | Prefill Latency↓ | | |
|---|---|---|---|---|---|---|---|---|---|---|---|---|---|---|---|---|
| | | | | | Token/s | S.-OO. | S.-G. | Time | S.-OO. | S.-G. | Time | S.-OO. | S.-G. | Time | S.-OO. | S.-G. |
| | | | | | | | | BS=1 | | | | | | | | |
| | 16 | 41.9M | 2.316 | 75.82 | 130.9 | 1.00x | 1.00x | 1.96 | 1.00x | 1.00x | 0.0072 | 1.00x | 1.00x | 0.109 | 1.00x | 1.00x |
| | 32 | 83.9M | 2.309 | 75.87 | 129.7 | 1.00x | 1.00x | 1.97 | 1.00x | 1.00x | 0.0073 | 1.00x | 1.00x | 0.113 | 1.00x | 1.00x |
| NFS-LoRA | 64 | 167.8M | 2.303 | 76.01 | 121.2 | 1.00x | 1.00x | 2.11 | 1.00x | 1.00x | 0.0078 | 1.00x | 1.00x | 0.121 | 1.00x | 1.00x |
| | 128 | 335.5M | 2.297 | 76.28 | 112.2 | 1.00x | 1.00x | 2.28 | 1.00x | 1.00x | 0.0084 | 1.00x | 1.00x | 0.139 | 1.00x | 1.00x |
| | 256 | 671.1M | 2.290 | 76.39 | 94.5 | 1.00x | 1.00x | 2.71 | 1.00x | 1.00x | 0.0099 | 1.00x | 1.00x | 0.179 | 1.00x | 1.00x |
| | 32 | 36.2M | 2.318 | 75.58 | 132.2 | N/A | N/A | 1.94 | N/A | N/A | 0.0071 | N/A | N/A | 0.116 | N/A | N/A |
| | 64 | 72.4M | 2.310 | 75.90 | 131.5 | 1.00x | 1.01x | 1.95 | 1.00x | 1.01x | 0.0071 | 1.01x | 1.02x | 0.117 | 0.93x | 0.97x |
| BD-LoRA | 128 | 144.7M | 2.303 | 76.17 | 131.1 | 1.01x | 1.08x | 1.95 | 1.01x | 1.08x | 0.0072 | 1.01x | 1.08x | 0.110 | 1.03x | 1.10x |
| | 256 | 289.4M | 2.296 | 76.55 | 130.1 | 1.16x | 1.38x | 1.97 | 1.16x | 1.38x | 0.0072 | 1.15x | 1.36x | 0.111 | 1.25x | 1.60x |
| | 512 | 578.8M | **2.289** | **76.59** | 120.6 | 1.28x | 1.28x | 2.12 | 1.28x | 1.28x | 0.0078 | 1.26x | 1.26x | 0.122 | 1.46x | 1.46x |
| | | | | | | | | BS=16 | | | | | | | | |
| | 16 | 41.9M | 2.316 | 75.82 | 58.0 | 1.00x | 1.00x | 4.41 | 1.00x | 1.00x | 0.0139 | 1.00x | 1.00x | 0.862 | 1.00x | 1.00x |
| | 32 | 83.9M | 2.309 | 75.87 | 54.6 | 1.00x | 1.00x | 4.69 | 1.00x | 1.00x | 0.0149 | 1.00x | 1.00x | 0.883 | 1.00x | 1.00x |
| NFS-LoRA | 64 | 167.8M | 2.303 | 76.01 | 46.8 | 1.00x | 1.00x | 5.47 | 1.00x | 1.00x | 0.0176 | 1.00x | 1.00x | 0.961 | 1.00x | 1.00x |
| | 128 | 335.5M | 2.297 | 76.28 | 34.9 | 1.00x | 1.00x | 7.34 | 1.00x | 1.00x | 0.0243 | 1.00x | 1.00x | 1.117 | 1.00x | 1.00x |
| | 256 | 671.1M | 2.290 | 76.39 | 23.6 | 1.00x | 1.00x | 10.83 | 1.00x | 1.00x | 0.0367 | 1.00x | 1.00x | 1.440 | 1.00x | 1.00x |
| | 32 | 36.2M | 2.318 | 75.58 | 57.9 | N/A | N/A | 4.42 | N/A | N/A | 0.0137 | N/A | N/A | 0.907 | N/A | N/A |
| | 64 | 72.4M | 2.310 | 75.90 | 57.3 | 0.99x | 1.05x | 4.47 | 0.99x | 1.05x | 0.0140 | 0.99x | 1.07x | 0.896 | 0.96x | 0.99x |
| BD-LoRA | 128 | 144.7M | 2.303 | 76.17 | 58.0 | 1.06x | 1.24x | 4.42 | 1.06x | 1.24x | 0.0140 | 1.07x | 1.26x | 0.842 | 1.05x | 1.14x |
| | 256 | 289.4M | 2.296 | 76.55 | 54.1 | 1.55x | 2.29x | 4.74 | 1.55x | 2.29x | 0.0151 | 1.61x | 2.43x | 0.872 | 1.28x | 1.65x |
| | 512 | 578.8M | **2.289** | **76.59** | 41.6 | 1.76x | 1.76x | 6.15 | 1.76x | 1.76x | 0.0203 | 1.81x | 1.81x | 0.964 | 1.49x | 1.49x |
| | | | | | | | | BS=32 | | | | | | | | |
| | 16 | 41.9M | 2.316 | 75.82 | 38.6 | 1.00x | 1.00x | 6.63 | 1.00x | 1.00x | 0.0198 | 1.00x | 1.00x | 1.564 | 1.00x | 1.00x |
| | 32 | 83.9M | 2.309 | 75.87 | 35.3 | 1.00x | 1.00x | 7.25 | 1.00x | 1.00x | 0.0220 | 1.00x | 1.00x | 1.617 | 1.00x | 1.00x |
| NFS-LoRA | 64 | 167.8M | 2.303 | 76.01 | 29.6 | 1.00x | 1.00x | 8.65 | 1.00x | 1.00x | 0.0268 | 1.00x | 1.00x | 1.790 | 1.00x | 1.00x |
| | 128 | 335.5M | 2.297 | 76.28 | 21.0 | 1.00x | 1.00x | 12.21 | 1.00x | 1.00x | 0.0396 | 1.00x | 1.00x | 2.070 | 1.00x | 1.00x |
| | 256 | 671.1M | 2.290 | 76.39 | 13.5 | 1.00x | 1.00x | 18.99 | 1.00x | 1.00x | 0.0637 | 1.00x | 1.00x | 2.671 | 1.00x | 1.00x |
| | 32 | 36.2M | 2.318 | 75.58 | 38.2 | N/A | N/A | 6.70 | N/A | N/A | 0.0197 | N/A | N/A | 1.667 | N/A | N/A |
| | 64 | 72.4M | 2.310 | 75.90 | 38.0 | 0.98x | 1.08x | 6.74 | 0.98x | 1.08x | 0.0201 | 0.99x | 1.10x | 1.596 | 0.98x | 1.01x |
| BD-LoRA | 128 | 144.7M | 2.303 | 76.17 | 38.3 | 1.09x | 1.30x | 6.68 | 1.09x | 1.30x | 0.0201 | 1.10x | 1.34x | 1.544 | 1.05x | 1.16x |
| | 256 | 289.4M | 2.296 | 76.55 | 34.8 | 1.66x | 2.58x | 7.35 | 1.66x | 2.58x | 0.0224 | 1.77x | 2.84x | 1.613 | 1.28x | 1.66x |
| | 512 | 578.8M | **2.289** | **76.59** | 25.6 | 1.90x | 1.90x | 10.00 | 1.90x | 1.90x | 0.0320 | 1.99x | 1.99x | 1.796 | 1.49x | 1.49x |
| | | | | | | | | BS=64 | | | | | | | | |
| | 16 | 41.9M | 2.316 | 75.82 | 23.1 | 1.00x | 1.00x | 11.07 | 1.00x | 1.00x | 0.0322 | 1.00x | 1.00x | 2.810 | 1.00x | 1.00x |
| | 32 | 83.9M | 2.309 | 75.87 | 21.2 | 1.00x | 1.00x | 12.07 | 1.00x | 1.00x | 0.0358 | 1.00x | 1.00x | 2.887 | 1.00x | 1.00x |
| NFS-LoRA | 64 | 167.8M | 2.303 | 76.01 | 17.2 | 1.00x | 1.00x | 14.90 | 1.00x | 1.00x | 0.0459 | 1.00x | 1.00x | 3.157 | 1.00x | 1.00x |
| | 128 | 335.5M | 2.297 | 76.28 | 11.6 | 1.00x | 1.00x | 22.11 | 1.00x | 1.00x | 0.0717 | 1.00x | 1.00x | 3.749 | 1.00x | 1.00x |
| | 256 | 671.1M | 2.290 | 76.39 | 7.2 | 1.00x | 1.00x | 35.49 | 1.00x | 1.00x | 0.1190 | 1.00x | 1.00x | 5.025 | 1.00x | 1.00x |
| | 32 | 36.2M | 2.318 | 75.58 | 23.0 | N/A | N/A | 11.13 | N/A | N/A | 0.0319 | N/A | N/A | 2.965 | N/A | N/A |
| | 64 | 72.4M | 2.310 | 75.90 | 22.5 | 0.97x | 1.06x | 11.36 | 0.97x | 1.06x | 0.0329 | 0.98x | 1.09x | 2.933 | 0.96x | 0.98x |
| BD-LoRA | 128 | 144.7M | 2.303 | 76.17 | 23.2 | 1.09x | 1.35x | 11.06 | 1.09x | 1.35x | 0.0327 | 1.10x | 1.40x | 2.679 | 1.08x | 1.18x |
| | 256 | 289.4M | 2.296 | 76.55 | 20.6 | 1.78x | **2.85x** | 12.45 | 1.78x | **2.85x** | 0.0374 | 1.91x | **3.18x** | 2.854 | 1.31x | **1.76x** |
| | 512 | 578.8M | **2.289** | **76.59** | 14.5 | **2.01x** | 2.01x | 17.65 | **2.01x** | 2.01x | 0.0566 | **2.10x** | 2.10x | 3.152 | **1.59x** | 1.59x |

Table 6: Speedup of throughput, end-to-end (E2E) latency, decoding latency, and prefill latency of **Llama-3.1-8B** with an input token (**IT**) length of **4096**, an output token (**OT**) length of **256**, and batch sizes (**BS**) of **1, 16, 32, and 64**. **S.-OO.** and **S.-G.** denote the speedup with respect to OpenOrca and GLUE, respectively.

| Method | Rank | # Trainable Parameters | OpenOrca ↓ | GLUE ↑ | Throughput ↑ | | | E2E Latency ↓ | | | Decoding Latency ↓ | | | Prefill Latency ↓ | | |
|---|---|---|---|---|---|---|---|---|---|---|---|---|---|---|---|---|
| | | | | | Token/s | S.-OO. | S.-G. | Time | S.-OO. | S.-G. | Time | S.-OO. | S.-G. | Time | S.-OO. | S.-G. |
| **BS=1** | | | | | | | | | | | | | | | | |
| S-LoRA | 16 | 41.9M | 2.316 | 75.82 | 104.6 | 1.00x | 1.00x | 2.45 | 1.00x | 1.00x | 0.0091 | 1.00x | 1.00x | 0.121 | 1.00x | 1.00x |
| | 32 | 83.9M | 2.309 | 75.87 | 103.8 | 1.00x | 1.00x | 2.47 | 1.00x | 1.00x | 0.0092 | 1.00x | 1.00x | 0.120 | 1.00x | 1.00x |
| | 64 | 167.8M | 2.303 | 76.01 | 102.0 | 1.00x | 1.00x | 2.51 | 1.00x | 1.00x | 0.0093 | 1.00x | 1.00x | 0.121 | 1.00x | 1.00x |
| | 128 | 335.5M | 2.297 | 76.28 | 97.2 | 1.00x | 1.00x | 2.63 | 1.00x | 1.00x | 0.0098 | 1.00x | 1.00x | 0.123 | 1.00x | 1.00x |
| | 256 | 671.1M | 2.290 | 76.39 | 88.2 | 1.00x | 1.00x | 2.90 | 1.00x | 1.00x | 0.0108 | 1.00x | 1.00x | 0.136 | 1.00x | 1.00x |
| BD-LoRA | 32 | 36.2M | 2.318 | 75.58 | 132.2 | N/A | N/A | 1.94 | N/A | N/A | 0.0071 | N/A | N/A | 0.116 | N/A | N/A |
| | 64 | 72.4M | 2.310 | 75.90 | 131.5 | 1.26x | 1.27x | 1.95 | 1.26x | 1.27x | 0.0071 | 1.27x | 1.28x | 0.117 | 1.03x | 1.03x |
| | 128 | 144.7M | 2.303 | 76.17 | 131.1 | 1.26x | 1.29x | 1.95 | 1.26x | 1.29x | 0.0072 | 1.27x | 1.30x | 0.110 | 1.10x | 1.11x |
| | 256 | 289.4M | 2.296 | 76.55 | 130.1 | 1.34x | 1.48x | 1.97 | 1.34x | 1.48x | 0.0072 | 1.35x | 1.49x | 0.111 | 1.11x | 1.22x |
| | 512 | 578.8M | **2.289** | **76.59** | 120.6 | 1.37x | 1.37x | 2.12 | 1.37x | 1.37x | 0.0078 | 1.38x | 1.38x | 0.122 | 1.11x | 1.11x |
| **BS=16** | | | | | | | | | | | | | | | | |
| S-LoRA | 16 | 41.9M | 2.316 | 75.82 | 51.0 | 1.00x | 1.00x | 5.02 | 1.00x | 1.00x | 0.0160 | 1.00x | 1.00x | 0.933 | 1.00x | 1.00x |
| | 32 | 83.9M | 2.309 | 75.87 | 50.1 | 1.00x | 1.00x | 5.11 | 1.00x | 1.00x | 0.0164 | 1.00x | 1.00x | 0.899 | 1.00x | 1.00x |
| | 64 | 167.8M | 2.303 | 76.01 | 46.2 | 1.00x | 1.00x | 5.54 | 1.00x | 1.00x | 0.0179 | 1.00x | 1.00x | 0.966 | 1.00x | 1.00x |
| | 128 | 335.5M | 2.297 | 76.28 | 40.2 | 1.00x | 1.00x | 6.36 | 1.00x | 1.00x | 0.0210 | 1.00x | 1.00x | 0.979 | 1.00x | 1.00x |
| | 256 | 671.1M | 2.290 | 76.39 | 30.4 | 1.00x | 1.00x | 8.41 | 1.00x | 1.00x | 0.0286 | 1.00x | 1.00x | 1.094 | 1.00x | 1.00x |
| BD-LoRA | 32 | 36.2M | 2.318 | 75.58 | 57.9 | N/A | N/A | 4.42 | N/A | N/A | 0.0137 | N/A | N/A | 0.907 | N/A | N/A |
| | 64 | 72.4M | 2.310 | 75.90 | 57.3 | 1.12x | 1.14x | 4.47 | 1.12x | 1.14x | 0.0140 | 1.14x | 1.18x | 0.896 | 1.04x | 1.00x |
| | 128 | 144.7M | 2.303 | 76.17 | 58.0 | 1.16x | 1.25x | 4.42 | 1.16x | 1.25x | 0.0140 | 1.18x | 1.28x | 0.842 | 1.07x | 1.15x |
| | 256 | 289.4M | 2.296 | 76.55 | 54.1 | 1.34x | 1.78x | 4.74 | 1.34x | 1.78x | 0.0151 | 1.39x | 1.89x | 0.872 | 1.12x | 1.25x |
| | 512 | 578.8M | **2.289** | **76.59** | 41.6 | 1.37x | 1.37x | 6.15 | 1.37x | 1.37x | 0.0203 | 1.41x | 1.41x | 0.964 | 1.13x | 1.13x |
| **BS=32** | | | | | | | | | | | | | | | | |
| S-LoRA | 16 | 41.9M | 2.316 | 75.82 | 34.5 | 1.00x | 1.00x | 7.42 | 1.00x | 1.00x | 0.0223 | 1.00x | 1.00x | 1.714 | 1.00x | 1.00x |
| | 32 | 83.9M | 2.309 | 75.87 | 34.7 | 1.00x | 1.00x | 7.37 | 1.00x | 1.00x | 0.0227 | 1.00x | 1.00x | 1.562 | 1.00x | 1.00x |
| | 64 | 167.8M | 2.303 | 76.01 | 30.5 | 1.00x | 1.00x | 8.39 | 1.00x | 1.00x | 0.0258 | 1.00x | 1.00x | 1.789 | 1.00x | 1.00x |
| | 128 | 335.5M | 2.297 | 76.28 | 25.5 | 1.00x | 1.00x | 10.02 | 1.00x | 1.00x | 0.0321 | 1.00x | 1.00x | 1.800 | 1.00x | 1.00x |
| | 256 | 671.1M | 2.290 | 76.39 | 18.7 | 1.00x | 1.00x | 13.65 | 1.00x | 1.00x | 0.0454 | 1.00x | 1.00x | 2.028 | 1.00x | 1.00x |
| BD-LoRA | 32 | 36.2M | 2.318 | 75.58 | 38.2 | N/A | N/A | 6.70 | N/A | N/A | 0.0197 | N/A | N/A | 1.667 | N/A | N/A |
| | 64 | 72.4M | 2.310 | 75.90 | 38.0 | 1.10x | 1.09x | 6.74 | 1.10x | 1.09x | 0.0201 | 1.11x | 1.13x | 1.596 | 1.07x | 0.98x |
| | 128 | 144.7M | 2.303 | 76.17 | 38.3 | 1.10x | 1.26x | 6.68 | 1.10x | 1.26x | 0.0201 | 1.13x | 1.29x | 1.544 | 1.01x | 1.16x |
| | 256 | 289.4M | 2.296 | 76.55 | 34.8 | 1.36x | 1.86x | 7.35 | 1.36x | 1.86x | 0.0224 | 1.43x | 2.03x | 1.613 | 1.12x | 1.26x |
| | 512 | 578.8M | **2.289** | **76.59** | 25.6 | 1.37x | 1.37x | 10.00 | 1.37x | 1.37x | 0.0320 | 1.42x | 1.42x | 1.796 | 1.13x | 1.13x |
| **BS=64** | | | | | | | | | | | | | | | | |
| S-LoRA | 16 | 41.9M | 2.316 | 75.82 | 21.1 | 1.00x | 1.00x | 12.13 | 1.00x | 1.00x | 0.0351 | 1.00x | 1.00x | 3.149 | 1.00x | 1.00x |
| | 32 | 83.9M | 2.309 | 75.87 | 20.2 | 1.00x | 1.00x | 12.65 | 1.00x | 1.00x | 0.0371 | 1.00x | 1.00x | 3.139 | 1.00x | 1.00x |
| | 64 | 167.8M | 2.303 | 76.01 | 18.4 | 1.00x | 1.00x | 13.94 | 1.00x | 1.00x | 0.0419 | 1.00x | 1.00x | 3.213 | 1.00x | 1.00x |
| | 128 | 335.5M | 2.297 | 76.28 | 15.1 | 1.00x | 1.00x | 16.98 | 1.00x | 1.00x | 0.0539 | 1.00x | 1.00x | 3.185 | 1.00x | 1.00x |
| | 256 | 671.1M | 2.290 | 76.39 | 10.5 | 1.00x | 1.00x | 24.43 | 1.00x | 1.00x | 0.0810 | 1.00x | 1.00x | 3.688 | 1.00x | 1.00x |
| BD-LoRA | 32 | 36.2M | 2.318 | 75.58 | 23.0 | N/A | N/A | 11.13 | N/A | N/A | 0.0319 | N/A | N/A | 2.965 | N/A | N/A |
| | 64 | 72.4M | 2.310 | 75.90 | 22.5 | 1.07x | 1.11x | 11.36 | 1.07x | 1.11x | 0.0329 | 1.07x | 1.13x | 2.933 | 1.07x | 1.07x |
| | 128 | 144.7M | 2.303 | 76.17 | 23.2 | 1.14x | 1.26x | 11.06 | 1.14x | 1.26x | 0.0327 | 1.14x | 1.28x | 2.679 | **1.17x** | 1.20x |
| | 256 | 289.4M | 2.296 | 76.55 | 20.6 | 1.36x | **1.96x** | 12.45 | 1.36x | **1.96x** | 0.0374 | **1.44x** | **2.16x** | 2.854 | 1.12x | **1.29x** |
| | 512 | 578.8M | **2.289** | **76.59** | 14.5 | **1.38x** | 1.38x | 17.65 | **1.38x** | 1.38x | 0.0566 | 1.43x | 1.43x | 3.152 | **1.17x** | 1.17x |

Table 7: Speedup of throughput, end-to-end (E2E) latency, decoding latency, and prefill latency of **Llama-3.1-8B** with an input token (**IT**) length of **128**, an output token (**OT**) length of **1024**, and batch sizes (**BS**) of **1, 16, 32, and 64**. **S.-OO.** and **S.-G.** denote the speedup with respect to OpenOrca and GLUE, respectively.

| Method | Rank | # Trainable Parameters | OpenOrca↓ | GLUE↑ | Throughput↑ | | | E2E Latency↓ | | | Decoding Latency↓ | | | Prefill Latency↓ | | |
|---|---|---|---|---|---|---|---|---|---|---|---|---|---|---|---|---|
| | | | | | Token/s | S.-OO. | S.-G. | Time | S.-OO. | S.-G. | Time | S.-OO. | S.-G. | Time | S.-OO. | S.-G. |
| **BS=1** | | | | | | | | | | | | | | | | |
| NFS-LoRA | 16 | 41.9M | 2.316 | 75.82 | 141.7 | 1.00x | 1.00x | 7.23 | 1.00x | 1.00x | 0.0070 | 1.00x | 1.00x | 0.088 | 1.00x | 1.00x |
| | 32 | 83.9M | 2.309 | 75.87 | 137.4 | 1.00x | 1.00x | 7.45 | 1.00x | 1.00x | 0.0072 | 1.00x | 1.00x | 0.095 | 1.00x | 1.00x |
| | 64 | 167.8M | 2.303 | 76.01 | 131.0 | 1.00x | 1.00x | 7.82 | 1.00x | 1.00x | 0.0075 | 1.00x | 1.00x | 0.095 | 1.00x | 1.00x |
| | 128 | 335.5M | 2.297 | 76.28 | 122.5 | 1.00x | 1.00x | 8.36 | 1.00x | 1.00x | 0.0081 | 1.00x | 1.00x | 0.088 | 1.00x | 1.00x |
| | 256 | 671.1M | 2.290 | 76.39 | 101.4 | 1.00x | 1.00x | 10.10 | 1.00x | 1.00x | 0.0098 | 1.00x | 1.00x | 0.096 | 1.00x | 1.00x |
| BD-LoRA | 32 | 36.2M | 2.318 | 75.58 | 143.6 | N/A | N/A | 7.13 | N/A | N/A | 0.0069 | N/A | N/A | 0.088 | N/A | N/A |
| | 64 | 72.4M | 2.310 | 75.90 | 142.7 | 1.01x | 1.04x | 7.18 | 1.01x | 1.04x | 0.0069 | 1.01x | 1.04x | 0.086 | 1.02x | 1.11x |
| | 128 | 144.7M | 2.303 | 76.17 | 141.8 | 1.03x | 1.08x | 7.22 | 1.03x | 1.08x | 0.0070 | 1.03x | 1.08x | 0.088 | 1.09x | 1.09x |
| | 256 | 289.4M | 2.296 | 76.55 | 137.5 | 1.12x | 1.36x | 7.45 | 1.12x | 1.36x | 0.0072 | 1.12x | 1.36x | 0.093 | 0.95x | 1.03x |
| | 512 | 578.8M | **2.289** | **76.59** | 131.2 | 1.29x | 1.29x | 7.81 | 1.29x | 1.29x | 0.0075 | 1.30x | 1.30x | 0.087 | 1.11x | 1.11x |
| **BS=16** | | | | | | | | | | | | | | | | |
| NFS-LoRA | 16 | 41.9M | 2.316 | 75.82 | 88.6 | 1.00x | 1.00x | 11.57 | 1.00x | 1.00x | 0.0111 | 1.00x | 1.00x | 0.248 | 1.00x | 1.00x |
| | 32 | 83.9M | 2.309 | 75.87 | 83.5 | 1.00x | 1.00x | 12.27 | 1.00x | 1.00x | 0.0117 | 1.00x | 1.00x | 0.248 | 1.00x | 1.00x |
| | 64 | 167.8M | 2.303 | 76.01 | 70.3 | 1.00x | 1.00x | 14.56 | 1.00x | 1.00x | 0.0140 | 1.00x | 1.00x | 0.252 | 1.00x | 1.00x |
| | 128 | 335.5M | 2.297 | 76.28 | 49.3 | 1.00x | 1.00x | 20.77 | 1.00x | 1.00x | 0.0201 | 1.00x | 1.00x | 0.233 | 1.00x | 1.00x |
| | 256 | 671.1M | 2.290 | 76.39 | 31.5 | 1.00x | 1.00x | 32.54 | 1.00x | 1.00x | 0.0315 | 1.00x | 1.00x | 0.289 | 1.00x | 1.00x |
| BD-LoRA | 32 | 36.2M | 2.318 | 75.58 | 90.4 | N/A | N/A | 11.32 | N/A | N/A | 0.0108 | N/A | N/A | 0.261 | N/A | N/A |
| | 64 | 72.4M | 2.310 | 75.90 | 89.0 | 1.01x | 1.07x | 11.50 | 1.01x | 1.07x | 0.0110 | 1.00x | 1.07x | 0.232 | 1.07x | 1.07x |
| | 128 | 144.7M | 2.303 | 76.17 | 88.3 | 1.06x | 1.26x | 11.59 | 1.06x | 1.26x | 0.0111 | 1.06x | 1.26x | 0.236 | 1.05x | 1.06x |
| | 256 | 289.4M | 2.296 | 76.55 | 81.4 | 1.65x | 2.59x | 12.58 | 1.65x | 2.59x | 0.0120 | 1.66x | 2.61x | 0.245 | 0.95x | 1.18x |
| | 512 | 578.8M | **2.289** | **76.59** | 59.0 | 1.87x | 1.87x | 17.36 | 1.87x | 1.87x | 0.0167 | 1.89x | 1.89x | 0.256 | **1.13x** | 1.13x |
| **BS=32** | | | | | | | | | | | | | | | | |
| NFS-LoRA | 16 | 41.9M | 2.316 | 75.82 | 73.9 | 1.00x | 1.00x | 13.85 | 1.00x | 1.00x | 0.0132 | 1.00x | 1.00x | 0.291 | 1.00x | 1.00x |
| | 32 | 83.9M | 2.309 | 75.87 | 65.4 | 1.00x | 1.00x | 15.65 | 1.00x | 1.00x | 0.0150 | 1.00x | 1.00x | 0.292 | 1.00x | 1.00x |
| | 64 | 167.8M | 2.303 | 76.01 | 50.4 | 1.00x | 1.00x | 20.31 | 1.00x | 1.00x | 0.0195 | 1.00x | 1.00x | 0.285 | 1.00x | 1.00x |
| | 128 | 335.5M | 2.297 | 76.28 | 31.7 | 1.00x | 1.00x | 32.30 | 1.00x | 1.00x | 0.0313 | 1.00x | 1.00x | 0.279 | 1.00x | 1.00x |
| | 256 | 671.1M | 2.290 | 76.39 | 18.8 | 1.00x | 1.00x | 54.46 | 1.00x | 1.00x | 0.0529 | 1.00x | 1.00x | 0.300 | 1.00x | 1.00x |
| BD-LoRA | 32 | 36.2M | 2.318 | 75.58 | 77.7 | N/A | N/A | 13.19 | N/A | N/A | 0.0126 | N/A | N/A | 0.264 | N/A | N/A |
| | 64 | 72.4M | 2.310 | 75.90 | 73.7 | 1.00x | 1.13x | 13.89 | 1.00x | 1.13x | 0.0133 | 0.99x | 1.13x | 0.258 | **1.13x** | 1.13x |
| | 128 | 144.7M | 2.303 | 76.17 | 72.6 | 1.11x | 1.44x | 14.10 | 1.11x | 1.44x | 0.0135 | 1.11x | 1.45x | 0.277 | 1.06x | 1.03x |
| | 256 | 289.4M | 2.296 | 76.55 | 63.0 | 1.99x | 3.35x | 16.27 | 1.99x | 3.35x | 0.0156 | 2.00x | 3.39x | 0.277 | 1.01x | 1.08x |
| | 512 | 578.8M | **2.289** | **76.59** | 40.0 | 2.13x | 2.13x | 25.58 | 2.13x | 2.13x | 0.0247 | 2.14x | 2.14x | 0.276 | 1.09x | 1.09x |
| **BS=64** | | | | | | | | | | | | | | | | |
| NFS-LoRA | 16 | 41.9M | 2.316 | 75.82 | 52.4 | 1.00x | 1.00x | 19.53 | 1.00x | 1.00x | 0.0188 | 1.00x | 1.00x | 0.306 | 1.00x | 1.00x |
| | 32 | 83.9M | 2.309 | 75.87 | 44.7 | 1.00x | 1.00x | 22.90 | 1.00x | 1.00x | 0.0221 | 1.00x | 1.00x | 0.301 | 1.00x | 1.00x |
| | 64 | 167.8M | 2.303 | 76.01 | 31.9 | 1.00x | 1.00x | 32.07 | 1.00x | 1.00x | 0.0309 | 1.00x | 1.00x | 0.397 | 1.00x | 1.00x |
| | 128 | 335.5M | 2.297 | 76.28 | 18.3 | 1.00x | 1.00x | 55.82 | 1.00x | 1.00x | 0.0542 | 1.00x | 1.00x | 0.329 | 1.00x | 1.00x |
| | 256 | 671.1M | 2.290 | 76.39 | 10.3 | 1.00x | 1.00x | 99.83 | 1.00x | 1.00x | 0.0971 | 1.00x | 1.00x | 0.345 | 1.00x | 1.00x |
| BD-LoRA | 32 | 36.2M | 2.318 | 75.58 | 56.1 | N/A | N/A | 18.27 | N/A | N/A | 0.0175 | N/A | N/A | 0.316 | N/A | N/A |
| | 64 | 72.4M | 2.310 | 75.90 | 51.9 | 0.99x | 1.16x | 19.74 | 0.99x | 1.16x | 0.0191 | 0.99x | 1.16x | 0.321 | 0.95x | 0.94x |
| | 128 | 144.7M | 2.303 | 76.17 | 50.6 | 1.13x | 1.59x | 20.23 | 1.13x | 1.59x | 0.0194 | 1.13x | 1.59x | 0.315 | 0.96x | **1.26x** |
| | 256 | 289.4M | 2.296 | 76.55 | 41.5 | 2.26x | **4.05x** | 24.68 | 2.26x | **4.05x** | 0.0238 | 2.28x | **4.09x** | 0.336 | 0.98x | 1.03x |
| | 512 | 578.8M | **2.289** | **76.59** | 24.0 | **2.34x** | 2.34x | 42.66 | **2.34x** | 2.34x | 0.0413 | **2.35x** | 2.35x | 0.340 | 1.02x | 1.02x |

Table 8: Speedup of throughput, end-to-end (E2E) latency, decoding latency, and prefill latency of **Llama-3.1-8B** with an input token (**IT**) length of **128**, an output token (**OT**) length of **1024**, and batch sizes (**BS**) of **1, 16, 32, and 64**. **S.-OO.** and **S.-G.** denote the speedup with respect to OpenOrca and GLUE, respectively.

| Method | Rank | # Trainable Parameters | OpenOrca ↓ | GLUE ↑ | Throughput ↑ | | | E2E Latency ↓ | | | Decoding Latency ↓ | | | Prefill Latency ↓ | | |
|---|---|---|---|---|---|---|---|---|---|---|---|---|---|---|---|---|
| | | | | | Token/s | S.-OO. | S.-G. | Time | S.-OO. | S.-G. | Time | S.-OO. | S.-G. | Time | S.-OO. | S.-G. |
| **BS=1** | | | | | | | | | | | | | | | | |
| | 16 | 41.9M | 2.316 | 75.82 | 110.7 | 1.00x | 1.00x | 9.25 | 1.00x | 1.00x | 0.0089 | 1.00x | 1.00x | 0.104 | 1.00x | 1.00x |
| | 32 | 83.9M | 2.309 | 75.87 | 110.4 | 1.00x | 1.00x | 9.28 | 1.00x | 1.00x | 0.0090 | 1.00x | 1.00x | 0.101 | 1.00x | 1.00x |
| **S-LoRA** | 64 | 167.8M | 2.303 | 76.01 | 108.2 | 1.00x | 1.00x | 9.47 | 1.00x | 1.00x | 0.0091 | 1.00x | 1.00x | 0.105 | 1.00x | 1.00x |
| | 128 | 335.5M | 2.297 | 76.28 | 104.7 | 1.00x | 1.00x | 9.78 | 1.00x | 1.00x | 0.0095 | 1.00x | 1.00x | 0.101 | 1.00x | 1.00x |
| | 256 | 671.1M | 2.290 | 76.39 | 92.8 | 1.00x | 1.00x | 11.03 | 1.00x | 1.00x | 0.0107 | 1.00x | 1.00x | 0.102 | 1.00x | 1.00x |
| | 32 | 36.2M | 2.318 | 75.58 | 143.6 | N/A | N/A | 7.13 | N/A | N/A | 0.0069 | N/A | N/A | 0.088 | N/A | N/A |
| | 64 | 72.4M | 2.310 | 75.90 | 142.7 | 1.29x | 1.29x | 7.18 | 1.29x | 1.29x | 0.0069 | 1.29x | 1.29x | 0.086 | 1.21x | 1.17x |
| **BD-LoRA** | 128 | 144.7M | 2.303 | 76.17 | 141.8 | 1.28x | 1.31x | 7.22 | 1.28x | 1.31x | 0.0070 | 1.29x | 1.31x | 0.088 | 1.15x | 1.20x |
| | 256 | 289.4M | 2.296 | 76.55 | 137.5 | 1.31x | 1.48x | 7.45 | 1.31x | 1.48x | 0.0072 | 1.32x | 1.49x | 0.093 | 1.09x | 1.09x |
| | 512 | 578.8M | **2.289** | **76.59** | 131.2 | 1.41x | 1.41x | 7.81 | 1.41x | 1.41x | 0.0075 | 1.42x | 1.42x | 0.087 | 1.18x | 1.18x |
| **BS=16** | | | | | | | | | | | | | | | | |
| | 16 | 41.9M | 2.316 | 75.82 | 76.7 | 1.00x | 1.00x | 13.36 | 1.00x | 1.00x | 0.0128 | 1.00x | 1.00x | 0.297 | 1.00x | 1.00x |
| | 32 | 83.9M | 2.309 | 75.87 | 73.6 | 1.00x | 1.00x | 13.92 | 1.00x | 1.00x | 0.0133 | 1.00x | 1.00x | 0.288 | 1.00x | 1.00x |
| **S-LoRA** | 64 | 167.8M | 2.303 | 76.01 | 68.2 | 1.00x | 1.00x | 15.01 | 1.00x | 1.00x | 0.0144 | 1.00x | 1.00x | 0.283 | 1.00x | 1.00x |
| | 128 | 335.5M | 2.297 | 76.28 | 56.3 | 1.00x | 1.00x | 18.19 | 1.00x | 1.00x | 0.0175 | 1.00x | 1.00x | 0.287 | 1.00x | 1.00x |
| | 256 | 671.1M | 2.290 | 76.39 | 40.3 | 1.00x | 1.00x | 25.41 | 1.00x | 1.00x | 0.0245 | 1.00x | 1.00x | 0.292 | 1.00x | 1.00x |
| | 32 | 36.2M | 2.318 | 75.58 | 90.4 | N/A | N/A | 11.32 | N/A | N/A | 0.0108 | N/A | N/A | 0.261 | N/A | N/A |
| | 64 | 72.4M | 2.310 | 75.90 | 89.0 | 1.16x | 1.21x | 11.50 | 1.16x | 1.21x | 0.0110 | 1.16x | 1.21x | 0.232 | 1.28x | 1.24x |
| **BD-LoRA** | 128 | 144.7M | 2.303 | 76.17 | 88.3 | 1.20x | 1.29x | 11.59 | 1.20x | 1.29x | 0.0111 | 1.20x | 1.30x | 0.236 | 1.22x | 1.20x |
| | 256 | 289.4M | 2.296 | 76.55 | 81.4 | 1.45x | 2.02x | 12.58 | 1.45x | 2.02x | 0.0120 | 1.45x | 2.04x | 0.245 | 1.17x | 1.19x |
| | 512 | 578.8M | **2.289** | **76.59** | 59.0 | 1.46x | 1.46x | 17.36 | 1.46x | 1.46x | 0.0167 | 1.47x | 1.47x | 0.256 | 1.14x | 1.14x |
| **BS=32** | | | | | | | | | | | | | | | | |
| | 16 | 41.9M | 2.316 | 75.82 | 64.0 | 1.00x | 1.00x | 16.01 | 1.00x | 1.00x | 0.0153 | 1.00x | 1.00x | 0.343 | 1.00x | 1.00x |
| | 32 | 83.9M | 2.309 | 75.87 | 60.6 | 1.00x | 1.00x | 16.91 | 1.00x | 1.00x | 0.0162 | 1.00x | 1.00x | 0.330 | 1.00x | 1.00x |
| **S-LoRA** | 64 | 167.8M | 2.303 | 76.01 | 53.4 | 1.00x | 1.00x | 19.17 | 1.00x | 1.00x | 0.0184 | 1.00x | 1.00x | 0.328 | 1.00x | 1.00x |
| | 128 | 335.5M | 2.297 | 76.28 | 40.0 | 1.00x | 1.00x | 25.61 | 1.00x | 1.00x | 0.0247 | 1.00x | 1.00x | 0.325 | 1.00x | 1.00x |
| | 256 | 671.1M | 2.290 | 76.39 | 26.5 | 1.00x | 1.00x | 38.67 | 1.00x | 1.00x | 0.0374 | 1.00x | 1.00x | 0.334 | 1.00x | 1.00x |
| | 32 | 36.2M | 2.318 | 75.58 | 77.7 | N/A | N/A | 13.19 | N/A | N/A | 0.0126 | N/A | N/A | 0.264 | N/A | N/A |
| | 64 | 72.4M | 2.310 | 75.90 | 73.7 | 1.15x | 1.22x | 13.89 | 1.15x | 1.22x | 0.0133 | 1.15x | 1.22x | 0.258 | **1.33x** | **1.28x** |
| **BD-LoRA** | 128 | 144.7M | 2.303 | 76.17 | 72.6 | 1.20x | 1.36x | 14.10 | 1.20x | 1.36x | 0.0135 | 1.20x | 1.36x | 0.277 | 1.19x | 1.18x |
| | 256 | 289.4M | 2.296 | 76.55 | 63.0 | 1.57x | 2.38x | 16.27 | 1.57x | 2.38x | 0.0156 | 1.58x | 2.40x | 0.277 | 1.18x | 1.21x |
| | 512 | 578.8M | **2.289** | **76.59** | 40.0 | 1.51x | 1.51x | 25.58 | 1.51x | 1.51x | 0.0247 | 1.52x | 1.52x | 0.276 | 1.21x | 1.21x |
| **BS=64** | | | | | | | | | | | | | | | | |
| | 16 | 41.9M | 2.316 | 75.82 | 48.3 | 1.00x | 1.00x | 21.21 | 1.00x | 1.00x | 0.0204 | 1.00x | 1.00x | 0.353 | 1.00x | 1.00x |
| | 32 | 83.9M | 2.309 | 75.87 | 44.0 | 1.00x | 1.00x | 23.28 | 1.00x | 1.00x | 0.0224 | 1.00x | 1.00x | 0.337 | 1.00x | 1.00x |
| **S-LoRA** | 64 | 167.8M | 2.303 | 76.01 | 36.5 | 1.00x | 1.00x | 28.02 | 1.00x | 1.00x | 0.0270 | 1.00x | 1.00x | 0.341 | 1.00x | 1.00x |
| | 128 | 335.5M | 2.297 | 76.28 | 25.4 | 1.00x | 1.00x | 40.32 | 1.00x | 1.00x | 0.0390 | 1.00x | 1.00x | 0.379 | 1.00x | 1.00x |
| | 256 | 671.1M | 2.290 | 76.39 | 15.5 | 1.00x | 1.00x | 66.17 | 1.00x | 1.00x | 0.0643 | 1.00x | 1.00x | 0.370 | 1.00x | 1.00x |
| | 32 | 36.2M | 2.318 | 75.58 | 56.1 | N/A | N/A | 18.27 | N/A | N/A | 0.0175 | N/A | N/A | 0.316 | N/A | N/A |
| | 64 | 72.4M | 2.310 | 75.90 | 51.9 | 1.07x | 1.18x | 19.74 | 1.07x | 1.18x | 0.0190 | 1.07x | 1.18x | 0.321 | 1.10x | 1.05x |
| **BD-LoRA** | 128 | 144.7M | 2.303 | 76.17 | 50.6 | 1.15x | 1.39x | 20.23 | 1.15x | 1.39x | 0.0194 | 1.15x | 1.39x | 0.315 | 1.07x | 1.08x |
| | 256 | 289.4M | 2.296 | 76.55 | 41.5 | **1.63x** | **2.68x** | 24.68 | **1.63x** | **2.68x** | 0.0238 | **1.64x** | **2.70x** | 0.336 | 1.13x | 1.10x |
| | 512 | 578.8M | **2.289** | **76.59** | 24.0 | 1.55x | 1.55x | 42.66 | 1.55x | 1.55x | 0.0413 | 1.56x | 1.56x | 0.340 | 1.09x | 1.09x |

Table 9: Speedup of throughput, end-to-end (E2E) latency, decoding latency, and prefill latency of **Llama-3.1-8B** with an input token (**IT**) length of **1024**, an output token (**OT**) length of **128**, and batch sizes (**BS**) of **1, 16, 32, and 64**. **S.-0.86x** denotes the speedup when BD-LoRA has 0.86x the number of trainable parameters compared to S-LoRA or NFS-LoRA. **S.-1.73x** denotes the speedup when BD-LoRA has 1.73x the number of trainable parameters compared to S-LoRA or NFS-LoRA.

| Method | Rank | # Trainable Parameters | Throughput ↑ | | | E2E Latency ↓ | | | Decoding Latency ↓ | | | Prefill Latency ↓ | | |
|---|---|---|---|---|---|---|---|---|---|---|---|---|---|---|
| | | | Token/s | S.-0.86x | S.-1.73x | Time | S.-0.86x | S.-1.73x | Time | S.-0.86x | S.-1.73x | Time | S.-0.86x | S.-1.73x |
| | | | | | | Llama-3.1-8B BS=1 | | | | | | | | |
| NFS-LoRA | 16 | 41.9M | 132.9 | 1.00x | 1.00x | 0.96 | 1.00x | 1.00x | 0.0069 | 1.00x | 1.00x | 0.084 | 1.00x | 1.00x |
| | 32 | 83.9M | 125.7 | 1.00x | 1.00x | 1.02 | 1.00x | 1.00x | 0.0072 | 1.00x | 1.00x | 0.099 | 1.00x | 1.00x |
| | 64 | 167.8M | 122.7 | 1.00x | 1.00x | 1.04 | 1.00x | 1.00x | 0.0075 | 1.00x | 1.00x | 0.089 | 1.00x | 1.00x |
| | 128 | 335.5M | 112.6 | 1.00x | 1.00x | 1.14 | 1.00x | 1.00x | 0.0081 | 1.00x | 1.00x | 0.095 | 1.00x | 1.00x |
| | 256 | 671.1M | 95.1 | 1.00x | 1.00x | 1.35 | 1.00x | 1.00x | 0.0098 | 1.00x | 1.00x | 0.097 | 1.00x | 1.00x |
| BD-LoRA | 32 | 36.2M | 131.7 | 0.99x | N/A | 0.97 | 0.99x | N/A | 0.0069 | 1.00x | N/A | 0.091 | 0.92x | N/A |
| | 64 | 72.4M | 131.7 | 1.05x | 0.99x | 0.97 | 1.05x | 0.99x | 0.0069 | 1.04x | 0.99x | 0.088 | 1.13x | 0.96x |
| | 128 | 144.7M | 130.6 | 1.06x | 1.04x | 0.98 | 1.06x | 1.04x | 0.0069 | 1.07x | 1.03x | 0.091 | 0.98x | 1.09x |
| | 256 | 289.4M | 130.4 | 1.16x | 1.06x | 0.98 | 1.16x | 1.06x | 0.0070 | 1.16x | 1.06x | 0.086 | 1.10x | 1.04x |
| | 512 | 578.8M | 121.5 | 1.28x | 1.08x | 1.05 | 1.28x | 1.08x | 0.0076 | 1.29x | 1.08x | 0.085 | 1.14x | 1.11x |
| | | | | | | Llama-3.1-8B BS=16 | | | | | | | | |
| NFS-LoRA | 16 | 41.9M | 58.6 | 1.00x | 1.00x | 2.18 | 1.00x | 1.00x | 0.0131 | 1.00x | 1.00x | 0.504 | 1.00x | 1.00x |
| | 32 | 83.9M | 59.5 | 1.00x | 1.00x | 2.15 | 1.00x | 1.00x | 0.0134 | 1.00x | 1.00x | 0.437 | 1.00x | 1.00x |
| | 64 | 167.8M | 49.9 | 1.00x | 1.00x | 2.56 | 1.00x | 1.00x | 0.0161 | 1.00x | 1.00x | 0.506 | 1.00x | 1.00x |
| | 128 | 335.5M | 39.3 | 1.00x | 1.00x | 3.25 | 1.00x | 1.00x | 0.0218 | 1.00x | 1.00x | 0.456 | 1.00x | 1.00x |
| | 256 | 671.1M | 26.8 | 1.00x | 1.00x | 4.77 | 1.00x | 1.00x | 0.0334 | 1.00x | 1.00x | 0.502 | 1.00x | 1.00x |
| BD-LoRA | 32 | 36.2M | 60.1 | 1.03x | N/A | 2.13 | 1.03x | N/A | 0.0129 | 1.02x | N/A | 0.480 | 1.05x | N/A |
| | 64 | 72.4M | 59.6 | 1.00x | 1.02x | 2.15 | 1.00x | 1.02x | 0.0130 | 1.03x | 1.01x | 0.481 | 0.91x | 1.05x |
| | 128 | 144.7M | 61.2 | 1.23x | 1.03x | 2.09 | 1.23x | 1.03x | 0.0128 | 1.25x | 1.04x | 0.445 | 1.14x | 0.98x |
| | 256 | 289.4M | 56.4 | 1.44x | 1.13x | 2.27 | 1.43x | 1.13x | 0.0140 | 1.56x | 1.15x | 0.474 | 0.96x | 1.07x |
| | 512 | 578.8M | 44.2 | 1.65x | 1.12x | 2.89 | 1.65x | 1.12x | 0.0188 | 1.77x | 1.16x | 0.483 | 1.04x | 0.94x |
| | | | | | | Llama-3.1-8B BS=32 | | | | | | | | |
| NFS-LoRA | 16 | 41.9M | 40.9 | 1.00x | 1.00x | 3.13 | 1.00x | 1.00x | 0.0180 | 1.00x | 1.00x | 0.819 | 1.00x | 1.00x |
| | 32 | 83.9M | 38.6 | 1.00x | 1.00x | 3.32 | 1.00x | 1.00x | 0.0198 | 1.00x | 1.00x | 0.789 | 1.00x | 1.00x |
| | 64 | 167.8M | 32.6 | 1.00x | 1.00x | 3.93 | 1.00x | 1.00x | 0.0243 | 1.00x | 1.00x | 0.808 | 1.00x | 1.00x |
| | 128 | 335.5M | 24.4 | 1.00x | 1.00x | 5.25 | 1.00x | 1.00x | 0.0353 | 1.00x | 1.00x | 0.732 | 1.00x | 1.00x |
| | 256 | 671.1M | 15.6 | 1.00x | 1.00x | 8.20 | 1.00x | 1.00x | 0.0576 | 1.00x | 1.00x | 0.830 | 1.00x | 1.00x |
| BD-LoRA | 32 | 36.2M | 42.3 | 1.03x | N/A | 3.03 | 1.03x | N/A | 0.0173 | 1.04x | N/A | 0.806 | 1.02x | N/A |
| | 64 | 72.4M | 44.1 | 1.14x | 1.08x | 2.90 | 1.14x | 1.08x | 0.0171 | 1.15x | 1.05x | 0.706 | 1.12x | 1.16x |
| | 128 | 144.7M | 41.1 | 1.26x | 1.06x | 3.11 | 1.26x | 1.07x | 0.0181 | 1.34x | 1.09x | 0.790 | 1.02x | 1.00x |
| | 256 | 289.4M | 38.2 | 1.57x | 1.17x | 3.35 | 1.57x | 1.17x | 0.0201 | 1.76x | 1.21x | 0.778 | 0.94x | 1.04x |
| | 512 | 578.8M | 28.7 | 1.84x | 1.18x | 4.46 | 1.84x | 1.18x | 0.0287 | 2.01x | 1.23x | 0.785 | 1.06x | 0.93x |
| | | | | | | Llama-3.1-8B BS=64 | | | | | | | | |
| NFS-LoRA | 16 | 41.9M | 26.3 | 1.00x | 1.00x | 4.90 | 1.00x | 1.00x | 0.0277 | 1.00x | 1.00x | 1.339 | 1.00x | 1.00x |
| | 32 | 83.9M | 24.3 | 1.00x | 1.00x | 5.29 | 1.00x | 1.00x | 0.0314 | 1.00x | 1.00x | 1.267 | 1.00x | 1.00x |
| | 64 | 167.8M | 19.6 | 1.00x | 1.00x | 6.56 | 1.00x | 1.00x | 0.0407 | 1.00x | 1.00x | 1.346 | 1.00x | 1.00x |
| | 128 | 335.5M | 13.5 | 1.00x | 1.00x | 9.52 | 1.00x | 1.00x | 0.0639 | 1.00x | 1.00x | 1.338 | 1.00x | 1.00x |
| | 256 | 671.1M | 8.5 | 1.00x | 1.00x | 15.15 | 1.00x | 1.00x | 0.1074 | 1.00x | 1.00x | 1.388 | 1.00x | 1.00x |
| BD-LoRA | 32 | 36.2M | 28.8 | 1.10x | N/A | 4.46 | 1.10x | N/A | 0.0259 | 1.07x | N/A | 1.132 | **1.18x** | N/A |
| | 64 | 72.4M | 27.9 | 1.15x | 1.06x | 4.59 | 1.15x | 1.07x | 0.0269 | 1.17x | 1.03x | 1.141 | 1.11x | **1.17x** |
| | 128 | 144.7M | 27.3 | 1.39x | 1.12x | 4.71 | 1.39x | 1.12x | 0.0275 | 1.48x | 1.14x | 1.180 | 1.14x | 1.07x |
| | 256 | 289.4M | 23.7 | 1.76x | 1.21x | 5.41 | 1.76x | **1.21x** | 0.0325 | 1.97x | **1.25x** | 1.247 | 1.07x | 1.08x |
| | 512 | 578.8M | 16.3 | **1.93x** | **1.22x** | 7.84 | **1.93x** | **1.21x** | 0.0509 | **2.11x** | **1.25x** | 1.314 | 1.06x | 1.02x |

Table 10: Speedup of throughput, end-to-end (E2E) latency, decoding latency, and prefill latency of **Llama-3.1-8B** with an input token (**IT**) length of **1024**, an output token (**OT**) length of **128**, and batch sizes (**BS**) of **1, 16, 32, and 64**. **S.-0.86x** denotes the speedup when BD-LoRA has 0.86x the number of trainable parameters compared to S-LoRA or NFS-LoRA. **S.-1.73x** denotes the speedup when BD-LoRA has 1.73x the number of trainable parameters compared to S-LoRA or NFS-LoRA.

| Method | Rank | # Trainable Parameters | Throughput ↑ | | | E2E Latency ↓ | | | Decoding Latency ↓ | | | Prefill Latency ↓ | | |
|---|---|---|---|---|---|---|---|---|---|---|---|---|---|---|
| | | | Token/s | S.-0.86x | S.-1.73x | Time | S.-0.86x | S.-1.73x | Time | S.-0.86x | S.-1.73x | Time | S.-0.86x | S.-1.73x |
| **Llama-3.1-8B BS=1** | | | | | | | | | | | | | | |
| | 16 | 41.9M | 103.5 | 1.00x | 1.00x | 1.24 | 1.00x | 1.00x | 0.0088 | 1.00x | 1.00x | 0.108 | 1.00x | 1.00x |
| | 32 | 83.9M | 102.3 | 1.00x | 1.00x | 1.25 | 1.00x | 1.00x | 0.0089 | 1.00x | 1.00x | 0.109 | 1.00x | 1.00x |
| **S-LoRA** | 64 | 167.8M | 100.2 | 1.00x | 1.00x | 1.28 | 1.00x | 1.00x | 0.0091 | 1.00x | 1.00x | 0.111 | 1.00x | 1.00x |
| | 128 | 335.5M | 96.5 | 1.00x | 1.00x | 1.33 | 1.00x | 1.00x | 0.0095 | 1.00x | 1.00x | 0.108 | 1.00x | 1.00x |
| | 256 | 671.1M | 86.6 | 1.00x | 1.00x | 1.48 | 1.00x | 1.00x | 0.0107 | 1.00x | 1.00x | 0.112 | 1.00x | 1.00x |
| | 32 | 36.2M | 131.7 | 1.27x | N/A | 0.97 | 1.27x | N/A | 0.0069 | 1.28x | N/A | 0.091 | 1.18x | N/A |
| | 64 | 72.4M | 131.7 | 1.29x | 1.27x | 0.97 | 1.29x | 1.27x | 0.0069 | 1.29x | 1.28x | 0.088 | 1.24x | 1.23x |
| **BD-LoRA** | 128 | 144.7M | 130.6 | 1.30x | 1.28x | 0.98 | 1.30x | 1.28x | 0.0069 | 1.31x | 1.29x | 0.091 | 1.22x | 1.20x |
| | 256 | 289.4M | 130.4 | 1.35x | **1.30x** | 0.98 | 1.35x | **1.30x** | 0.0070 | 1.36x | **1.30x** | 0.086 | 1.26x | 1.29x |
| | 512 | 578.8M | 121.5 | 1.40x | 1.26x | 1.05 | 1.40x | 1.26x | 0.0076 | 1.41x | 1.26x | 0.085 | 1.32x | 1.27x |
| **Llama-3.1-8B BS=16** | | | | | | | | | | | | | | |
| | 16 | 41.9M | 51.9 | 1.00x | 1.00x | 2.47 | 1.00x | 1.00x | 0.0150 | 1.00x | 1.00x | 0.549 | 1.00x | 1.00x |
| | 32 | 83.9M | 51.9 | 1.00x | 1.00x | 2.47 | 1.00x | 1.00x | 0.0152 | 1.00x | 1.00x | 0.518 | 1.00x | 1.00x |
| **S-LoRA** | 64 | 167.8M | 46.7 | 1.00x | 1.00x | 2.74 | 1.00x | 1.00x | 0.0169 | 1.00x | 1.00x | 0.582 | 1.00x | 1.00x |
| | 128 | 335.5M | 41.1 | 1.00x | 1.00x | 3.12 | 1.00x | 1.00x | 0.0199 | 1.00x | 1.00x | 0.570 | 1.00x | 1.00x |
| | 256 | 671.1M | 32.5 | 1.00x | 1.00x | 3.93 | 1.00x | 1.00x | 0.0266 | 1.00x | 1.00x | 0.528 | 1.00x | 1.00x |
| | 32 | 36.2M | 60.1 | 1.16x | N/A | 2.13 | 1.16x | N/A | 0.0129 | 1.16x | N/A | 0.480 | 1.14x | N/A |
| | 64 | 72.4M | 59.6 | 1.15x | 1.15x | 2.15 | 1.15x | 1.15x | 0.0130 | 1.17x | 1.15x | 0.481 | 1.08x | 1.14x |
| **BD-LoRA** | 128 | 144.7M | 61.2 | 1.31x | 1.18x | 2.09 | 1.31x | 1.18x | 0.0128 | 1.31x | 1.18x | 0.445 | 1.31x | 1.16x |
| | 256 | 289.4M | 56.4 | 1.37x | 1.21x | 2.27 | 1.38x | 1.21x | 0.0140 | 1.42x | 1.20x | 0.474 | 1.20x | 1.23x |
| | 512 | 578.8M | 44.2 | 1.36x | 1.08x | 2.89 | 1.36x | 1.08x | 0.0188 | 1.41x | 1.06x | 0.483 | 1.09x | 1.18x |
| **Llama-3.1-8B BS=32** | | | | | | | | | | | | | | |
| | 16 | 41.9M | 35.3 | 1.00x | 1.00x | 3.63 | 1.00x | 1.00x | 0.0208 | 1.00x | 1.00x | 0.962 | 1.00x | 1.00x |
| | 32 | 83.9M | 34.0 | 1.00x | 1.00x | 3.77 | 1.00x | 1.00x | 0.0219 | 1.00x | 1.00x | 0.960 | 1.00x | 1.00x |
| **S-LoRA** | 64 | 167.8M | 33.7 | 1.00x | 1.00x | 3.80 | 1.00x | 1.00x | 0.0230 | 1.00x | 1.00x | 0.848 | 1.00x | 1.00x |
| | 128 | 335.5M | 27.6 | 1.00x | 1.00x | 4.64 | 1.00x | 1.00x | 0.0295 | 1.00x | 1.00x | 0.851 | 1.00x | 1.00x |
| | 256 | 671.1M | 20.0 | 1.00x | 1.00x | 6.42 | 1.00x | 1.00x | 0.0428 | 1.00x | 1.00x | 0.932 | 1.00x | 1.00x |
| | 32 | 36.2M | 42.3 | 1.20x | N/A | 3.03 | 1.20x | N/A | 0.0173 | 1.20x | N/A | 0.806 | 1.19x | N/A |
| | 64 | 72.4M | 44.1 | 1.30x | 1.25x | 2.90 | 1.30x | 1.25x | 0.0171 | 1.28x | 1.21x | 0.706 | 1.36x | 1.36x |
| **BD-LoRA** | 128 | 144.7M | 41.1 | 1.22x | 1.21x | 3.11 | 1.22x | 1.21x | 0.0181 | 1.27x | 1.21x | 0.790 | 1.07x | 1.21x |
| | 256 | 289.4M | 38.2 | 1.38x | 1.13x | 3.35 | 1.38x | 1.13x | 0.0201 | 1.47x | 1.14x | 0.778 | 1.09x | 1.09x |
| | 512 | 578.8M | 28.7 | **1.44x** | 1.04x | 4.46 | **1.44x** | 1.04x | 0.0287 | 1.49x | 1.03x | 0.785 | 1.19x | 1.08x |
| **Llama-3.1-8B BS=64** | | | | | | | | | | | | | | |
| | 16 | 41.9M | 22.7 | 1.00x | 1.00x | 5.64 | 1.00x | 1.00x | 0.0318 | 1.00x | 1.00x | 1.565 | 1.00x | 1.00x |
| | 32 | 83.9M | 22.3 | 1.00x | 1.00x | 5.76 | 1.00x | 1.00x | 0.0331 | 1.00x | 1.00x | 1.508 | 1.00x | 1.00x |
| **S-LoRA** | 64 | 167.8M | 20.3 | 1.00x | 1.00x | 6.32 | 1.00x | 1.00x | 0.0378 | 1.00x | 1.00x | 1.484 | 1.00x | 1.00x |
| | 128 | 335.5M | 16.7 | 1.00x | 1.00x | 7.66 | 1.00x | 1.00x | 0.0489 | 1.00x | 1.00x | 1.394 | 1.00x | 1.00x |
| | 256 | 671.1M | 11.4 | 1.00x | 1.00x | 11.21 | 1.00x | 1.00x | 0.0759 | 1.00x | 1.00x | 1.486 | 1.00x | 1.00x |
| | 32 | 36.2M | 28.8 | 1.27x | N/A | 4.46 | 1.27x | N/A | 0.0259 | 1.23x | N/A | 1.132 | **1.38x** | N/A |
| | 64 | 72.4M | 27.9 | 1.25x | 1.23x | 4.59 | 1.25x | 1.23x | 0.0269 | 1.23x | 1.18x | 1.141 | 1.32x | **1.37x** |
| **BD-LoRA** | 128 | 144.7M | 27.3 | 1.34x | 1.22x | 4.71 | 1.34x | 1.22x | 0.0275 | 1.37x | 1.20x | 1.180 | 1.26x | 1.28x |
| | 256 | 289.4M | 23.7 | 1.41x | 1.16x | 5.41 | 1.42x | 1.17x | 0.0325 | **1.51x** | 1.16x | 1.247 | 1.12x | 1.19x |
| | 512 | 578.8M | 16.3 | 1.43x | 0.98x | 7.84 | 1.43x | 0.98x | 0.0509 | 1.49x | 0.96x | 1.314 | 1.13x | 1.06x |

Table 11: Speedup of throughput, end-to-end (E2E) latency, decoding latency, and prefill latency of **Llama-3.1-70B** with an input token (**IT**) length of **1024**, an output token (**OT**) length of **128**, and batch sizes (**BS**) of **1, 16, 32, and 64**. **S.-0.87x** denotes the speedup when BD-LoRA has 0.87x the number of trainable parameters compared to S-LoRA or NFS-LoRA. **S.-1.74x** denotes the speedup when BD-LoRA has 1.74x the number of trainable parameters compared to S-LoRA or NFS-LoRA.

| Method | Rank | # Trainable Parameters | Throughput ↑ | | | E2E Latency ↓ | | | Decoding Latency ↓ | | | Prefill Latency ↓ | | |
|---|---|---|---|---|---|---|---|---|---|---|---|---|---|---|
| | | | Token/s | S.-0.87x | S.-1.74x | Time | S.-0.87x | S.-1.74x | Time | S.-0.87x | S.-1.74x | Time | S.-0.87x | S.-1.74x |
| | | | | | | Llama-3.1-70B BS=1 | | | | | | | | |
| NFS-LoRA | 16 | 207.1M | 38.2 | 1.00x | 1.00x | 3.35 | 1.00x | 1.00x | 0.0243 | 1.00x | 1.00x | 0.236 | 1.00x | 1.00x |
| | 32 | 414.2M | 37.4 | 1.00x | 1.00x | 3.42 | 1.00x | 1.00x | 0.0249 | 1.00x | 1.00x | 0.239 | 1.00x | 1.00x |
| | 64 | 828.4M | 36.0 | 1.00x | 1.00x | 3.56 | 1.00x | 1.00x | 0.0259 | 1.00x | 1.00x | 0.241 | 1.00x | 1.00x |
| | 128 | 1656.8M | 34.0 | 1.00x | 1.00x | 3.76 | 1.00x | 1.00x | 0.0276 | 1.00x | 1.00x | 0.233 | 1.00x | 1.00x |
| | 256 | 3313.5M | 29.0 | 1.00x | 1.00x | 4.41 | 1.00x | 1.00x | 0.0323 | 1.00x | 1.00x | 0.279 | 1.00x | 1.00x |
| BD-LoRA | 32 | 180.2M | 38.7 | 1.01x | N/A | 3.31 | 1.01x | N/A | 0.0240 | 1.01x | N/A | 0.228 | 1.03x | N/A |
| | 64 | 360.4M | 38.5 | 1.03x | 1.01x | 3.33 | 1.03x | 1.01x | 0.0242 | 1.03x | 1.01x | 0.232 | 1.03x | 1.02x |
| | 128 | 720.9M | 38.5 | 1.07x | 1.03x | 3.33 | 1.07x | 1.03x | 0.0242 | 1.07x | 1.03x | 0.232 | 1.04x | 1.03x |
| | 256 | 1441.8M | 37.7 | 1.11x | 1.05x | 3.40 | 1.11x | 1.05x | 0.0247 | 1.11x | 1.05x | 0.234 | 1.00x | 1.03x |
| | 512 | 2883.6M | 36.2 | 1.25x | 1.06x | 3.54 | 1.25x | 1.06x | 0.0259 | 1.25x | 1.06x | 0.222 | 1.25x | 1.05x |
| | | | | | | Llama-3.1-70B BS=16 | | | | | | | | |
| NFS-LoRA | 16 | 207.1M | 19.5 | 1.00x | 1.00x | 6.57 | 1.00x | 1.00x | 0.0397 | 1.00x | 1.00x | 1.486 | 1.00x | 1.00x |
| | 32 | 414.2M | 18.8 | 1.00x | 1.00x | 6.82 | 1.00x | 1.00x | 0.0417 | 1.00x | 1.00x | 1.482 | 1.00x | 1.00x |
| | 64 | 828.4M | 16.4 | 1.00x | 1.00x | 7.80 | 1.00x | 1.00x | 0.0485 | 1.00x | 1.00x | 1.600 | 1.00x | 1.00x |
| | 128 | 1656.8M | 12.7 | 1.00x | 1.00x | 10.08 | 1.00x | 1.00x | 0.0645 | 1.00x | 1.00x | 1.821 | 1.00x | 1.00x |
| | 256 | 3313.5M | 8.7 | 1.00x | 1.00x | 14.78 | 1.00x | 1.00x | 0.0977 | 1.00x | 1.00x | 2.270 | 1.00x | 1.00x |
| BD-LoRA | 32 | 180.2M | 19.6 | 1.01x | N/A | 6.51 | 1.01x | N/A | 0.0390 | 1.02x | N/A | 1.525 | 0.97x | N/A |
| | 64 | 360.4M | 19.6 | 1.04x | 1.00x | 6.55 | 1.04x | 1.00x | 0.0394 | 1.06x | 1.01x | 1.502 | 0.99x | 0.99x |
| | 128 | 720.9M | 19.8 | 1.21x | 1.05x | 6.47 | 1.21x | 1.05x | 0.0393 | 1.23x | 1.06x | 1.436 | 1.11x | 1.03x |
| | 256 | 1441.8M | 18.7 | 1.47x | 1.14x | 6.85 | 1.47x | 1.14x | 0.0420 | 1.54x | 1.15x | 1.472 | 1.24x | 1.09x |
| | 512 | 2883.6M | 15.8 | 1.82x | 1.24x | 8.10 | 1.82x | 1.24x | 0.0509 | 1.92x | 1.27x | 1.578 | **1.44x** | **1.15x** |
| | | | | | | Llama-3.1-70B BS=32 | | | | | | | | |
| NFS-LoRA | 16 | 207.1M | 13.3 | 1.00x | 1.00x | 9.65 | 1.00x | 1.00x | 0.0550 | 1.00x | 1.00x | 2.607 | 1.00x | 1.00x |
| | 32 | 414.2M | 12.5 | 1.00x | 1.00x | 10.27 | 1.00x | 1.00x | 0.0596 | 1.00x | 1.00x | 2.641 | 1.00x | 1.00x |
| | 64 | 828.4M | 10.6 | 1.00x | 1.00x | 12.04 | 1.00x | 1.00x | 0.0721 | 1.00x | 1.00x | 2.807 | 1.00x | 1.00x |
| | 128 | 1656.8M | 7.7 | 1.00x | 1.00x | 16.57 | 1.00x | 1.00x | 0.1042 | 1.00x | 1.00x | 3.231 | 1.00x | 1.00x |
| | 256 | 3313.5M | 5.0 | 1.00x | 1.00x | 25.37 | 1.00x | 1.00x | 0.1668 | 1.00x | 1.00x | 4.021 | 1.00x | 1.00x |
| BD-LoRA | 32 | 180.2M | 13.3 | 1.00x | N/A | 9.65 | 1.00x | N/A | 0.0540 | 1.02x | N/A | 2.737 | 0.95x | N/A |
| | 64 | 360.4M | 13.1 | 1.05x | 0.99x | 9.78 | 1.05x | 0.99x | 0.0553 | 1.08x | 1.00x | 2.705 | 0.98x | 0.96x |
| | 128 | 720.9M | 13.4 | 1.26x | 1.07x | 9.58 | 1.26x | 1.07x | 0.0547 | 1.32x | 1.09x | 2.573 | 1.09x | 1.03x |
| | 256 | 1441.8M | 12.4 | 1.61x | 1.17x | 10.31 | 1.61x | 1.17x | 0.0600 | 1.74x | 1.20x | 2.621 | 1.23x | 1.07x |
| | 512 | 2883.6M | 10.1 | 2.00x | 1.30x | 12.71 | 2.00x | 1.30x | 0.0772 | 2.16x | 1.35x | 2.832 | 1.42x | 1.14x |
| | | | | | | Llama-3.1-70B BS=64 | | | | | | | | |
| NFS-LoRA | 16 | 207.1M | 8.4 | 1.00x | 1.00x | 15.17 | 1.00x | 1.00x | 0.0821 | 1.00x | 1.00x | 4.652 | 1.00x | 1.00x |
| | 32 | 414.2M | 7.7 | 1.00x | 1.00x | 16.63 | 1.00x | 1.00x | 0.0926 | 1.00x | 1.00x | 4.768 | 1.00x | 1.00x |
| | 64 | 828.4M | 6.4 | 1.00x | 1.00x | 19.94 | 1.00x | 1.00x | 0.1164 | 1.00x | 1.00x | 5.040 | 1.00x | 1.00x |
| | 128 | 1656.8M | 4.4 | 1.00x | 1.00x | 28.98 | 1.00x | 1.00x | 0.1806 | 1.00x | 1.00x | 5.860 | 1.00x | 1.00x |
| | 256 | 3313.5M | 2.8 | 1.00x | 1.00x | 46.15 | 1.00x | 1.00x | 0.3034 | 1.00x | 1.00x | 7.307 | 1.00x | 1.00x |
| BD-LoRA | 32 | 180.2M | 8.4 | 0.99x | N/A | 15.32 | 0.99x | N/A | 0.0810 | 1.01x | N/A | 4.944 | 0.94x | N/A |
| | 64 | 360.4M | 8.2 | 1.07x | 0.97x | 15.56 | 1.07x | 0.97x | 0.0836 | 1.11x | 0.98x | 4.855 | 0.98x | 0.96x |
| | 128 | 720.9M | 8.4 | 1.31x | 1.09x | 15.19 | 1.31x | 1.09x | 0.0826 | 1.41x | 1.12x | 4.616 | 1.09x | 1.03x |
| | 256 | 1441.8M | 7.6 | 1.72x | 1.19x | 16.81 | 1.72x | 1.19x | 0.0939 | 1.92x | 1.24x | 4.785 | 1.22x | 1.05x |
| | 512 | 2883.6M | 5.9 | **2.13x** | **1.34x** | 21.65 | **2.13x** | **1.34x** | 0.1270 | **2.39x** | **1.42x** | 5.315 | 1.37x | 1.10x |

Table 12: Speedup of throughput, end-to-end (E2E) latency, decoding latency, and prefill latency of **Llama-3.1-70B** with an input token (**IT**) length of **1024**, an output token (**OT**) length of **128**, and batch sizes (**BS**) of **1, 16, 32, and 64**. **S.-0.87x** denotes the speedup when BD-LoRA has 0.87x the number of trainable parameters compared to S-LoRA or NFS-LoRA. **S.-1.74x** denotes the speedup when BD-LoRA has 1.74x the number of trainable parameters compared to S-LoRA or NFS-LoRA.

| Method | Rank | # Trainable Parameters | Throughput ↑ | | | E2E Latency ↓ | | | Decoding Latency ↓ | | | Prefill Latency ↓ | | |
|---|---|---|---|---|---|---|---|---|---|---|---|---|---|---|
| | | | Token/s | S.-0.87x | S.-1.74x | Time | S.-0.87x | S.-1.74x | Time | S.-0.87x | S.-1.74x | Time | S.-0.87x | S.-1.74x |
| | | | | | | Llama-3.1-70B BS=1 | | | | | | | | |
| S-LoRA | 16 | 207.1M | 31.8 | 1.00x | 1.00x | 4.02 | 1.00x | 1.00x | 0.0295 | 1.00x | 1.00x | 0.248 | 1.00x | 1.00x |
| | 32 | 414.2M | 31.3 | 1.00x | 1.00x | 4.09 | 1.00x | 1.00x | 0.0298 | 1.00x | 1.00x | 0.278 | 1.00x | 1.00x |
| | 64 | 828.4M | 30.8 | 1.00x | 1.00x | 4.15 | 1.00x | 1.00x | 0.0302 | 1.00x | 1.00x | 0.278 | 1.00x | 1.00x |
| | 128 | 1656.8M | 29.7 | 1.00x | 1.00x | 4.31 | 1.00x | 1.00x | 0.0314 | 1.00x | 1.00x | 0.281 | 1.00x | 1.00x |
| | 256 | 3313.5M | 27.5 | 1.00x | 1.00x | 4.66 | 1.00x | 1.00x | 0.0344 | 1.00x | 1.00x | 0.255 | 1.00x | 1.00x |
| BD-LoRA | 32 | 180.2M | 38.7 | 1.22x | N/A | 3.31 | 1.22x | N/A | 0.0240 | 1.23x | N/A | 0.228 | 1.09x | N/A |
| | 64 | 360.4M | 38.5 | 1.23x | 1.21x | 3.33 | 1.23x | 1.21x | 0.0242 | 1.23x | 1.22x | 0.232 | **1.20x** | 1.07x |
| | 128 | 720.9M | 38.5 | 1.25x | **1.23x** | 3.33 | 1.25x | **1.23x** | 0.0242 | 1.25x | **1.23x** | 0.232 | **1.20x** | 1.20x |
| | 256 | 1441.8M | 37.7 | 1.27x | 1.22x | 3.40 | 1.27x | 1.22x | 0.0247 | 1.27x | 1.22x | 0.234 | **1.20x** | 1.19x |
| | 512 | 2883.6M | 36.2 | 1.32x | 1.22x | 3.54 | 1.32x | 1.22x | 0.0259 | 1.33x | 1.21x | 0.222 | 1.15x | **1.26x** |
| | | | | | | Llama-3.1-70B BS=16 | | | | | | | | |
| S-LoRA | 16 | 207.1M | 17.5 | 1.00x | 1.00x | 7.30 | 1.00x | 1.00x | 0.0453 | 1.00x | 1.00x | 1.507 | 1.00x | 1.00x |
| | 32 | 414.2M | 17.1 | 1.00x | 1.00x | 7.50 | 1.00x | 1.00x | 0.0466 | 1.00x | 1.00x | 1.527 | 1.00x | 1.00x |
| | 64 | 828.4M | 16.2 | 1.00x | 1.00x | 7.92 | 1.00x | 1.00x | 0.0498 | 1.00x | 1.00x | 1.551 | 1.00x | 1.00x |
| | 128 | 1656.8M | 14.2 | 1.00x | 1.00x | 9.03 | 1.00x | 1.00x | 0.0579 | 1.00x | 1.00x | 1.621 | 1.00x | 1.00x |
| | 256 | 3313.5M | 11.1 | 1.00x | 1.00x | 11.58 | 1.00x | 1.00x | 0.0765 | 1.00x | 1.00x | 1.790 | 1.00x | 1.00x |
| BD-LoRA | 32 | 180.2M | 19.6 | 1.12x | N/A | 6.51 | 1.12x | N/A | 0.0390 | 1.16x | N/A | 1.525 | 0.99x | N/A |
| | 64 | 360.4M | 19.6 | 1.15x | 1.12x | 6.55 | 1.15x | 1.12x | 0.0394 | 1.18x | 1.15x | 1.502 | 1.02x | 1.00x |
| | 128 | 720.9M | 19.8 | 1.22x | 1.16x | 6.47 | 1.22x | 1.16x | 0.0393 | 1.27x | 1.19x | 1.436 | 1.08x | 1.06x |
| | 256 | 1441.8M | 18.7 | 1.32x | 1.16x | 6.85 | 1.32x | 1.16x | 0.0420 | 1.38x | 1.19x | 1.472 | 1.10x | 1.05x |
| | 512 | 2883.6M | 15.8 | 1.43x | 1.11x | 8.10 | 1.43x | 1.11x | 0.0509 | 1.50x | 1.14x | 1.578 | 1.13x | 1.03x |
| | | | | | | Llama-3.1-70B BS=32 | | | | | | | | |
| S-LoRA | 16 | 207.1M | 12.5 | 1.00x | 1.00x | 10.26 | 1.00x | 1.00x | 0.0594 | 1.00x | 1.00x | 2.651 | 1.00x | 1.00x |
| | 32 | 414.2M | 12.2 | 1.00x | 1.00x | 10.50 | 1.00x | 1.00x | 0.0616 | 1.00x | 1.00x | 2.617 | 1.00x | 1.00x |
| | 64 | 828.4M | 11.3 | 1.00x | 1.00x | 11.33 | 1.00x | 1.00x | 0.0677 | 1.00x | 1.00x | 2.658 | 1.00x | 1.00x |
| | 128 | 1656.8M | 9.5 | 1.00x | 1.00x | 13.53 | 1.00x | 1.00x | 0.0839 | 1.00x | 1.00x | 2.781 | 1.00x | 1.00x |
| | 256 | 3313.5M | 7.1 | 1.00x | 1.00x | 18.08 | 1.00x | 1.00x | 0.1172 | 1.00x | 1.00x | 3.084 | 1.00x | 1.00x |
| BD-LoRA | 32 | 180.2M | 13.3 | 1.06x | N/A | 9.65 | 1.06x | N/A | 0.0540 | 1.10x | N/A | 2.737 | 0.97x | N/A |
| | 64 | 360.4M | 13.1 | 1.07x | 1.05x | 9.78 | 1.07x | 1.05x | 0.0553 | 1.11x | 1.08x | 2.705 | 0.97x | 0.98x |
| | 128 | 720.9M | 13.4 | 1.18x | 1.10x | 9.58 | 1.18x | 1.10x | 0.0547 | 1.24x | 1.12x | 2.573 | 1.03x | 1.02x |
| | 256 | 1441.8M | 12.4 | 1.31x | 1.10x | 10.31 | 1.31x | 1.10x | 0.0600 | 1.40x | 1.13x | 2.621 | 1.06x | 1.01x |
| | 512 | 2883.6M | 10.1 | 1.42x | 1.06x | 12.71 | 1.42x | 1.06x | 0.0772 | 1.52x | 1.09x | 2.832 | 1.09x | 0.98x |
| | | | | | | Llama-3.1-70B BS=64 | | | | | | | | |
| S-LoRA | 16 | 207.1M | 8.1 | 1.00x | 1.00x | 15.87 | 1.00x | 1.00x | 0.0867 | 1.00x | 1.00x | 4.769 | 1.00x | 1.00x |
| | 32 | 414.2M | 7.8 | 1.00x | 1.00x | 16.50 | 1.00x | 1.00x | 0.0915 | 1.00x | 1.00x | 4.777 | 1.00x | 1.00x |
| | 64 | 828.4M | 7.1 | 1.00x | 1.00x | 18.06 | 1.00x | 1.00x | 0.1030 | 1.00x | 1.00x | 4.874 | 1.00x | 1.00x |
| | 128 | 1656.8M | 5.8 | 1.00x | 1.00x | 22.26 | 1.00x | 1.00x | 0.1337 | 1.00x | 1.00x | 5.138 | 1.00x | 1.00x |
| | 256 | 3313.5M | 4.1 | 1.00x | 1.00x | 31.34 | 1.00x | 1.00x | 0.2014 | 1.00x | 1.00x | 5.553 | 1.00x | 1.00x |
| BD-LoRA | 32 | 180.2M | 8.4 | 1.04x | N/A | 15.32 | 1.04x | N/A | 0.0810 | 1.07x | N/A | 4.944 | 0.96x | N/A |
| | 64 | 360.4M | 8.2 | 1.06x | 1.02x | 15.56 | 1.06x | 1.02x | 0.0836 | 1.09x | 1.04x | 4.855 | 0.98x | 0.98x |
| | 128 | 720.9M | 8.4 | 1.19x | 1.09x | 15.19 | 1.19x | 1.09x | 0.0826 | 1.25x | 1.11x | 4.616 | 1.06x | 1.03x |
| | 256 | 1441.8M | 7.6 | 1.32x | 1.07x | 16.81 | 1.32x | 1.07x | 0.0939 | 1.42x | 1.10x | 4.785 | 1.07x | 1.02x |
| | 512 | 2883.6M | 5.9 | **1.45x** | 1.03x | 21.65 | **1.45x** | 1.03x | 0.1270 | **1.59x** | 1.05x | 5.315 | 1.04x | 0.97x |

Table 13: Speedup of throughput, end-to-end (E2E) latency, decoding latency, and prefill latency of **Llama-3.1-8B** with an input token (**IT**) length of **4096**, an output token (**OT**) length of **256**, and batch sizes (**BS**) of **1, 16, 32, and 64**. **S.-0.86x** denotes the speedup when BD-LoRA has 0.86x the number of trainable parameters compared to S-LoRA or NFS-LoRA. **S.-1.73x** denotes the speedup when BD-LoRA has 1.73x the number of trainable parameters compared to S-LoRA or NFS-LoRA.

| Method | Rank | # Trainable Parameters | Throughput ↑ | | | E2E Latency ↓ | | | Decoding Latency ↓ | | | Prefill Latency ↓ | | |
|---|---|---|---|---|---|---|---|---|---|---|---|---|---|---|
| | | | Token/s | S.-0.86x | S.-1.73x | Time | S.-0.86x | S.-1.73x | Time | S.-0.86x | S.-1.73x | Time | S.-0.86x | S.-1.73x |
| | | | | | | Llama-3.1-8B BS=1 | | | | | | | | |
| NFS-LoRA | 16 | 41.9M | 130.9 | 1.00x | 1.00x | 1.96 | 1.00x | 1.00x | 0.0072 | 1.00x | 1.00x | 0.109 | 1.00x | 1.00x |
| | 32 | 83.9M | 129.7 | 1.00x | 1.00x | 1.97 | 1.00x | 1.00x | 0.0073 | 1.00x | 1.00x | 0.113 | 1.00x | 1.00x |
| | 64 | 167.8M | 121.2 | 1.00x | 1.00x | 2.11 | 1.00x | 1.00x | 0.0078 | 1.00x | 1.00x | 0.121 | 1.00x | 1.00x |
| | 128 | 335.5M | 112.2 | 1.00x | 1.00x | 2.28 | 1.00x | 1.00x | 0.0084 | 1.00x | 1.00x | 0.139 | 1.00x | 1.00x |
| | 256 | 671.1M | 94.5 | 1.00x | 1.00x | 2.71 | 1.00x | 1.00x | 0.0099 | 1.00x | 1.00x | 0.179 | 1.00x | 1.00x |
| BD-LoRA | 32 | 36.2M | 132.2 | 1.01x | N/A | 1.94 | 1.01x | N/A | 0.0071 | 1.01x | N/A | 0.116 | 0.94x | N/A |
| | 64 | 72.4M | 131.5 | 1.01x | 1.00x | 1.95 | 1.01x | 1.00x | 0.0071 | 1.02x | 1.01x | 0.117 | 0.97x | 0.93x |
| | 128 | 144.7M | 131.1 | 1.08x | 1.01x | 1.95 | 1.08x | 1.01x | 0.0072 | 1.08x | 1.01x | 0.110 | 1.10x | 1.03x |
| | 256 | 289.4M | 130.1 | 1.16x | 1.07x | 1.97 | 1.16x | 1.07x | 0.0072 | 1.15x | 1.07x | 0.111 | 1.25x | 1.09x |
| | 512 | 578.8M | 120.6 | 1.28x | 1.07x | 2.12 | 1.28x | 1.07x | 0.0078 | 1.26x | 1.07x | 0.122 | 1.46x | 1.14x |
| | | | | | | Llama-3.1-8B BS=16 | | | | | | | | |
| NFS-LoRA | 16 | 41.9M | 58.0 | 1.00x | 1.00x | 4.41 | 1.00x | 1.00x | 0.0139 | 1.00x | 1.00x | 0.862 | 1.00x | 1.00x |
| | 32 | 83.9M | 54.6 | 1.00x | 1.00x | 4.69 | 1.00x | 1.00x | 0.0149 | 1.00x | 1.00x | 0.883 | 1.00x | 1.00x |
| | 64 | 167.8M | 46.8 | 1.00x | 1.00x | 5.47 | 1.00x | 1.00x | 0.0176 | 1.00x | 1.00x | 0.961 | 1.00x | 1.00x |
| | 128 | 335.5M | 34.9 | 1.00x | 1.00x | 7.34 | 1.00x | 1.00x | 0.0243 | 1.00x | 1.00x | 1.117 | 1.00x | 1.00x |
| | 256 | 671.1M | 23.6 | 1.00x | 1.00x | 10.83 | 1.00x | 1.00x | 0.0367 | 1.00x | 1.00x | 1.440 | 1.00x | 1.00x |
| BD-LoRA | 32 | 36.2M | 57.9 | 1.00x | N/A | 4.42 | 1.00x | N/A | 0.0137 | 1.01x | N/A | 0.907 | 0.95x | N/A |
| | 64 | 72.4M | 57.3 | 1.05x | 0.99x | 4.47 | 1.05x | 0.99x | 0.0140 | 1.07x | 0.99x | 0.896 | 0.99x | 0.96x |
| | 128 | 144.7M | 58.0 | 1.24x | 1.06x | 4.42 | 1.24x | 1.06x | 0.0140 | 1.26x | 1.07x | 0.842 | 1.14x | 1.05x |
| | 256 | 289.4M | 54.1 | 1.55x | 1.16x | 4.74 | 1.55x | 1.16x | 0.0151 | 1.61x | 1.17x | 0.872 | 1.28x | 1.10x |
| | 512 | 578.8M | 41.6 | 1.76x | 1.19x | 6.15 | 1.76x | 1.19x | 0.0203 | 1.81x | 1.20x | 0.964 | 1.49x | 1.16x |
| | | | | | | Llama-3.1-8B BS=32 | | | | | | | | |
| NFS-LoRA | 16 | 41.9M | 38.6 | 1.00x | 1.00x | 6.63 | 1.00x | 1.00x | 0.0198 | 1.00x | 1.00x | 1.564 | 1.00x | 1.00x |
| | 32 | 83.9M | 35.3 | 1.00x | 1.00x | 7.25 | 1.00x | 1.00x | 0.0220 | 1.00x | 1.00x | 1.617 | 1.00x | 1.00x |
| | 64 | 167.8M | 29.6 | 1.00x | 1.00x | 8.65 | 1.00x | 1.00x | 0.0268 | 1.00x | 1.00x | 1.790 | 1.00x | 1.00x |
| | 128 | 335.5M | 21.0 | 1.00x | 1.00x | 12.21 | 1.00x | 1.00x | 0.0396 | 1.00x | 1.00x | 2.070 | 1.00x | 1.00x |
| | 256 | 671.1M | 13.5 | 1.00x | 1.00x | 18.99 | 1.00x | 1.00x | 0.0637 | 1.00x | 1.00x | 2.671 | 1.00x | 1.00x |
| BD-LoRA | 32 | 36.2M | 38.2 | 0.99x | N/A | 6.70 | 0.99x | N/A | 0.0197 | 1.01x | N/A | 1.667 | 0.94x | N/A |
| | 64 | 72.4M | 38.0 | 1.08x | 0.98x | 6.74 | 1.08x | 0.98x | 0.0201 | 1.10x | 0.99x | 1.596 | 1.01x | 0.98x |
| | 128 | 144.7M | 38.3 | 1.30x | 1.09x | 6.68 | 1.30x | 1.09x | 0.0201 | 1.34x | 1.10x | 1.544 | 1.16x | 1.05x |
| | 256 | 289.4M | 34.8 | 1.66x | 1.18x | 7.35 | 1.66x | 1.18x | 0.0224 | 1.77x | 1.20x | 1.613 | 1.28x | 1.11x |
| | 512 | 578.8M | 25.6 | 1.90x | 1.22x | 10.00 | 1.90x | 1.22x | 0.0320 | 1.99x | 1.24x | 1.796 | 1.49x | 1.15x |
| | | | | | | Llama-3.1-8B BS=64 | | | | | | | | |
| NFS-LoRA | 16 | 41.9M | 23.1 | 1.00x | 1.00x | 11.07 | 1.00x | 1.00x | 0.0322 | 1.00x | 1.00x | 2.810 | 1.00x | 1.00x |
| | 32 | 83.9M | 21.2 | 1.00x | 1.00x | 12.07 | 1.00x | 1.00x | 0.0358 | 1.00x | 1.00x | 2.887 | 1.00x | 1.00x |
| | 64 | 167.8M | 17.2 | 1.00x | 1.00x | 14.90 | 1.00x | 1.00x | 0.0459 | 1.00x | 1.00x | 3.157 | 1.00x | 1.00x |
| | 128 | 335.5M | 11.6 | 1.00x | 1.00x | 22.11 | 1.00x | 1.00x | 0.0717 | 1.00x | 1.00x | 3.749 | 1.00x | 1.00x |
| | 256 | 671.1M | 7.2 | 1.00x | 1.00x | 35.49 | 1.00x | 1.00x | 0.1190 | 1.00x | 1.00x | 5.025 | 1.00x | 1.00x |
| BD-LoRA | 32 | 36.2M | 23.0 | 1.00x | N/A | 11.13 | 1.00x | N/A | 0.0319 | 1.01x | N/A | 2.965 | 0.95x | N/A |
| | 64 | 72.4M | 22.5 | 1.06x | 0.97x | 11.36 | 1.06x | 0.97x | 0.0329 | 1.09x | 0.98x | 2.933 | 0.98x | 0.96x |
| | 128 | 144.7M | 23.2 | 1.35x | 1.09x | 11.06 | 1.35x | 1.09x | 0.0327 | 1.40x | 1.10x | 2.679 | 1.18x | 1.08x |
| | 256 | 289.4M | 20.6 | 1.78x | 1.20x | 12.45 | 1.78x | 1.20x | 0.0374 | 1.91x | 1.22x | 2.854 | 1.31x | 1.11x |
| | 512 | 578.8M | 14.5 | **2.01x** | **1.25x** | 17.65 | **2.01x** | **1.25x** | 0.0566 | **2.10x** | **1.27x** | 3.152 | **1.59x** | **1.19x** |

Table 14: Speedup of throughput, end-to-end (E2E) latency, decoding latency, and prefill latency of **Llama-3.1-8B** with an input token (**IT**) length of **4096**, an output token (**OT**) length of **256**, and batch sizes (**BS**) of **1, 16, 32, and 64**. **S.-0.86x** denotes the speedup when BD-LoRA has 0.86x the number of trainable parameters compared to S-LoRA or NFS-LoRA. **S.-1.73x** denotes the speedup when BD-LoRA has 1.73x the number of trainable parameters compared to S-LoRA or NFS-LoRA.

| Method | Rank | # Trainable Parameters | Throughput ↑ | | | E2E Latency ↓ | | | Decoding Latency ↓ | | | Prefill Latency ↓ | | |
|---|---|---|---|---|---|---|---|---|---|---|---|---|---|---|
| | | | Token/s | S.-0.86x | S.-1.73x | Time | S.-0.86x | S.-1.73x | Time | S.-0.86x | S.-1.73x | Time | S.-0.86x | S.-1.73x |
| | | | | | | Llama-3.1-8B BS=1 | | | | | | | | |
| | 16 | 41.9M | 104.6 | 1.00x | 1.00x | 2.45 | 1.00x | 1.00x | 0.0091 | 1.00x | 1.00x | 0.121 | 1.00x | 1.00x |
| | 32 | 83.9M | 103.8 | 1.00x | 1.00x | 2.47 | 1.00x | 1.00x | 0.0092 | 1.00x | 1.00x | 0.120 | 1.00x | 1.00x |
| **S-LoRA** | 64 | 167.8M | 102.0 | 1.00x | 1.00x | 2.51 | 1.00x | 1.00x | 0.0093 | 1.00x | 1.00x | 0.121 | 1.00x | 1.00x |
| | 128 | 335.5M | 97.2 | 1.00x | 1.00x | 2.63 | 1.00x | 1.00x | 0.0098 | 1.00x | 1.00x | 0.123 | 1.00x | 1.00x |
| | 256 | 671.1M | 88.2 | 1.00x | 1.00x | 2.90 | 1.00x | 1.00x | 0.0108 | 1.00x | 1.00x | 0.136 | 1.00x | 1.00x |
| | 32 | 36.2M | 132.2 | 1.26x | N/A | 1.94 | 1.26x | N/A | 0.0071 | 1.28x | N/A | 0.116 | 1.04x | N/A |
| | 64 | 72.4M | 131.5 | 1.27x | 1.26x | 1.95 | 1.27x | 1.26x | 0.0071 | 1.28x | 1.27x | 0.117 | 1.03x | 1.03x |
| **BD-LoRA** | 128 | 144.7M | 131.1 | 1.29x | 1.26x | 1.95 | 1.29x | 1.26x | 0.0072 | 1.30x | 1.27x | 0.110 | 1.11x | 1.10x |
| | 256 | 289.4M | 130.1 | 1.34x | **1.28x** | 1.97 | 1.34x | **1.28x** | 0.0072 | 1.35x | **1.29x** | 0.111 | 1.11x | 1.09x |
| | 512 | 578.8M | 120.6 | 1.37x | 1.24x | 2.12 | 1.37x | 1.24x | 0.0078 | 1.38x | 1.25x | 0.122 | 1.11x | 1.01x |
| | | | | | | Llama-3.1-8B BS=16 | | | | | | | | |
| | 16 | 41.9M | 51.0 | 1.00x | 1.00x | 5.02 | 1.00x | 1.00x | 0.0160 | 1.00x | 1.00x | 0.933 | 1.00x | 1.00x |
| | 32 | 83.9M | 50.1 | 1.00x | 1.00x | 5.11 | 1.00x | 1.00x | 0.0164 | 1.00x | 1.00x | 0.899 | 1.00x | 1.00x |
| **S-LoRA** | 64 | 167.8M | 46.2 | 1.00x | 1.00x | 5.54 | 1.00x | 1.00x | 0.0179 | 1.00x | 1.00x | 0.966 | 1.00x | 1.00x |
| | 128 | 335.5M | 40.2 | 1.00x | 1.00x | 6.36 | 1.00x | 1.00x | 0.0210 | 1.00x | 1.00x | 0.979 | 1.00x | 1.00x |
| | 256 | 671.1M | 30.4 | 1.00x | 1.00x | 8.41 | 1.00x | 1.00x | 0.0286 | 1.00x | 1.00x | 1.094 | 1.00x | 1.00x |
| | 32 | 36.2M | 57.9 | 1.14x | N/A | 4.42 | 1.14x | N/A | 0.0137 | 1.16x | N/A | 0.907 | 1.03x | N/A |
| | 64 | 72.4M | 57.3 | 1.14x | 1.12x | 4.47 | 1.14x | 1.12x | 0.0140 | 1.18x | 1.14x | 0.896 | 1.00x | 1.04x |
| **BD-LoRA** | 128 | 144.7M | 58.0 | 1.25x | 1.16x | 4.42 | 1.25x | 1.16x | 0.0140 | 1.28x | 1.18x | 0.842 | 1.15x | 1.07x |
| | 256 | 289.4M | 54.1 | 1.34x | 1.17x | 4.74 | 1.34x | 1.17x | 0.0151 | 1.39x | 1.18x | 0.872 | 1.12x | 1.11x |
| | 512 | 578.8M | 41.6 | 1.37x | 1.03x | 6.15 | 1.37x | 1.03x | 0.0203 | 1.41x | 1.04x | 0.964 | 1.13x | 1.02x |
| | | | | | | Llama-3.1-8B BS=32 | | | | | | | | |
| | 16 | 41.9M | 34.5 | 1.00x | 1.00x | 7.42 | 1.00x | 1.00x | 0.0223 | 1.00x | 1.00x | 1.714 | 1.00x | 1.00x |
| | 32 | 83.9M | 34.7 | 1.00x | 1.00x | 7.37 | 1.00x | 1.00x | 0.0227 | 1.00x | 1.00x | 1.562 | 1.00x | 1.00x |
| **S-LoRA** | 64 | 167.8M | 30.5 | 1.00x | 1.00x | 8.39 | 1.00x | 1.00x | 0.0258 | 1.00x | 1.00x | 1.789 | 1.00x | 1.00x |
| | 128 | 335.5M | 25.5 | 1.00x | 1.00x | 10.02 | 1.00x | 1.00x | 0.0321 | 1.00x | 1.00x | 1.800 | 1.00x | 1.00x |
| | 256 | 671.1M | 18.7 | 1.00x | 1.00x | 13.65 | 1.00x | 1.00x | 0.0454 | 1.00x | 1.00x | 2.028 | 1.00x | 1.00x |
| | 32 | 36.2M | 38.2 | 1.11x | N/A | 6.70 | 1.11x | N/A | 0.0197 | 1.13x | N/A | 1.667 | 1.03x | N/A |
| | 64 | 72.4M | 38.0 | 1.09x | 1.10x | 6.74 | 1.09x | 1.10x | 0.0201 | 1.13x | 1.11x | 1.596 | 0.98x | 1.07x |
| **BD-LoRA** | 128 | 144.7M | 38.3 | 1.26x | 1.10x | 6.68 | 1.26x | 1.10x | 0.0201 | 1.29x | 1.13x | 1.544 | 1.16x | 1.01x |
| | 256 | 289.4M | 34.8 | 1.36x | 1.14x | 7.35 | 1.36x | 1.14x | 0.0224 | 1.43x | 1.15x | 1.613 | 1.12x | 1.11x |
| | 512 | 578.8M | 25.6 | 1.37x | 1.00x | 10.00 | 1.37x | 1.00x | 0.0320 | 1.42x | 1.00x | 1.796 | 1.13x | 1.00x |
| | | | | | | Llama-3.1-8B BS=64 | | | | | | | | |
| | 16 | 41.9M | 21.1 | 1.00x | 1.00x | 12.13 | 1.00x | 1.00x | 0.0351 | 1.00x | 1.00x | 3.149 | 1.00x | 1.00x |
| | 32 | 83.9M | 20.2 | 1.00x | 1.00x | 12.65 | 1.00x | 1.00x | 0.0371 | 1.00x | 1.00x | 3.139 | 1.00x | 1.00x |
| **S-LoRA** | 64 | 167.8M | 18.4 | 1.00x | 1.00x | 13.94 | 1.00x | 1.00x | 0.0419 | 1.00x | 1.00x | 3.213 | 1.00x | 1.00x |
| | 128 | 335.5M | 15.1 | 1.00x | 1.00x | 16.98 | 1.00x | 1.00x | 0.0539 | 1.00x | 1.00x | 3.185 | 1.00x | 1.00x |
| | 256 | 671.1M | 10.5 | 1.00x | 1.00x | 24.43 | 1.00x | 1.00x | 0.0810 | 1.00x | 1.00x | 3.688 | 1.00x | 1.00x |
| | 32 | 36.2M | 23.0 | 1.09x | N/A | 11.13 | 1.09x | N/A | 0.0319 | 1.10x | N/A | 2.965 | 1.06x | N/A |
| | 64 | 72.4M | 22.5 | 1.11x | 1.07x | 11.36 | 1.11x | 1.07x | 0.0329 | 1.13x | 1.07x | 2.933 | 1.07x | 1.07x |
| **BD-LoRA** | 128 | 144.7M | 23.2 | 1.26x | 1.14x | 11.06 | 1.26x | 1.14x | 0.0327 | 1.28x | 1.14x | 2.679 | **1.20x** | **1.17x** |
| | 256 | 289.4M | 20.6 | 1.36x | 1.12x | 12.45 | 1.36x | 1.12x | 0.0374 | **1.44x** | 1.12x | 2.854 | 1.12x | 1.13x |
| | 512 | 578.8M | 14.5 | **1.38x** | 0.96x | 17.65 | **1.38x** | 0.96x | 0.0566 | 1.43x | 0.95x | 3.152 | 1.17x | 1.01x |

Table 15: Speedup of throughput, end-to-end (E2E) latency, decoding latency, and prefill latency of **Llama-3.1-70B** with an input token (**IT**) length of **4096**, an output token (**OT**) length of **256**, and batch sizes (**BS**) of **1, 16, 32, and 64**. **S.-0.87x** denotes the speedup when BD-LoRA has 0.87x the number of trainable parameters compared to S-LoRA or NFS-LoRA. **S.-1.74x** denotes the speedup when BD-LoRA has 1.74x the number of trainable parameters compared to S-LoRA or NFS-LoRA.

| Method | Rank | # Trainable Parameters | Throughput ↑ | | | E2E Latency ↓ | | | Decoding Latency ↓ | | | Prefill Latency ↓ | | |
|---|---|---|---|---|---|---|---|---|---|---|---|---|---|---|
| | | | Token/s | S.-0.87x | S.-1.74x | Time | S.-0.87x | S.-1.74x | Time | S.-0.87x | S.-1.74x | Time | S.-0.87x | S.-1.74x |
| | | | | | | Llama-3.1-70B BS=1 | | | | | | | | |
| NFS-LoRA | 16 | 207.1M | 37.1 | 1.00x | 1.00x | 6.91 | 1.00x | 1.00x | 0.0250 | 1.00x | 1.00x | 0.518 | 1.00x | 1.00x |
| | 32 | 414.2M | 36.3 | 1.00x | 1.00x | 7.06 | 1.00x | 1.00x | 0.0255 | 1.00x | 1.00x | 0.534 | 1.00x | 1.00x |
| | 64 | 828.4M | 34.8 | 1.00x | 1.00x | 7.35 | 1.00x | 1.00x | 0.0265 | 1.00x | 1.00x | 0.570 | 1.00x | 1.00x |
| | 128 | 1656.8M | 32.6 | 1.00x | 1.00x | 7.86 | 1.00x | 1.00x | 0.0282 | 1.00x | 1.00x | 0.653 | 1.00x | 1.00x |
| | 256 | 3313.5M | 27.7 | 1.00x | 1.00x | 9.26 | 1.00x | 1.00x | 0.0329 | 1.00x | 1.00x | 0.822 | 1.00x | 1.00x |
| BD-LoRA | 32 | 180.2M | 37.5 | 1.01x | N/A | 6.83 | 1.01x | N/A | 0.0245 | 1.02x | N/A | 0.551 | 0.94x | N/A |
| | 64 | 360.4M | 37.2 | 1.03x | 1.00x | 6.88 | 1.03x | 1.00x | 0.0248 | 1.03x | 1.01x | 0.542 | 0.98x | 0.96x |
| | 128 | 720.9M | 37.3 | 1.07x | 1.03x | 6.86 | 1.07x | 1.03x | 0.0248 | 1.07x | 1.03x | 0.517 | 1.10x | 1.03x |
| | 256 | 1441.8M | 36.4 | 1.12x | 1.05x | 7.02 | 1.12x | 1.05x | 0.0254 | 1.11x | 1.04x | 0.534 | 1.22x | 1.07x |
| | 512 | 2883.6M | 34.7 | 1.26x | 1.07x | 7.37 | 1.26x | 1.07x | 0.0266 | 1.24x | 1.06x | 0.571 | 1.44x | 1.14x |
| | | | | | | Llama-3.1-70B BS=16 | | | | | | | | |
| NFS-LoRA | 16 | 207.1M | 15.3 | 1.00x | 1.00x | 16.71 | 1.00x | 1.00x | 0.0479 | 1.00x | 1.00x | 4.447 | 1.00x | 1.00x |
| | 32 | 414.2M | 14.6 | 1.00x | 1.00x | 17.59 | 1.00x | 1.00x | 0.0508 | 1.00x | 1.00x | 4.598 | 1.00x | 1.00x |
| | 64 | 828.4M | 13.0 | 1.00x | 1.00x | 19.69 | 1.00x | 1.00x | 0.0577 | 1.00x | 1.00x | 4.923 | 1.00x | 1.00x |
| | 128 | 1656.8M | 10.3 | 1.00x | 1.00x | 24.93 | 1.00x | 1.00x | 0.0752 | 1.00x | 1.00x | 5.673 | 1.00x | 1.00x |
| | 256 | 3313.5M | 7.2 | 1.00x | 1.00x | 35.61 | 1.00x | 1.00x | 0.1110 | 1.00x | 1.00x | 7.194 | 1.00x | 1.00x |
| BD-LoRA | 32 | 180.2M | 15.1 | 0.98x | N/A | 16.97 | 0.98x | N/A | 0.0478 | 1.00x | N/A | 4.726 | 0.94x | N/A |
| | 64 | 360.4M | 15.0 | 1.03x | 0.98x | 17.08 | 1.03x | 0.98x | 0.0485 | 1.05x | 0.99x | 4.664 | 0.99x | 0.95x |
| | 128 | 720.9M | 15.4 | 1.18x | 1.05x | 16.68 | 1.18x | 1.05x | 0.0479 | 1.20x | 1.06x | 4.413 | 1.12x | 1.04x |
| | 256 | 1441.8M | 14.5 | 1.42x | 1.12x | 17.61 | 1.42x | 1.12x | 0.0509 | 1.48x | 1.13x | 4.570 | 1.24x | 1.08x |
| | 512 | 2883.6M | 12.6 | 1.75x | 1.23x | 20.33 | 1.75x | 1.23x | 0.0602 | 1.84x | 1.25x | 4.917 | **1.46x** | **1.15x** |
| | | | | | | Llama-3.1-70B BS=32 | | | | | | | | |
| NFS-LoRA | 16 | 207.1M | 9.7 | 1.00x | 1.00x | 26.51 | 1.00x | 1.00x | 0.0709 | 1.00x | 1.00x | 8.342 | 1.00x | 1.00x |
| | 32 | 414.2M | 9.1 | 1.00x | 1.00x | 28.16 | 1.00x | 1.00x | 0.0763 | 1.00x | 1.00x | 8.617 | 1.00x | 1.00x |
| | 64 | 828.4M | 7.9 | 1.00x | 1.00x | 32.27 | 1.00x | 1.00x | 0.0900 | 1.00x | 1.00x | 9.237 | 1.00x | 1.00x |
| | 128 | 1656.8M | 6.0 | 1.00x | 1.00x | 42.43 | 1.00x | 1.00x | 0.1243 | 1.00x | 1.00x | 10.612 | 1.00x | 1.00x |
| | 256 | 3313.5M | 4.1 | 1.00x | 1.00x | 62.80 | 1.00x | 1.00x | 0.1925 | 1.00x | 1.00x | 13.508 | 1.00x | 1.00x |
| BD-LoRA | 32 | 180.2M | 9.4 | 0.98x | N/A | 27.18 | 0.98x | N/A | 0.0714 | 0.99x | N/A | 8.896 | 0.94x | N/A |
| | 64 | 360.4M | 9.4 | 1.03x | 0.97x | 27.32 | 1.03x | 0.97x | 0.0725 | 1.05x | 0.98x | 8.743 | 0.99x | 0.95x |
| | 128 | 720.9M | 9.7 | 1.22x | 1.06x | 26.49 | 1.22x | 1.06x | 0.0710 | 1.27x | 1.07x | 8.304 | 1.11x | 1.04x |
| | 256 | 1441.8M | 9.1 | 1.50x | 1.14x | 28.23 | 1.50x | 1.14x | 0.0768 | 1.62x | 1.17x | 8.557 | 1.24x | 1.08x |
| | 512 | 2883.6M | 7.6 | 1.86x | 1.26x | 33.70 | 1.86x | 1.26x | 0.0955 | 2.02x | 1.30x | 9.251 | **1.46x** | **1.15x** |
| | | | | | | Llama-3.1-70B BS=64 | | | | | | | | |
| NFS-LoRA | 16 | 207.1M | 5.6 | 1.00x | 1.00x | 45.44 | 1.00x | 1.00x | 0.1153 | 1.00x | 1.00x | 15.918 | 1.00x | 1.00x |
| | 32 | 414.2M | 5.3 | 1.00x | 1.00x | 48.75 | 1.00x | 1.00x | 0.1262 | 1.00x | 1.00x | 16.448 | 1.00x | 1.00x |
| | 64 | 828.4M | 4.5 | 1.00x | 1.00x | 56.62 | 1.00x | 1.00x | 0.1521 | 1.00x | 1.00x | 17.665 | 1.00x | 1.00x |
| | 128 | 1656.8M | 3.3 | 1.00x | 1.00x | 77.14 | 1.00x | 1.00x | 0.2219 | 1.00x | 1.00x | 20.313 | 1.00x | 1.00x |
| | 256 | 3313.5M | 2.2 | 1.00x | 1.00x | 116.68 | 1.00x | 1.00x | 0.3547 | 1.00x | 1.00x | 25.857 | 1.00x | 1.00x |
| BD-LoRA | 32 | 180.2M | 5.5 | 0.97x | N/A | 46.74 | 0.97x | N/A | 0.1161 | 0.99x | N/A | 17.003 | 0.94x | N/A |
| | 64 | 360.4M | 5.5 | 1.04x | 0.97x | 46.92 | 1.04x | 0.97x | 0.1179 | 1.07x | 0.98x | 16.720 | 0.98x | 0.95x |
| | 128 | 720.9M | 5.6 | 1.25x | 1.08x | 45.34 | 1.25x | 1.08x | 0.1153 | 1.32x | 1.09x | 15.815 | 1.12x | 1.04x |
| | 256 | 1441.8M | 5.2 | 1.58x | 1.16x | 48.82 | 1.58x | 1.16x | 0.1269 | 1.75x | 1.20x | 16.335 | 1.24x | 1.08x |
| | 512 | 2883.6M | 4.3 | **1.96x** | **1.30x** | 59.54 | **1.96x** | **1.30x** | 0.1629 | **2.18x** | **1.36x** | 17.831 | 1.45x | 1.14x |

Table 16: Speedup of throughput, end-to-end (E2E) latency, decoding latency, and prefill latency of **Llama-3.1-70B** with an input token (**IT**) length of **4096**, an output token (**OT**) length of **256**, and batch sizes (**BS**) of **1, 16, 32, and 64**. **S.-0.87x** denotes the speedup when BD-LoRA has 0.87x the number of trainable parameters compared to S-LoRA or NFS-LoRA. **S.-1.74x** denotes the speedup when BD-LoRA has 1.74x the number of trainable parameters compared to S-LoRA or NFS-LoRA.

| Method | Rank | # Trainable Parameters | Throughput ↑ | | | E2E Latency ↓ | | | Decoding Latency ↓ | | | Prefill Latency ↓ | | |
|---|---|---|---|---|---|---|---|---|---|---|---|---|---|---|
| | | | Token/s | S.-0.87x | S.-1.74x | Time | S.-0.87x | S.-1.74x | Time | S.-0.87x | S.-1.74x | Time | S.-0.87x | S.-1.74x |
| | | | | | | Llama-3.1-70B BS=1 | | | | | | | | |
| | 16 | 207.1M | 31.0 | 1.00x | 1.00x | 8.25 | 1.00x | 1.00x | 0.0302 | 1.00x | 1.00x | 0.511 | 1.00x | 1.00x |
| | 32 | 414.2M | 30.8 | 1.00x | 1.00x | 8.32 | 1.00x | 1.00x | 0.0305 | 1.00x | 1.00x | 0.513 | 1.00x | 1.00x |
| **S-LoRA** | 64 | 828.4M | 30.2 | 1.00x | 1.00x | 8.46 | 1.00x | 1.00x | 0.0310 | 1.00x | 1.00x | 0.527 | 1.00x | 1.00x |
| | 128 | 1656.8M | 29.1 | 1.00x | 1.00x | 8.79 | 1.00x | 1.00x | 0.0322 | 1.00x | 1.00x | 0.555 | 1.00x | 1.00x |
| | 256 | 3313.5M | 26.6 | 1.00x | 1.00x | 9.62 | 1.00x | 1.00x | 0.0352 | 1.00x | 1.00x | 0.615 | 1.00x | 1.00x |
| | 32 | 180.2M | 37.5 | 1.21x | N/A | 6.83 | 1.21x | N/A | 0.0245 | 1.23x | N/A | 0.551 | 0.93x | N/A |
| | 64 | 360.4M | 37.2 | 1.21x | 1.20x | 6.88 | 1.21x | 1.20x | 0.0248 | 1.23x | 1.22x | 0.542 | 0.95x | 0.94x |
| **BD-LoRA** | 128 | 720.9M | 37.3 | 1.23x | **1.21x** | 6.86 | 1.23x | **1.21x** | 0.0248 | 1.25x | **1.23x** | 0.517 | 1.02x | 0.99x |
| | 256 | 1441.8M | 36.4 | 1.25x | 1.20x | 7.02 | 1.25x | 1.20x | 0.0254 | 1.27x | 1.22x | 0.534 | 1.04x | 0.99x |
| | 512 | 2883.6M | 34.7 | 1.30x | 1.19x | 7.37 | 1.30x | 1.19x | 0.0266 | 1.32x | 1.21x | 0.571 | 1.08x | 0.97x |
| | | | | | | Llama-3.1-70B BS=16 | | | | | | | | |
| | 16 | 207.1M | 14.1 | 1.00x | 1.00x | 18.13 | 1.00x | 1.00x | 0.0536 | 1.00x | 1.00x | 4.407 | 1.00x | 1.00x |
| | 32 | 414.2M | 13.8 | 1.00x | 1.00x | 18.54 | 1.00x | 1.00x | 0.0550 | 1.00x | 1.00x | 4.458 | 1.00x | 1.00x |
| **S-LoRA** | 64 | 828.4M | 13.1 | 1.00x | 1.00x | 19.53 | 1.00x | 1.00x | 0.0584 | 1.00x | 1.00x | 4.581 | 1.00x | 1.00x |
| | 128 | 1656.8M | 11.7 | 1.00x | 1.00x | 21.95 | 1.00x | 1.00x | 0.0669 | 1.00x | 1.00x | 4.830 | 1.00x | 1.00x |
| | 256 | 3313.5M | 9.3 | 1.00x | 1.00x | 27.43 | 1.00x | 1.00x | 0.0862 | 1.00x | 1.00x | 5.358 | 1.00x | 1.00x |
| | 32 | 180.2M | 15.1 | 1.07x | N/A | 16.97 | 1.07x | N/A | 0.0478 | 1.12x | N/A | 4.726 | 0.93x | N/A |
| | 64 | 360.4M | 15.0 | 1.09x | 1.06x | 17.08 | 1.09x | 1.06x | 0.0485 | 1.13x | 1.11x | 4.664 | 0.96x | 0.94x |
| **BD-LoRA** | 128 | 720.9M | 15.4 | 1.17x | 1.11x | 16.68 | 1.17x | 1.11x | 0.0479 | 1.22x | 1.15x | 4.413 | 1.04x | **1.01x** |
| | 256 | 1441.8M | 14.5 | 1.25x | 1.11x | 17.61 | 1.25x | 1.11x | 0.0509 | 1.31x | 1.15x | 4.570 | 1.06x | 1.00x |
| | 512 | 2883.6M | 12.6 | **1.35x** | 1.08x | 20.33 | **1.35x** | 1.08x | 0.0602 | 1.43x | 1.11x | 4.917 | **1.09x** | 0.98x |
| | | | | | | Llama-3.1-70B BS=32 | | | | | | | | |
| | 16 | 207.1M | 9.4 | 1.00x | 1.00x | 27.26 | 1.00x | 1.00x | 0.0748 | 1.00x | 1.00x | 8.113 | 1.00x | 1.00x |
| | 32 | 414.2M | 9.1 | 1.00x | 1.00x | 28.01 | 1.00x | 1.00x | 0.0775 | 1.00x | 1.00x | 8.171 | 1.00x | 1.00x |
| **S-LoRA** | 64 | 828.4M | 8.5 | 1.00x | 1.00x | 30.01 | 1.00x | 1.00x | 0.0841 | 1.00x | 1.00x | 8.483 | 1.00x | 1.00x |
| | 128 | 1656.8M | 7.4 | 1.00x | 1.00x | 34.73 | 1.00x | 1.00x | 0.1010 | 1.00x | 1.00x | 8.864 | 1.00x | 1.00x |
| | 256 | 3313.5M | 5.7 | 1.00x | 1.00x | 44.66 | 1.00x | 1.00x | 0.1358 | 1.00x | 1.00x | 9.887 | 1.00x | 1.00x |
| | 32 | 180.2M | 9.4 | 1.00x | N/A | 27.18 | 1.00x | N/A | 0.0714 | 1.05x | N/A | 8.896 | 0.91x | N/A |
| | 64 | 360.4M | 9.4 | 1.03x | 1.00x | 27.32 | 1.03x | 1.00x | 0.0725 | 1.07x | 1.03x | 8.743 | 0.93x | 0.93x |
| **BD-LoRA** | 128 | 720.9M | 9.7 | 1.13x | 1.06x | 26.49 | 1.13x | 1.06x | 0.0710 | 1.18x | 1.09x | 8.304 | 1.02x | 0.98x |
| | 256 | 1441.8M | 9.1 | 1.23x | 1.06x | 28.23 | 1.23x | 1.06x | 0.0768 | 1.32x | 1.09x | 8.557 | 1.04x | 0.99x |
| | 512 | 2883.6M | 7.6 | 1.33x | 1.03x | 33.70 | 1.33x | 1.03x | 0.0955 | 1.42x | 1.06x | 9.251 | 1.07x | 0.96x |
| | | | | | | Llama-3.1-70B BS=64 | | | | | | | | |
| | 16 | 207.1M | 5.7 | 1.00x | 1.00x | 45.29 | 1.00x | 1.00x | 0.1167 | 1.00x | 1.00x | 15.411 | 1.00x | 1.00x |
| | 32 | 414.2M | 5.5 | 1.00x | 1.00x | 46.86 | 1.00x | 1.00x | 0.1222 | 1.00x | 1.00x | 15.570 | 1.00x | 1.00x |
| **S-LoRA** | 64 | 828.4M | 5.0 | 1.00x | 1.00x | 50.91 | 1.00x | 1.00x | 0.1360 | 1.00x | 1.00x | 16.094 | 1.00x | 1.00x |
| | 128 | 1656.8M | 4.3 | 1.00x | 1.00x | 59.73 | 1.00x | 1.00x | 0.1671 | 1.00x | 1.00x | 16.935 | 1.00x | 1.00x |
| | 256 | 3313.5M | 3.2 | 1.00x | 1.00x | 79.87 | 1.00x | 1.00x | 0.2374 | 1.00x | 1.00x | 19.083 | 1.00x | 1.00x |
| | 32 | 180.2M | 5.5 | 0.97x | N/A | 46.74 | 0.97x | N/A | 0.1161 | 1.00x | N/A | 17.003 | 0.91x | N/A |
| | 64 | 360.4M | 5.5 | 1.00x | 0.97x | 46.92 | 1.00x | 0.97x | 0.1179 | 1.04x | 0.99x | 16.720 | 0.93x | 0.92x |
| **BD-LoRA** | 128 | 720.9M | 5.6 | 1.12x | 1.03x | 45.34 | 1.12x | 1.03x | 0.1153 | 1.18x | 1.06x | 15.815 | 1.02x | 0.98x |
| | 256 | 1441.8M | 5.2 | 1.22x | 1.04x | 48.82 | 1.22x | 1.04x | 0.1269 | 1.32x | 1.07x | 16.335 | 1.04x | 0.99x |
| | 512 | 2883.6M | 4.3 | 1.34x | 1.00x | 59.54 | 1.34x | 1.00x | 0.1629 | **1.46x** | 1.03x | 17.831 | 1.07x | 0.95x |

Table 17: Speedup of throughput, end-to-end (E2E) latency, decoding latency, and prefill latency of **Llama-3.1-8B** with an input token (**IT**) length of **128**, an output token (**OT**) length of **1024**, and batch sizes (**BS**) of **1, 16, 32, and 64**. **S.-0.86x** denotes the speedup when BD-LoRA has 0.86x the number of trainable parameters compared to S-LoRA or NFS-LoRA. **S.-1.73x** denotes the speedup when BD-LoRA has 1.73x the number of trainable parameters compared to S-LoRA or NFS-LoRA.

| Method | Rank | # Trainable Parameters | Throughput ↑ | | | E2E Latency ↓ | | | Decoding Latency ↓ | | | Prefill Latency ↓ | | |
|---|---|---|---|---|---|---|---|---|---|---|---|---|---|---|
| | | | Token/s | S.-0.86x | S.-1.73x | Time | S.-0.86x | S.-1.73x | Time | S.-0.86x | S.-1.73x | Time | S.-0.86x | S.-1.73x |
| | | | | | | Llama-3.1-8B BS=1 | | | | | | | | |
| | 16 | 41.9M | 141.7 | 1.00x | 1.00x | 7.23 | 1.00x | 1.00x | 0.0070 | 1.00x | 1.00x | 0.088 | 1.00x | 1.00x |
| | 32 | 83.9M | 137.4 | 1.00x | 1.00x | 7.45 | 1.00x | 1.00x | 0.0072 | 1.00x | 1.00x | 0.095 | 1.00x | 1.00x |
| NFS-LoRA | 64 | 167.8M | 131.0 | 1.00x | 1.00x | 7.82 | 1.00x | 1.00x | 0.0075 | 1.00x | 1.00x | 0.095 | 1.00x | 1.00x |
| | 128 | 335.5M | 122.5 | 1.00x | 1.00x | 8.36 | 1.00x | 1.00x | 0.0081 | 1.00x | 1.00x | 0.088 | 1.00x | 1.00x |
| | 256 | 671.1M | 101.4 | 1.00x | 1.00x | 10.10 | 1.00x | 1.00x | 0.0098 | 1.00x | 1.00x | 0.096 | 1.00x | 1.00x |
| | 32 | 36.2M | 143.6 | 1.01x | N/A | 7.13 | 1.01x | N/A | 0.0069 | 1.01x | N/A | 0.088 | 1.00x | N/A |
| | 64 | 72.4M | 142.7 | 1.04x | 1.01x | 7.18 | 1.04x | 1.01x | 0.0069 | 1.04x | 1.01x | 0.086 | 1.11x | 1.02x |
| BD-LoRA | 128 | 144.7M | 141.8 | 1.08x | 1.03x | 7.22 | 1.08x | 1.03x | 0.0070 | 1.08x | 1.03x | 0.088 | 1.09x | 1.09x |
| | 256 | 289.4M | 137.5 | 1.12x | 1.05x | 7.45 | 1.12x | 1.05x | 0.0072 | 1.12x | 1.05x | 0.093 | 0.95x | 1.02x |
| | 512 | 578.8M | 131.2 | 1.29x | 1.07x | 7.81 | 1.29x | 1.07x | 0.0075 | 1.30x | 1.07x | 0.087 | 1.11x | 1.02x |
| | | | | | | Llama-3.1-8B BS=16 | | | | | | | | |
| | 16 | 41.9M | 88.6 | 1.00x | 1.00x | 11.57 | 1.00x | 1.00x | 0.0111 | 1.00x | 1.00x | 0.248 | 1.00x | 1.00x |
| | 32 | 83.9M | 83.5 | 1.00x | 1.00x | 12.27 | 1.00x | 1.00x | 0.0117 | 1.00x | 1.00x | 0.248 | 1.00x | 1.00x |
| NFS-LoRA | 64 | 167.8M | 70.3 | 1.00x | 1.00x | 14.56 | 1.00x | 1.00x | 0.0140 | 1.00x | 1.00x | 0.252 | 1.00x | 1.00x |
| | 128 | 335.5M | 49.3 | 1.00x | 1.00x | 20.77 | 1.00x | 1.00x | 0.0201 | 1.00x | 1.00x | 0.233 | 1.00x | 1.00x |
| | 256 | 671.1M | 31.5 | 1.00x | 1.00x | 32.54 | 1.00x | 1.00x | 0.0315 | 1.00x | 1.00x | 0.289 | 1.00x | 1.00x |
| | 32 | 36.2M | 90.4 | 1.02x | N/A | 11.32 | 1.02x | N/A | 0.0108 | 1.02x | N/A | 0.261 | 0.95x | N/A |
| | 64 | 72.4M | 89.0 | 1.07x | 1.01x | 11.50 | 1.07x | 1.01x | 0.0110 | 1.07x | 1.00x | 0.232 | 1.07x | 1.07x |
| BD-LoRA | 128 | 144.7M | 88.3 | 1.26x | 1.06x | 11.59 | 1.26x | 1.06x | 0.0111 | 1.26x | 1.06x | 0.236 | 1.06x | 1.05x |
| | 256 | 289.4M | 81.4 | 1.65x | 1.16x | 12.58 | 1.65x | 1.16x | 0.0120 | 1.66x | 1.16x | 0.245 | 0.95x | 1.03x |
| | 512 | 578.8M | 59.0 | 1.87x | 1.20x | 17.36 | 1.87x | 1.20x | 0.0167 | 1.89x | 1.20x | 0.256 | 1.13x | 0.91x |
| | | | | | | Llama-3.1-8B BS=32 | | | | | | | | |
| | 16 | 41.9M | 73.9 | 1.00x | 1.00x | 13.85 | 1.00x | 1.00x | 0.0132 | 1.00x | 1.00x | 0.291 | 1.00x | 1.00x |
| | 32 | 83.9M | 65.4 | 1.00x | 1.00x | 15.65 | 1.00x | 1.00x | 0.0150 | 1.00x | 1.00x | 0.292 | 1.00x | 1.00x |
| NFS-LoRA | 64 | 167.8M | 50.4 | 1.00x | 1.00x | 20.31 | 1.00x | 1.00x | 0.0195 | 1.00x | 1.00x | 0.285 | 1.00x | 1.00x |
| | 128 | 335.5M | 31.7 | 1.00x | 1.00x | 32.30 | 1.00x | 1.00x | 0.0313 | 1.00x | 1.00x | 0.279 | 1.00x | 1.00x |
| | 256 | 671.1M | 18.8 | 1.00x | 1.00x | 54.46 | 1.00x | 1.00x | 0.0529 | 1.00x | 1.00x | 0.300 | 1.00x | 1.00x |
| | 32 | 36.2M | 77.7 | 1.05x | N/A | 13.19 | 1.05x | N/A | 0.0126 | 1.05x | N/A | 0.264 | 1.10x | N/A |
| | 64 | 72.4M | 73.7 | 1.13x | 1.00x | 13.89 | 1.13x | 1.00x | 0.0133 | 1.13x | 0.99x | 0.258 | 1.13x | 1.13x |
| BD-LoRA | 128 | 144.7M | 72.6 | 1.44x | 1.11x | 14.10 | 1.44x | 1.11x | 0.0135 | 1.45x | 1.11x | 0.277 | 1.03x | 1.06x |
| | 256 | 289.4M | 63.0 | 1.99x | 1.25x | 16.27 | 1.99x | 1.25x | 0.0156 | 2.00x | 1.25x | 0.277 | 1.01x | 1.03x |
| | 512 | 578.8M | 40.0 | 2.13x | 1.26x | 25.58 | 2.13x | 1.26x | 0.0247 | 2.14x | 1.27x | 0.276 | 1.09x | 1.01x |
| | | | | | | Llama-3.1-8B BS=64 | | | | | | | | |
| | 16 | 41.9M | 52.4 | 1.00x | 1.00x | 19.53 | 1.00x | 1.00x | 0.0188 | 1.00x | 1.00x | 0.306 | 1.00x | 1.00x |
| | 32 | 83.9M | 44.7 | 1.00x | 1.00x | 22.90 | 1.00x | 1.00x | 0.0221 | 1.00x | 1.00x | 0.301 | 1.00x | 1.00x |
| NFS-LoRA | 64 | 167.8M | 31.9 | 1.00x | 1.00x | 32.07 | 1.00x | 1.00x | 0.0309 | 1.00x | 1.00x | 0.397 | 1.00x | 1.00x |
| | 128 | 335.5M | 18.3 | 1.00x | 1.00x | 55.82 | 1.00x | 1.00x | 0.0542 | 1.00x | 1.00x | 0.329 | 1.00x | 1.00x |
| | 256 | 671.1M | 10.3 | 1.00x | 1.00x | 99.83 | 1.00x | 1.00x | 0.0971 | 1.00x | 1.00x | 0.345 | 1.00x | 1.00x |
| | 32 | 36.2M | 56.1 | 1.07x | N/A | 18.27 | 1.07x | N/A | 0.0175 | 1.07x | N/A | 0.316 | 0.97x | N/A |
| | 64 | 72.4M | 51.9 | 1.16x | 0.99x | 19.74 | 1.16x | 0.99x | 0.0190 | 1.16x | 0.99x | 0.321 | 0.94x | 0.95x |
| BD-LoRA | 128 | 144.7M | 50.6 | 1.59x | 1.13x | 20.23 | 1.59x | 1.13x | 0.0194 | 1.59x | 1.13x | 0.315 | **1.26x** | 0.96x |
| | 256 | 289.4M | 41.5 | 2.26x | 1.30x | 24.68 | 2.26x | 1.30x | 0.0238 | 2.28x | 1.30x | 0.336 | 0.98x | **1.18x** |
| | 512 | 578.8M | 24.0 | **2.34x** | **1.31x** | 42.66 | **2.34x** | **1.31x** | 0.0413 | **2.35x** | **1.31x** | 0.340 | 1.02x | 0.97x |

Table 18: Speedup of throughput, end-to-end (E2E) latency, decoding latency, and prefill latency of **Llama-3.1-8B** with an input token (**IT**) length of **128**, an output token (**OT**) length of **1024**, and batch sizes (**BS**) of **1, 16, 32, and 64**. **S.-0.86x** denotes the speedup when BD-LoRA has 0.86x the number of trainable parameters compared to S-LoRA or NFS-LoRA. **S.-1.73x** denotes the speedup when BD-LoRA has 1.73x the number of trainable parameters compared to S-LoRA or NFS-LoRA.

| Method | Rank | # Trainable Parameters | Throughput ↑ | | | E2E Latency ↓ | | | Decoding Latency ↓ | | | Prefill Latency ↓ | | |
|---|---|---|---|---|---|---|---|---|---|---|---|---|---|---|
| | | | Token/s | S.-0.86x | S.-1.73x | Time | S.-0.86x | S.-1.73x | Time | S.-0.86x | S.-1.73x | Time | S.-0.86x | S.-1.73x |
| | | | | | | Llama-3.1-8B BS=1 | | | | | | | | |
| S-LoRA | 16 | 41.9M | 110.7 | 1.00x | 1.00x | 9.25 | 1.00x | 1.00x | 0.0089 | 1.00x | 1.00x | 0.104 | 1.00x | 1.00x |
| | 32 | 83.9M | 110.4 | 1.00x | 1.00x | 9.28 | 1.00x | 1.00x | 0.0090 | 1.00x | 1.00x | 0.101 | 1.00x | 1.00x |
| | 64 | 167.8M | 108.2 | 1.00x | 1.00x | 9.47 | 1.00x | 1.00x | 0.0091 | 1.00x | 1.00x | 0.105 | 1.00x | 1.00x |
| | 128 | 335.5M | 104.7 | 1.00x | 1.00x | 9.78 | 1.00x | 1.00x | 0.0095 | 1.00x | 1.00x | 0.101 | 1.00x | 1.00x |
| | 256 | 671.1M | 92.8 | 1.00x | 1.00x | 11.03 | 1.00x | 1.00x | 0.0107 | 1.00x | 1.00x | 0.102 | 1.00x | 1.00x |
| BD-LoRA | 32 | 36.2M | 143.6 | 1.30x | N/A | 7.13 | 1.30x | N/A | 0.0069 | 1.30x | N/A | 0.088 | 1.18x | N/A |
| | 64 | 72.4M | 142.7 | 1.29x | **1.29x** | 7.18 | 1.29x | **1.29x** | 0.0069 | 1.29x | **1.29x** | 0.086 | 1.17x | 1.21x |
| | 128 | 144.7M | 141.8 | 1.31x | 1.28x | 7.22 | 1.31x | 1.28x | 0.0070 | 1.31x | **1.29x** | 0.088 | 1.20x | 1.15x |
| | 256 | 289.4M | 137.5 | 1.31x | 1.27x | 7.45 | 1.31x | 1.27x | 0.0072 | 1.32x | 1.27x | 0.093 | 1.09x | 1.13x |
| | 512 | 578.8M | 131.2 | 1.41x | 1.25x | 7.81 | 1.41x | 1.25x | 0.0075 | 1.42x | 1.25x | 0.087 | 1.18x | 1.17x |
| | | | | | | Llama-3.1-8B BS=16 | | | | | | | | |
| S-LoRA | 16 | 41.9M | 76.7 | 1.00x | 1.00x | 13.36 | 1.00x | 1.00x | 0.0128 | 1.00x | 1.00x | 0.297 | 1.00x | 1.00x |
| | 32 | 83.9M | 73.6 | 1.00x | 1.00x | 13.92 | 1.00x | 1.00x | 0.0133 | 1.00x | 1.00x | 0.288 | 1.00x | 1.00x |
| | 64 | 167.8M | 68.2 | 1.00x | 1.00x | 15.01 | 1.00x | 1.00x | 0.0144 | 1.00x | 1.00x | 0.283 | 1.00x | 1.00x |
| | 128 | 335.5M | 56.3 | 1.00x | 1.00x | 18.19 | 1.00x | 1.00x | 0.0175 | 1.00x | 1.00x | 0.287 | 1.00x | 1.00x |
| | 256 | 671.1M | 40.3 | 1.00x | 1.00x | 25.41 | 1.00x | 1.00x | 0.0245 | 1.00x | 1.00x | 0.292 | 1.00x | 1.00x |
| BD-LoRA | 32 | 36.2M | 90.4 | 1.18x | N/A | 11.32 | 1.18x | N/A | 0.0108 | 1.18x | N/A | 0.261 | 1.14x | N/A |
| | 64 | 72.4M | 89.0 | 1.21x | 1.16x | 11.50 | 1.21x | 1.16x | 0.0110 | 1.21x | 1.16x | 0.232 | 1.24x | 1.28x |
| | 128 | 144.7M | 88.3 | 1.29x | 1.20x | 11.59 | 1.29x | 1.20x | 0.0111 | 1.30x | 1.20x | 0.236 | 1.20x | 1.22x |
| | 256 | 289.4M | 81.4 | 1.45x | 1.19x | 12.58 | 1.45x | 1.19x | 0.0120 | 1.45x | 1.19x | 0.245 | 1.17x | 1.16x |
| | 512 | 578.8M | 59.0 | 1.46x | 1.05x | 17.36 | 1.46x | 1.05x | 0.0167 | 1.47x | 1.05x | 0.256 | 1.14x | 1.12x |
| | | | | | | Llama-3.1-8B BS=32 | | | | | | | | |
| S-LoRA | 16 | 41.9M | 64.0 | 1.00x | 1.00x | 16.01 | 1.00x | 1.00x | 0.0153 | 1.00x | 1.00x | 0.343 | 1.00x | 1.00x |
| | 32 | 83.9M | 60.6 | 1.00x | 1.00x | 16.91 | 1.00x | 1.00x | 0.0162 | 1.00x | 1.00x | 0.330 | 1.00x | 1.00x |
| | 64 | 167.8M | 53.4 | 1.00x | 1.00x | 19.17 | 1.00x | 1.00x | 0.0184 | 1.00x | 1.00x | 0.328 | 1.00x | 1.00x |
| | 128 | 335.5M | 40.0 | 1.00x | 1.00x | 25.61 | 1.00x | 1.00x | 0.0247 | 1.00x | 1.00x | 0.325 | 1.00x | 1.00x |
| | 256 | 671.1M | 26.5 | 1.00x | 1.00x | 38.67 | 1.00x | 1.00x | 0.0374 | 1.00x | 1.00x | 0.334 | 1.00x | 1.00x |
| BD-LoRA | 32 | 36.2M | 77.7 | 1.21x | N/A | 13.19 | 1.21x | N/A | 0.0126 | 1.21x | N/A | 0.264 | **1.30x** | N/A |
| | 64 | 72.4M | 73.7 | 1.22x | 1.15x | 13.89 | 1.22x | 1.15x | 0.0133 | 1.22x | 1.15x | 0.258 | 1.28x | **1.33x** |
| | 128 | 144.7M | 72.6 | 1.36x | 1.20x | 14.10 | 1.36x | 1.20x | 0.0135 | 1.36x | 1.20x | 0.277 | 1.18x | 1.19x |
| | 256 | 289.4M | 63.0 | 1.57x | 1.18x | 16.27 | 1.57x | 1.18x | 0.0156 | 1.58x | 1.18x | 0.277 | 1.18x | 1.18x |
| | 512 | 578.8M | 40.0 | 1.51x | 1.00x | 25.58 | 1.51x | 1.00x | 0.0247 | 1.52x | 1.00x | 0.276 | 1.21x | 1.18x |
| | | | | | | Llama-3.1-8B BS=64 | | | | | | | | |
| S-LoRA | 16 | 41.9M | 48.3 | 1.00x | 1.00x | 21.21 | 1.00x | 1.00x | 0.0204 | 1.00x | 1.00x | 0.353 | 1.00x | 1.00x |
| | 32 | 83.9M | 44.0 | 1.00x | 1.00x | 23.28 | 1.00x | 1.00x | 0.0224 | 1.00x | 1.00x | 0.337 | 1.00x | 1.00x |
| | 64 | 167.8M | 36.5 | 1.00x | 1.00x | 28.02 | 1.00x | 1.00x | 0.0270 | 1.00x | 1.00x | 0.341 | 1.00x | 1.00x |
| | 128 | 335.5M | 25.4 | 1.00x | 1.00x | 40.32 | 1.00x | 1.00x | 0.0390 | 1.00x | 1.00x | 0.379 | 1.00x | 1.00x |
| | 256 | 671.1M | 15.5 | 1.00x | 1.00x | 66.17 | 1.00x | 1.00x | 0.0643 | 1.00x | 1.00x | 0.370 | 1.00x | 1.00x |
| BD-LoRA | 32 | 36.2M | 56.1 | 1.16x | N/A | 18.27 | 1.16x | N/A | 0.0175 | 1.16x | N/A | 0.316 | 1.12x | N/A |
| | 64 | 72.4M | 51.9 | 1.18x | 1.07x | 19.74 | 1.18x | 1.07x | 0.0190 | 1.18x | 1.07x | 0.321 | 1.05x | 1.10x |
| | 128 | 144.7M | 50.6 | 1.39x | 1.15x | 20.23 | 1.39x | 1.15x | 0.0194 | 1.39x | 1.15x | 0.315 | 1.08x | 1.07x |
| | 256 | 289.4M | 41.5 | **1.63x** | 1.14x | 24.68 | **1.63x** | 1.14x | 0.0238 | **1.64x** | 1.14x | 0.336 | 1.13x | 1.01x |
| | 512 | 578.8M | 24.0 | 1.55x | 0.95x | 42.66 | 1.55x | 0.95x | 0.0413 | 1.56x | 0.94x | 0.340 | 1.09x | 1.11x |

Table 19: Speedup of throughput, end-to-end (E2E) latency, decoding latency, and prefill latency of **Llama-3.1-70B** with an input token (**IT**) length of **128**, an output token (**OT**) length of **1024**, and batch sizes (**BS**) of **1, 16, 32, and 64**. **S.-0.87x** denotes the speedup when BD-LoRA has 0.87x the number of trainable parameters compared to S-LoRA or NFS-LoRA. **S.-1.74x** denotes the speedup when BD-LoRA has 1.74x the number of trainable parameters compared to S-LoRA or NFS-LoRA.

| Method | Rank | # Trainable Parameters | Throughput ↑ | | | E2E Latency ↓ | | | Decoding Latency ↓ | | | Prefill Latency ↓ | | |
|---|---|---|---|---|---|---|---|---|---|---|---|---|---|---|
| | | | Token/s | S.-0.87x | S.-1.74x | Time | S.-0.87x | S.-1.74x | Time | S.-0.87x | S.-1.74x | Time | S.-0.87x | S.-1.74x |
| | | | | | | Llama-3.1-70B BS=1 | | | | | | | | |
| NFS-LoRA | 16 | 207.1M | 40.5 | 1.00x | 1.00x | 25.30 | 1.00x | 1.00x | 0.0245 | 1.00x | 1.00x | 0.226 | 1.00x | 1.00x |
| | 32 | 414.2M | 39.8 | 1.00x | 1.00x | 25.73 | 1.00x | 1.00x | 0.0249 | 1.00x | 1.00x | 0.219 | 1.00x | 1.00x |
| | 64 | 828.4M | 38.1 | 1.00x | 1.00x | 26.87 | 1.00x | 1.00x | 0.0260 | 1.00x | 1.00x | 0.227 | 1.00x | 1.00x |
| | 128 | 1656.8M | 35.9 | 1.00x | 1.00x | 28.50 | 1.00x | 1.00x | 0.0276 | 1.00x | 1.00x | 0.218 | 1.00x | 1.00x |
| | 256 | 3313.5M | 30.7 | 1.00x | 1.00x | 33.36 | 1.00x | 1.00x | 0.0324 | 1.00x | 1.00x | 0.222 | 1.00x | 1.00x |
| BD-LoRA | 32 | 180.2M | 41.1 | 1.02x | N/A | 24.92 | 1.02x | N/A | 0.0241 | 1.01x | N/A | 0.211 | 1.07x | N/A |
| | 64 | 360.4M | 40.8 | 1.03x | 1.01x | 25.09 | 1.03x | 1.01x | 0.0243 | 1.03x | 1.01x | 0.221 | 1.00x | 1.03x |
| | 128 | 720.9M | 40.8 | 1.07x | 1.03x | 25.08 | 1.07x | 1.03x | 0.0243 | 1.07x | 1.03x | 0.221 | 1.03x | 0.99x |
| | 256 | 1441.8M | 40.0 | 1.11x | 1.05x | 25.62 | 1.11x | 1.05x | 0.0248 | 1.11x | 1.05x | 0.222 | 0.98x | 1.02x |
| | 512 | 2883.6M | 38.0 | 1.24x | 1.06x | 26.92 | 1.24x | 1.06x | 0.0261 | 1.24x | 1.06x | 0.224 | 0.99x | 0.97x |
| | | | | | | Llama-3.1-70B BS=16 | | | | | | | | |
| NFS-LoRA | 16 | 207.1M | 29.9 | 1.00x | 1.00x | 34.28 | 1.00x | 1.00x | 0.0328 | 1.00x | 1.00x | 0.648 | 1.00x | 1.00x |
| | 32 | 414.2M | 27.8 | 1.00x | 1.00x | 36.78 | 1.00x | 1.00x | 0.0353 | 1.00x | 1.00x | 0.667 | 1.00x | 1.00x |
| | 64 | 828.4M | 23.9 | 1.00x | 1.00x | 42.88 | 1.00x | 1.00x | 0.0412 | 1.00x | 1.00x | 0.673 | 1.00x | 1.00x |
| | 128 | 1656.8M | 17.5 | 1.00x | 1.00x | 58.56 | 1.00x | 1.00x | 0.0566 | 1.00x | 1.00x | 0.644 | 1.00x | 1.00x |
| | 256 | 3313.5M | 11.3 | 1.00x | 1.00x | 90.78 | 1.00x | 1.00x | 0.0880 | 1.00x | 1.00x | 0.689 | 1.00x | 1.00x |
| BD-LoRA | 32 | 180.2M | 30.9 | 1.03x | N/A | 33.16 | 1.03x | N/A | 0.0318 | 1.03x | N/A | 0.639 | 1.01x | N/A |
| | 64 | 360.4M | 30.0 | 1.08x | 1.00x | 34.16 | 1.08x | 1.00x | 0.0327 | 1.08x | 1.00x | 0.652 | 1.02x | 0.99x |
| | 128 | 720.9M | 29.9 | 1.25x | 1.07x | 34.23 | 1.25x | 1.07x | 0.0328 | 1.26x | 1.07x | 0.631 | 1.07x | 1.06x |
| | 256 | 1441.8M | 27.8 | 1.59x | 1.16x | 36.81 | 1.59x | 1.16x | 0.0353 | 1.60x | 1.17x | 0.630 | 1.02x | 1.07x |
| | 512 | 2883.6M | 22.4 | 1.99x | 1.28x | 45.61 | 1.99x | 1.28x | 0.0439 | 2.00x | 1.29x | 0.667 | 1.03x | 0.97x |
| | | | | | | Llama-3.1-70B BS=32 | | | | | | | | |
| NFS-LoRA | 16 | 207.1M | 25.2 | 1.00x | 1.00x | 40.71 | 1.00x | 1.00x | 0.0389 | 1.00x | 1.00x | 0.825 | 1.00x | 1.00x |
| | 32 | 414.2M | 22.6 | 1.00x | 1.00x | 45.29 | 1.00x | 1.00x | 0.0434 | 1.00x | 1.00x | 0.809 | 1.00x | 1.00x |
| | 64 | 828.4M | 17.8 | 1.00x | 1.00x | 57.56 | 1.00x | 1.00x | 0.0554 | 1.00x | 1.00x | 0.858 | 1.00x | 1.00x |
| | 128 | 1656.8M | 11.6 | 1.00x | 1.00x | 88.34 | 1.00x | 1.00x | 0.0854 | 1.00x | 1.00x | 0.907 | 1.00x | 1.00x |
| | 256 | 3313.5M | 6.9 | 1.00x | 1.00x | 148.59 | 1.00x | 1.00x | 0.1441 | 1.00x | 1.00x | 1.041 | 1.00x | 1.00x |
| BD-LoRA | 32 | 180.2M | 26.3 | 1.04x | N/A | 38.96 | 1.04x | N/A | 0.0372 | 1.05x | N/A | 0.818 | 1.01x | N/A |
| | 64 | 360.4M | 25.2 | 1.12x | 1.00x | 40.60 | 1.12x | 1.00x | 0.0389 | 1.12x | 1.00x | 0.812 | 1.00x | 1.02x |
| | 128 | 720.9M | 25.1 | 1.41x | 1.11x | 40.76 | 1.41x | 1.11x | 0.0390 | 1.42x | 1.11x | 0.766 | 1.12x | 1.06x |
| | 256 | 1441.8M | 22.2 | 1.92x | 1.25x | 46.12 | 1.92x | 1.25x | 0.0442 | 1.93x | 1.25x | 0.814 | 1.11x | 1.05x |
| | 512 | 2883.6M | 16.3 | 2.36x | 1.41x | 62.86 | 2.36x | 1.41x | 0.0606 | 2.38x | 1.41x | 0.833 | 1.25x | 1.09x |
| | | | | | | Llama-3.1-70B BS=64 | | | | | | | | |
| NFS-LoRA | 16 | 207.1M | 19.9 | 1.00x | 1.00x | 51.55 | 1.00x | 1.00x | 0.0492 | 1.00x | 1.00x | 1.144 | 1.00x | 1.00x |
| | 32 | 414.2M | 16.8 | 1.00x | 1.00x | 60.95 | 1.00x | 1.00x | 0.0583 | 1.00x | 1.00x | 1.196 | 1.00x | 1.00x |
| | 64 | 828.4M | 12.2 | 1.00x | 1.00x | 84.24 | 1.00x | 1.00x | 0.0810 | 1.00x | 1.00x | 1.239 | 1.00x | 1.00x |
| | 128 | 1656.8M | 7.0 | 1.00x | 1.00x | 145.25 | 1.00x | 1.00x | 0.1407 | 1.00x | 1.00x | 1.218 | 1.00x | 1.00x |
| | 256 | 3313.5M | 3.9 | 1.00x | 1.00x | 262.10 | 1.00x | 1.00x | 0.2545 | 1.00x | 1.00x | 1.507 | 1.00x | 1.00x |
| BD-LoRA | 32 | 180.2M | 21.0 | 1.06x | N/A | 48.80 | 1.06x | N/A | 0.0465 | 1.06x | N/A | 1.212 | 0.94x | N/A |
| | 64 | 360.4M | 19.8 | 1.18x | 1.00x | 51.68 | 1.18x | 1.00x | 0.0493 | 1.18x | 1.00x | 1.196 | 1.00x | 0.96x |
| | 128 | 720.9M | 19.6 | 1.61x | 1.17x | 52.26 | 1.61x | 1.17x | 0.0500 | 1.62x | 1.17x | 1.100 | 1.13x | 1.09x |
| | 256 | 1441.8M | 16.5 | 2.34x | 1.36x | 62.02 | 2.34x | 1.36x | 0.0596 | 2.36x | 1.36x | 1.033 | 1.18x | **1.20x** |
| | 512 | 2883.6M | 10.8 | **2.77x** | **1.53x** | 94.77 | **2.77x** | **1.53x** | 0.0915 | **2.78x** | **1.54x** | 1.095 | **1.38x** | 1.11x |

Table 20: Speedup of throughput, end-to-end (E2E) latency, decoding latency, and prefill latency of **Llama-3.1-70B** with an input token (**IT**) length of **128**, an output token (**OT**) length of **1024**, and batch sizes (**BS**) of **1, 16, 32, and 64**. **S.-0.87x** denotes the speedup when BD-LoRA has 0.87x the number of trainable parameters compared to S-LoRA or NFS-LoRA. **S.-1.74x** denotes the speedup when BD-LoRA has 1.74x the number of trainable parameters compared to S-LoRA or NFS-LoRA.

| Method | Rank | # Trainable Parameters | Throughput ↑ | | | E2E Latency ↓ | | | Decoding Latency ↓ | | | Prefill Latency ↓ | | |
|---|---|---|---|---|---|---|---|---|---|---|---|---|---|---|
| | | | Token/s | S.-0.87x | S.-1.74x | Time | S.-0.87x | S.-1.74x | Time | S.-0.87x | S.-1.74x | Time | S.-0.87x | S.-1.74x |
| | | | | | | Llama-3.1-70B BS=1 | | | | | | | | |
| | 16 | 207.1M | 33.4 | 1.00x | 1.00x | 30.64 | 1.00x | 1.00x | 0.0297 | 1.00x | 1.00x | 0.243 | 1.00x | 1.00x |
| | 32 | 414.2M | 33.1 | 1.00x | 1.00x | 30.89 | 1.00x | 1.00x | 0.0299 | 1.00x | 1.00x | 0.255 | 1.00x | 1.00x |
| S-LoRA | 64 | 828.4M | 32.6 | 1.00x | 1.00x | 31.37 | 1.00x | 1.00x | 0.0304 | 1.00x | 1.00x | 0.250 | 1.00x | 1.00x |
| | 128 | 1656.8M | 31.4 | 1.00x | 1.00x | 32.63 | 1.00x | 1.00x | 0.0316 | 1.00x | 1.00x | 0.262 | 1.00x | 1.00x |
| | 256 | 3313.5M | 28.6 | 1.00x | 1.00x | 35.80 | 1.00x | 1.00x | 0.0347 | 1.00x | 1.00x | 0.266 | 1.00x | 1.00x |
| | 32 | 180.2M | 41.1 | 1.23x | N/A | 24.92 | 1.23x | N/A | 0.0241 | 1.23x | N/A | 0.211 | 1.15x | N/A |
| | 64 | 360.4M | 40.8 | 1.23x | 1.22x | 25.09 | 1.23x | 1.22x | 0.0243 | 1.23x | 1.22x | 0.221 | 1.16x | 1.10x |
| BD-LoRA | 128 | 720.9M | 40.8 | 1.25x | **1.23x** | 25.08 | 1.25x | **1.23x** | 0.0243 | 1.25x | **1.23x** | 0.221 | 1.13x | 1.15x |
| | 256 | 1441.8M | 40.0 | 1.27x | 1.22x | 25.62 | 1.27x | 1.22x | 0.0248 | 1.27x | **1.23x** | 0.222 | 1.18x | 1.13x |
| | 512 | 2883.6M | 38.0 | 1.33x | 1.21x | 26.92 | 1.33x | 1.21x | 0.0261 | 1.33x | 1.21x | 0.224 | **1.19x** | **1.17x** |
| | | | | | | Llama-3.1-70B BS=16 | | | | | | | | |
| | 16 | 207.1M | 25.3 | 1.00x | 1.00x | 40.40 | 1.00x | 1.00x | 0.0388 | 1.00x | 1.00x | 0.715 | 1.00x | 1.00x |
| | 32 | 414.2M | 24.5 | 1.00x | 1.00x | 41.86 | 1.00x | 1.00x | 0.0402 | 1.00x | 1.00x | 0.710 | 1.00x | 1.00x |
| S-LoRA | 64 | 828.4M | 22.9 | 1.00x | 1.00x | 44.79 | 1.00x | 1.00x | 0.0430 | 1.00x | 1.00x | 0.716 | 1.00x | 1.00x |
| | 128 | 1656.8M | 19.4 | 1.00x | 1.00x | 52.87 | 1.00x | 1.00x | 0.0509 | 1.00x | 1.00x | 0.741 | 1.00x | 1.00x |
| | 256 | 3313.5M | 14.4 | 1.00x | 1.00x | 71.21 | 1.00x | 1.00x | 0.0689 | 1.00x | 1.00x | 0.619 | 1.00x | 1.00x |
| | 32 | 180.2M | 30.9 | 1.22x | N/A | 33.16 | 1.22x | N/A | 0.0318 | 1.22x | N/A | 0.639 | 1.12x | N/A |
| | 64 | 360.4M | 30.0 | 1.23x | 1.18x | 34.16 | 1.23x | 1.18x | 0.0327 | 1.23x | 1.18x | 0.652 | 1.09x | 1.10x |
| BD-LoRA | 128 | 720.9M | 29.9 | 1.31x | 1.22x | 34.23 | 1.31x | 1.22x | 0.0328 | 1.31x | 1.22x | 0.631 | 1.13x | 1.13x |
| | 256 | 1441.8M | 27.8 | 1.44x | 1.22x | 36.81 | 1.44x | 1.22x | 0.0353 | 1.44x | 1.22x | 0.630 | 1.18x | 1.14x |
| | 512 | 2883.6M | 22.4 | 1.56x | 1.16x | 45.61 | 1.56x | 1.16x | 0.0439 | 1.57x | 1.16x | 0.667 | 0.93x | 1.11x |
| | | | | | | Llama-3.1-70B BS=32 | | | | | | | | |
| | 16 | 207.1M | 22.3 | 1.00x | 1.00x | 45.94 | 1.00x | 1.00x | 0.0440 | 1.00x | 1.00x | 0.870 | 1.00x | 1.00x |
| | 32 | 414.2M | 21.2 | 1.00x | 1.00x | 48.29 | 1.00x | 1.00x | 0.0463 | 1.00x | 1.00x | 0.862 | 1.00x | 1.00x |
| S-LoRA | 64 | 828.4M | 18.9 | 1.00x | 1.00x | 54.26 | 1.00x | 1.00x | 0.0522 | 1.00x | 1.00x | 0.854 | 1.00x | 1.00x |
| | 128 | 1656.8M | 14.6 | 1.00x | 1.00x | 70.19 | 1.00x | 1.00x | 0.0677 | 1.00x | 1.00x | 0.879 | 1.00x | 1.00x |
| | 256 | 3313.5M | 9.9 | 1.00x | 1.00x | 103.09 | 1.00x | 1.00x | 0.0999 | 1.00x | 1.00x | 0.814 | 1.00x | 1.00x |
| | 32 | 180.2M | 26.3 | 1.18x | N/A | 38.96 | 1.18x | N/A | 0.0372 | 1.18x | N/A | 0.818 | 1.06x | N/A |
| | 64 | 360.4M | 25.2 | 1.19x | 1.13x | 40.60 | 1.19x | 1.13x | 0.0389 | 1.19x | 1.13x | 0.812 | 1.06x | 1.07x |
| BD-LoRA | 128 | 720.9M | 25.1 | 1.33x | 1.18x | 40.76 | 1.33x | 1.18x | 0.0390 | 1.34x | 1.19x | 0.766 | 1.12x | 1.12x |
| | 256 | 1441.8M | 22.2 | 1.52x | 1.18x | 46.12 | 1.52x | 1.18x | 0.0442 | 1.53x | 1.18x | 0.814 | 1.08x | 1.05x |
| | 512 | 2883.6M | 16.3 | 1.64x | 1.12x | 62.86 | 1.64x | 1.12x | 0.0606 | 1.65x | 1.12x | 0.833 | 0.98x | 1.06x |
| | | | | | | Llama-3.1-70B BS=64 | | | | | | | | |
| | 16 | 207.1M | 18.4 | 1.00x | 1.00x | 55.64 | 1.00x | 1.00x | 0.0533 | 1.00x | 1.00x | 1.096 | 1.00x | 1.00x |
| | 32 | 414.2M | 16.8 | 1.00x | 1.00x | 60.99 | 1.00x | 1.00x | 0.0585 | 1.00x | 1.00x | 1.105 | 1.00x | 1.00x |
| S-LoRA | 64 | 828.4M | 14.0 | 1.00x | 1.00x | 72.89 | 1.00x | 1.00x | 0.0701 | 1.00x | 1.00x | 1.111 | 1.00x | 1.00x |
| | 128 | 1656.8M | 10.0 | 1.00x | 1.00x | 102.66 | 1.00x | 1.00x | 0.0992 | 1.00x | 1.00x | 1.114 | 1.00x | 1.00x |
| | 256 | 3313.5M | 6.0 | 1.00x | 1.00x | 169.45 | 1.00x | 1.00x | 0.1643 | 1.00x | 1.00x | 1.202 | 1.00x | 1.00x |
| | 32 | 180.2M | 21.0 | 1.14x | N/A | 48.80 | 1.14x | N/A | 0.0465 | 1.15x | N/A | 1.212 | 0.90x | N/A |
| | 64 | 360.4M | 19.8 | 1.18x | 1.08x | 51.68 | 1.18x | 1.08x | 0.0493 | 1.19x | 1.08x | 1.196 | 0.92x | 0.92x |
| BD-LoRA | 128 | 720.9M | 19.6 | 1.39x | 1.17x | 52.26 | 1.39x | 1.17x | 0.0500 | 1.40x | 1.17x | 1.100 | 1.01x | 1.00x |
| | 256 | 1441.8M | 16.5 | 1.66x | 1.18x | 62.02 | 1.66x | 1.18x | 0.0596 | 1.67x | 1.18x | 1.033 | 1.08x | 1.08x |
| | 512 | 2883.6M | 10.8 | **1.79x** | 1.08x | 94.77 | **1.79x** | 1.08x | 0.0915 | **1.80x** | 1.08x | 1.095 | 1.10x | 1.02x |

Table 21: Speedup of throughput, end-to-end (E2E) latency, decoding latency, and prefill latency of **Llama-3.1-8B** with a **TP** of **4** input token (**IT**) length of **1024**, an output token (**OT**) length of **128**, and batch sizes (**BS**) of **1, 16, 32, and 64**. **S.-0.51** denotes the speedup when BD-LoRA has 0.51 the number of trainable parameters compared to S-LoRA or NFS-LoRA. **S.-1.03x** denotes the speedup when BD-LoRA has 1.03x the number of trainable parameters compared to S-LoRA or NFS-LoRA.

| Method | Rank | # Trainable Parameters | Throughput ↑ | | | E2E Latency ↓ | | | Decoding Latency ↓ | | | Prefill Latency ↓ | | |
|---|---|---|---|---|---|---|---|---|---|---|---|---|---|---|
| | | | Token/s | S.-0.51 | S.-1.03x | Time | S.-0.51 | S.-1.03x | Time | S.-0.51 | S.-1.03x | Time | S.-0.51 | S.-1.03x |
| **Llama-3.1-8B TP=4 BS=1** | | | | | | | | | | | | | | |
| NFS-LoRA | 16 | 41.9M | 109.5 | 1.00x | 1.00x | 1.17 | 1.00x | 1.00x | 0.0084 | 1.00x | 1.00x | 0.093 | 1.00x | 1.00x |
| | 32 | 83.9M | 107.5 | 1.00x | 1.00x | 1.19 | 1.00x | 1.00x | 0.0086 | 1.00x | 1.00x | 0.093 | 1.00x | 1.00x |
| | 64 | 167.8M | 104.6 | 1.00x | 1.00x | 1.22 | 1.00x | 1.00x | 0.0089 | 1.00x | 1.00x | 0.087 | 1.00x | 1.00x |
| | 128 | 335.5M | 99.4 | 1.00x | 1.00x | 1.29 | 1.00x | 1.00x | 0.0094 | 1.00x | 1.00x | 0.083 | 1.00x | 1.00x |
| | 256 | 671.1M | 82.3 | 1.00x | 1.00x | 1.56 | 1.00x | 1.00x | 0.0114 | 1.00x | 1.00x | 0.093 | 1.00x | 1.00x |
| BD-LoRA | 32 | 43.0M | 112.4 | 1.04x | 1.03x | 1.14 | 1.04x | 1.03x | 0.0082 | 1.04x | 1.02x | 0.086 | 1.08x | 1.08x |
| | 64 | 86.0M | 113.3 | 1.08x | 1.05x | 1.13 | 1.08x | 1.05x | 0.0082 | 1.08x | 1.04x | 0.080 | 1.09x | 1.17x |
| | 128 | 172.0M | 110.9 | 1.11x | 1.06x | 1.15 | 1.11x | 1.06x | 0.0084 | 1.12x | 1.06x | 0.081 | 1.02x | 1.07x |
| | 256 | 343.9M | 104.4 | 1.27x | 1.05x | 1.23 | 1.27x | 1.05x | 0.0089 | 1.29x | 1.06x | 0.093 | 0.99x | 0.89x |
| | 512 | 687.9M | 97.9 | N/A | 1.19x | 1.31 | N/A | 1.19x | 0.0096 | N/A | 1.19x | 0.082 | N/A | 1.13x |
| **Llama-3.1-8B TP=4 BS=16** | | | | | | | | | | | | | | |
| NFS-LoRA | 16 | 41.9M | 59.5 | 1.00x | 1.00x | 2.16 | 1.00x | 1.00x | 0.0134 | 1.00x | 1.00x | 0.436 | 1.00x | 1.00x |
| | 32 | 83.9M | 55.1 | 1.00x | 1.00x | 2.33 | 1.00x | 1.00x | 0.0145 | 1.00x | 1.00x | 0.467 | 1.00x | 1.00x |
| | 64 | 167.8M | 49.6 | 1.00x | 1.00x | 2.59 | 1.00x | 1.00x | 0.0167 | 1.00x | 1.00x | 0.450 | 1.00x | 1.00x |
| | 128 | 335.5M | 37.2 | 1.00x | 1.00x | 3.45 | 1.00x | 1.00x | 0.0230 | 1.00x | 1.00x | 0.507 | 1.00x | 1.00x |
| | 256 | 671.1M | 25.0 | 1.00x | 1.00x | 5.14 | 1.00x | 1.00x | 0.0347 | 1.00x | 1.00x | 0.697 | 1.00x | 1.00x |
| BD-LoRA | 32 | 43.0M | 55.1 | 1.00x | 0.93x | 2.47 | 0.94x | 0.87x | 0.0136 | 1.07x | 0.99x | 0.733 | 0.64x | 0.60x |
| | 64 | 86.0M | 55.6 | 1.12x | 1.01x | 2.46 | 1.05x | 0.95x | 0.0134 | 1.25x | 1.08x | 0.745 | 0.60x | 0.63x |
| | 128 | 172.0M | 53.2 | 1.43x | 1.07x | 2.41 | 1.43x | 1.07x | 0.0150 | 1.53x | 1.12x | 0.495 | 1.02x | 0.91x |
| | 256 | 343.9M | 48.9 | 1.96x | 1.32x | 2.62 | 1.96x | 1.32x | 0.0169 | 2.05x | 1.36x | 0.455 | **1.53x** | 1.11x |
| | 512 | 687.9M | 34.2 | N/A | 1.37x | 3.74 | N/A | 1.38x | 0.0252 | N/A | 1.38x | 0.506 | N/A | **1.38x** |
| **Llama-3.1-8B TP=4 BS=32** | | | | | | | | | | | | | | |
| NFS-LoRA | 16 | 41.9M | 37.0 | 1.00x | 1.00x | 3.50 | 1.00x | 1.00x | 0.0198 | 1.00x | 1.00x | 0.961 | 1.00x | 1.00x |
| | 32 | 83.9M | 35.7 | 1.00x | 1.00x | 3.61 | 1.00x | 1.00x | 0.0210 | 1.00x | 1.00x | 0.923 | 1.00x | 1.00x |
| | 64 | 167.8M | 27.7 | 1.00x | 1.00x | 4.73 | 1.00x | 1.00x | 0.0276 | 1.00x | 1.00x | 1.189 | 1.00x | 1.00x |
| | 128 | 335.5M | 21.5 | 1.00x | 1.00x | 6.00 | 1.00x | 1.00x | 0.0392 | 1.00x | 1.00x | 0.986 | 1.00x | 1.00x |
| | 256 | 671.1M | 14.4 | 1.00x | 1.00x | 8.93 | 1.00x | 1.00x | 0.0614 | 1.00x | 1.00x | 1.070 | 1.00x | 1.00x |
| BD-LoRA | 32 | 43.0M | 38.0 | 1.06x | 1.03x | 3.40 | 1.06x | 1.03x | 0.0194 | 1.08x | 1.02x | 0.911 | 1.01x | 1.06x |
| | 64 | 86.0M | 39.0 | 1.41x | 1.09x | 3.31 | 1.43x | 1.09x | 0.0192 | 1.44x | 1.10x | 0.856 | 1.39x | 1.08x |
| | 128 | 172.0M | 36.1 | 1.68x | 1.30x | 3.58 | 1.68x | 1.32x | 0.0209 | 1.87x | 1.32x | 0.901 | 1.09x | 1.32x |
| | 256 | 343.9M | 30.2 | **2.10x** | **1.41x** | 4.26 | **2.09x** | **1.41x** | 0.0264 | **2.32x** | **1.48x** | 0.878 | 1.22x | 1.12x |
| | 512 | 687.9M | 19.6 | N/A | 1.37x | 6.56 | N/A | 1.36x | 0.0432 | N/A | 1.42x | 1.027 | N/A | 1.04x |
| **Llama-3.1-8B TP=4 BS=64** | | | | | | | | | | | | | | |
| NFS-LoRA | 16 | 41.9M | 19.3 | 1.00x | 1.00x | 6.74 | 1.00x | 1.00x | 0.0383 | 1.00x | 1.00x | 1.824 | 1.00x | 1.00x |
| | 32 | 83.9M | 18.8 | 1.00x | 1.00x | 6.96 | 1.00x | 1.00x | 0.0408 | 1.00x | 1.00x | 1.735 | 1.00x | 1.00x |
| | 64 | 167.8M | 16.0 | 1.00x | 1.00x | 8.06 | 1.00x | 1.00x | 0.0495 | 1.00x | 1.00x | 1.714 | 1.00x | 1.00x |
| | 128 | 335.5M | 11.6 | 1.00x | 1.00x | 11.10 | 1.00x | 1.00x | 0.0728 | 1.00x | 1.00x | 1.780 | 1.00x | 1.00x |
| | 256 | 671.1M | 7.6 | 1.00x | 1.00x | 16.92 | 1.00x | 1.00x | 0.1149 | 1.00x | 1.00x | 2.206 | 1.00x | 1.00x |
| BD-LoRA | 32 | 43.0M | 21.7 | 1.16x | 1.12x | 5.91 | 1.18x | 1.14x | 0.0334 | 1.22x | 1.15x | 1.626 | 1.07x | 1.12x |
| | 64 | 86.0M | 22.6 | 1.41x | 1.20x | 5.72 | 1.41x | 1.22x | 0.0332 | 1.49x | 1.23x | 1.461 | 1.17x | 1.19x |
| | 128 | 172.0M | 20.5 | 1.77x | 1.28x | 6.27 | 1.77x | 1.28x | 0.0369 | 1.97x | 1.34x | 1.548 | 1.15x | 1.11x |
| | 256 | 343.9M | 15.9 | **2.10x** | 1.37x | 8.17 | 2.07x | 1.36x | 0.0512 | 2.24x | 1.42x | 1.609 | 1.37x | 1.11x |
| | 512 | 687.9M | 10.3 | N/A | 1.36x | 12.52 | N/A | 1.35x | 0.0820 | N/A | 1.40x | 2.015 | N/A | 1.09x |

Table 22: Speedup of throughput, end-to-end (E2E) latency, decoding latency, and prefill latency of **Llama-3.1-8B** with a **TP** of **4** input token (**IT**) length of **1024**, an output token (**OT**) length of **128**, and batch sizes (**BS**) of **1, 16, 32, and 64**. **S.-0.51** denotes the speedup when BD-LoRA has 0.51 the number of trainable parameters compared to S-LoRA or NFS-LoRA. **S.-1.03x** denotes the speedup when BD-LoRA has 1.03x the number of trainable parameters compared to S-LoRA or NFS-LoRA.

| Method | Rank | # Trainable Parameters | Throughput ↑ | | | E2E Latency ↓ | | | Decoding Latency ↓ | | | Prefill Latency ↓ | | |
|---|---|---|---|---|---|---|---|---|---|---|---|---|---|---|
| | | | Token/s | S.-0.51 | S.-1.03x | Time | S.-0.51 | S.-1.03x | Time | S.-0.51 | S.-1.03x | Time | S.-0.51 | S.-1.03x |
| | | | | | | Llama-3.1-8B TP=4 BS=1 | | | | | | | | |
| S-LoRA | 16 | 41.9M | 91.9 | 1.00x | 1.00x | 1.39 | 1.00x | 1.00x | 0.0101 | 1.00x | 1.00x | 0.107 | 1.00x | 1.00x |
| | 32 | 83.9M | 91.2 | 1.00x | 1.00x | 1.40 | 1.00x | 1.00x | 0.0101 | 1.00x | 1.00x | 0.106 | 1.00x | 1.00x |
| | 64 | 167.8M | 89.3 | 1.00x | 1.00x | 1.43 | 1.00x | 1.00x | 0.0104 | 1.00x | 1.00x | 0.107 | 1.00x | 1.00x |
| | 128 | 335.5M | 85.5 | 1.00x | 1.00x | 1.50 | 1.00x | 1.00x | 0.0108 | 1.00x | 1.00x | 0.114 | 1.00x | 1.00x |
| | 256 | 671.1M | 79.1 | 1.00x | 1.00x | 1.62 | 1.00x | 1.00x | 0.0119 | 1.00x | 1.00x | 0.099 | 1.00x | 1.00x |
| BD-LoRA | 32 | 43.0M | 112.4 | 1.23x | 1.22x | 1.14 | 1.23x | 1.22x | 0.0082 | 1.23x | 1.22x | 0.086 | 1.22x | 1.24x |
| | 64 | 86.0M | 113.3 | 1.27x | 1.24x | 1.13 | 1.27x | 1.24x | 0.0082 | 1.26x | 1.24x | 0.080 | 1.35x | 1.33x |
| | 128 | 172.0M | 110.9 | 1.30x | 1.24x | 1.15 | 1.30x | 1.24x | 0.0084 | 1.29x | 1.24x | 0.081 | 1.40x | 1.32x |
| | 256 | 343.9M | 104.4 | 1.32x | 1.22x | 1.23 | 1.32x | 1.22x | 0.0089 | 1.34x | 1.22x | 0.093 | 1.06x | 1.23x |
| | 512 | 687.9M | 97.9 | N/A | 1.24x | 1.31 | N/A | 1.24x | 0.0096 | N/A | 1.24x | 0.082 | N/A | 1.20x |
| | | | | | | Llama-3.1-8B TP=4 BS=16 | | | | | | | | |
| S-LoRA | 16 | 41.9M | 49.5 | 1.00x | 1.00x | 2.59 | 1.00x | 1.00x | 0.0159 | 1.00x | 1.00x | 0.556 | 1.00x | 1.00x |
| | 32 | 83.9M | 48.6 | 1.00x | 1.00x | 2.64 | 1.00x | 1.00x | 0.0163 | 1.00x | 1.00x | 0.555 | 1.00x | 1.00x |
| | 64 | 167.8M | 44.9 | 1.00x | 1.00x | 2.86 | 1.00x | 1.00x | 0.0178 | 1.00x | 1.00x | 0.572 | 1.00x | 1.00x |
| | 128 | 335.5M | 39.4 | 1.00x | 1.00x | 3.25 | 1.00x | 1.00x | 0.0210 | 1.00x | 1.00x | 0.557 | 1.00x | 1.00x |
| | 256 | 671.1M | 29.0 | 1.00x | 1.00x | 4.49 | 1.00x | 1.00x | 0.0289 | 1.00x | 1.00x | 0.797 | 1.00x | 1.00x |
| BD-LoRA | 32 | 43.0M | 55.1 | 1.13x | 1.11x | 2.47 | 1.07x | 1.05x | 0.0136 | 1.20x | 1.17x | 0.733 | 0.76x | 0.76x |
| | 64 | 86.0M | 55.6 | 1.24x | 1.14x | 2.46 | 1.16x | 1.07x | 0.0134 | 1.33x | 1.22x | 0.745 | 0.77x | 0.75x |
| | 128 | 172.0M | 53.2 | 1.35x | 1.18x | 2.41 | 1.35x | 1.19x | 0.0150 | 1.41x | 1.19x | 0.495 | 1.12x | 1.15x |
| | 256 | 343.9M | 48.9 | 1.68x | 1.24x | 2.62 | **1.71x** | 1.24x | 0.0169 | 1.71x | 1.24x | 0.455 | **1.75x** | 1.23x |
| | 512 | 687.9M | 34.2 | N/A | 1.18x | 3.74 | N/A | 1.20x | 0.0252 | N/A | 1.14x | 0.506 | N/A | **1.57x** |
| | | | | | | Llama-3.1-8B TP=4 BS=32 | | | | | | | | |
| S-LoRA | 16 | 41.9M | 32.4 | 1.00x | 1.00x | 4.00 | 1.00x | 1.00x | 0.0230 | 1.00x | 1.00x | 1.051 | 1.00x | 1.00x |
| | 32 | 83.9M | 32.3 | 1.00x | 1.00x | 4.01 | 1.00x | 1.00x | 0.0236 | 1.00x | 1.00x | 0.990 | 1.00x | 1.00x |
| | 64 | 167.8M | 29.5 | 1.00x | 1.00x | 4.38 | 1.00x | 1.00x | 0.0263 | 1.00x | 1.00x | 1.014 | 1.00x | 1.00x |
| | 128 | 335.5M | 25.2 | 1.00x | 1.00x | 5.11 | 1.00x | 1.00x | 0.0324 | 1.00x | 1.00x | 0.963 | 1.00x | 1.00x |
| | 256 | 671.1M | 17.8 | 1.00x | 1.00x | 7.22 | 1.00x | 1.00x | 0.0473 | 1.00x | 1.00x | 1.158 | 1.00x | 1.00x |
| BD-LoRA | 32 | 43.0M | 38.0 | 1.18x | 1.17x | 3.40 | 1.18x | 1.18x | 0.0194 | 1.21x | 1.18x | 0.911 | 1.09x | 1.15x |
| | 64 | 86.0M | 39.0 | 1.32x | 1.21x | 3.31 | 1.32x | 1.21x | 0.0192 | 1.37x | 1.23x | 0.856 | 1.18x | 1.16x |
| | 128 | 172.0M | 36.1 | 1.43x | 1.22x | 3.58 | 1.43x | 1.22x | 0.0209 | 1.55x | **1.26x** | 0.901 | 1.07x | 1.13x |
| | 256 | 343.9M | 30.2 | **1.70x** | 1.20x | 4.26 | 1.69x | 1.20x | 0.0264 | **1.79x** | 1.22x | 0.878 | 1.32x | 1.10x |
| | 512 | 687.9M | 19.6 | N/A | 1.10x | 6.56 | N/A | 1.10x | 0.0432 | N/A | 1.10x | 1.027 | N/A | 1.13x |
| | | | | | | Llama-3.1-8B TP=4 BS=64 | | | | | | | | |
| S-LoRA | 16 | 41.9M | 17.9 | 1.00x | 1.00x | 7.22 | 1.00x | 1.00x | 0.0400 | 1.00x | 1.00x | 2.096 | 1.00x | 1.00x |
| | 32 | 83.9M | 17.9 | 1.00x | 1.00x | 7.26 | 1.00x | 1.00x | 0.0408 | 1.00x | 1.00x | 2.025 | 1.00x | 1.00x |
| | 64 | 167.8M | 16.8 | 1.00x | 1.00x | 7.72 | 1.00x | 1.00x | 0.0466 | 1.00x | 1.00x | 1.750 | 1.00x | 1.00x |
| | 128 | 335.5M | 13.8 | 1.00x | 1.00x | 9.37 | 1.00x | 1.00x | 0.0587 | 1.00x | 1.00x | 1.859 | 1.00x | 1.00x |
| | 256 | 671.1M | 9.5 | 1.00x | 1.00x | 13.58 | 1.00x | 1.00x | 0.0873 | 1.00x | 1.00x | 2.408 | 1.00x | 1.00x |
| BD-LoRA | 32 | 43.0M | 21.7 | 1.21x | 1.21x | 5.91 | 1.23x | 1.22x | 0.0334 | 1.22x | 1.20x | 1.626 | 1.25x | 1.29x |
| | 64 | 86.0M | 22.6 | 1.35x | **1.26x** | 5.72 | 1.35x | **1.27x** | 0.0332 | 1.40x | 1.23x | 1.461 | 1.20x | 1.39x |
| | 128 | 172.0M | 20.5 | 1.49x | 1.23x | 6.27 | 1.49x | 1.23x | 0.0369 | 1.59x | **1.26x** | 1.548 | 1.20x | 1.13x |
| | 256 | 343.9M | 15.9 | 1.67x | 1.15x | 8.17 | 1.66x | 1.15x | 0.0512 | 1.70x | 1.14x | 1.609 | 1.50x | 1.16x |
| | 512 | 687.9M | 10.3 | N/A | 1.08x | 12.52 | N/A | 1.08x | 0.0820 | N/A | 1.06x | 2.015 | N/A | 1.20x |

Table 23: Speedup of throughput, end-to-end (E2E) latency, decoding latency, and prefill latency of **Llama-3.1-8B** and **Llama-3.1-70B** using **multi-adapters** loaded from the disk with an input Token (**IT**) length of **1024**, an output Token (**OT**) length of **128**, and batch sizes (**BS**) of **1**. **S.-OO.** and **S.-G.** denote the speedup with respect to OpenOrca and GLUE, respectively.

| Method | Rank | # Trainable Parameters | OpenOrca↓ | GLUE↑ | Throughput ↑ | | | E2E Latency ↓ | | | Decoding Latency ↓ | | | Prefill Latency ↓ | | |
|---|---|---|---|---|---|---|---|---|---|---|---|---|---|---|---|---|
| | | | | | Token/s | S.-OO. | S.-G. | Time | S.-OO. | S.-G. | Time | S.-OO. | S.-G. | Time | S.-OO. | S.-G. |
| NFS-LoRA | 16 | 41.9M | 2.316 | 75.82 | 115.9 | 1.00x | 1.00x | 1.11 | 1.00x | 1.00x | 0.0071 | 1.00x | 1.00x | 0.194 | 1.00x | 1.00x |
| | 32 | 83.9M | 2.309 | 75.87 | 108.2 | 1.00x | 1.00x | 1.18 | 1.00x | 1.00x | 0.0071 | 1.00x | 1.00x | 0.273 | 1.00x | 1.00x |
| | 64 | 167.8M | 2.303 | 76.01 | 94.5 | 1.00x | 1.00x | 1.36 | 1.00x | 1.00x | 0.0075 | 1.00x | 1.00x | 0.394 | 1.00x | 1.00x |
| | 128 | 335.5M | 2.297 | 76.28 | 74.8 | 1.00x | 1.00x | 1.71 | 1.00x | 1.00x | 0.0082 | 1.00x | 1.00x | 0.666 | 1.00x | 1.00x |
| | 256 | 671.1M | 2.290 | 76.39 | 51.0 | 1.00x | 1.00x | 2.51 | 1.00x | 1.00x | 0.0098 | 1.00x | 1.00x | 1.256 | 1.00x | 1.00x |
| BD-LoRA | 32 | 36.2M | 2.318 | 75.58 | 121.5 | N/A | N/A | 1.05 | N/A | N/A | 0.0069 | N/A | N/A | 0.174 | N/A | N/A |
| | 64 | 72.4M | 2.310 | 75.90 | 113.8 | 0.98x | 1.05x | 1.13 | 0.98x | 1.05x | 0.0070 | 1.02x | 1.02x | 0.233 | 0.83x | 1.17x |
| | 128 | 144.7M | 2.303 | 76.17 | 106.2 | 0.98x | 1.12x | 1.21 | 0.98x | 1.12x | 0.0068 | 1.04x | 1.10x | 0.331 | 0.83x | 1.19x |
| | 256 | 289.4M | 2.296 | 76.55 | 88.4 | 1.18x | **1.73x** | 1.45 | 1.18x | **1.73x** | 0.0070 | 1.16x | **1.39x** | 0.549 | **1.21x** | 2.29x |
| | 512 | 578.8M | **2.289** | **76.59** | 63.5 | **1.25x** | 1.25x | 2.02 | **1.25x** | 1.25x | 0.0076 | **1.29x** | 1.29x | 1.045 | 1.20x | 1.20x |
| S-LoRA | 16 | 41.9M | 2.316 | 75.82 | 94.5 | 1.00x | 1.00x | 1.36 | 1.00x | 1.00x | 0.0089 | 1.00x | 1.00x | 0.218 | 1.00x | 1.00x |
| | 32 | 83.9M | 2.309 | 75.87 | 89.1 | 1.00x | 1.00x | 1.44 | 1.00x | 1.00x | 0.0090 | 1.00x | 1.00x | 0.291 | 1.00x | 1.00x |
| | 64 | 167.8M | 2.303 | 76.01 | 81.2 | 1.00x | 1.00x | 1.58 | 1.00x | 1.00x | 0.0091 | 1.00x | 1.00x | 0.413 | 1.00x | 1.00x |
| | 128 | 335.5M | 2.297 | 76.28 | 67.8 | 1.00x | 1.00x | 1.89 | 1.00x | 1.00x | 0.0095 | 1.00x | 1.00x | 0.669 | 1.00x | 1.00x |
| | 256 | 671.1M | 2.290 | 76.39 | 49.6 | 1.00x | 1.00x | 2.58 | 1.00x | 1.00x | 0.0107 | 1.00x | 1.00x | 1.214 | 1.00x | 1.00x |
| BD-LoRA | 32 | 36.2M | 2.318 | 75.58 | 121.5 | N/A | N/A | 1.05 | N/A | N/A | 0.0069 | N/A | N/A | 0.174 | N/A | N/A |
| | 64 | 72.4M | 2.310 | 75.90 | 113.8 | 1.20x | 1.28x | 1.13 | 1.20x | 1.28x | 0.0070 | 1.28x | 1.28x | 0.233 | 0.93x | 1.25x |
| | 128 | 144.7M | 2.303 | 76.17 | 106.2 | 1.19x | 1.31x | 1.21 | 1.19x | 1.31x | 0.0068 | 1.31x | 1.33x | 0.331 | 0.88x | 1.25x |
| | 256 | 289.4M | 2.296 | 76.55 | 88.4 | **1.30x** | **1.78x** | 1.45 | **1.30x** | **1.78x** | 0.0070 | 1.36x | **1.52x** | 0.549 | **1.22x** | 2.21x |
| | 512 | 578.8M | **2.289** | **76.59** | 63.5 | 1.28x | 1.28x | 2.02 | 1.28x | 1.28x | 0.0076 | **1.41x** | 1.41x | 1.045 | 1.16x | 1.16x |

Table 24: Speedup of throughput, end-to-end (E2E) latency, decoding latency, and prefill latency of **Llama-3.1-8B** using **multi-adapters** with an input Token (**IT**) length of **1024**, an output Token (**OT**) length of **128**, and batch sizes (**BS**) of **1**. **S.-0.86x** denotes the speedup when BD-LoRA has 0.86x the number of trainable parameters compared to S-LoRA or NFS-LoRA. **S.-1.73x** denotes the speedup when BD-LoRA has 1.73x the number of trainable parameters compared to S-LoRA or NFS-LoRA.

| Method | Rank | # Trainable Parameters | Throughput ↑ | | | E2E Latency ↓ | | | Decoding Latency ↓ | | | Prefill Latency ↓ | | |
|---|---|---|---|---|---|---|---|---|---|---|---|---|---|---|
| | | | Token/s | S.-0.86x | S.-1.73x | Time | S.-0.86x | S.-1.73x | Time | S.-0.86x | S.-1.73x | Time | S.-0.86x | S.-1.73x |
| NFS-LoRA | 16 | 41.9M | 115.9 | 1.00x | 1.00x | 1.11 | 1.00x | 1.00x | 0.0071 | 1.00x | 1.00x | 0.194 | 1.00x | 1.00x |
| | 32 | 83.9M | 108.2 | 1.00x | 1.00x | 1.18 | 1.00x | 1.00x | 0.0071 | 1.00x | 1.00x | 0.273 | 1.00x | 1.00x |
| | 64 | 167.8M | 94.5 | 1.00x | 1.00x | 1.36 | 1.00x | 1.00x | 0.0075 | 1.00x | 1.00x | 0.394 | 1.00x | 1.00x |
| | 128 | 335.5M | 74.8 | 1.00x | 1.00x | 1.71 | 1.00x | 1.00x | 0.0082 | 1.00x | 1.00x | 0.666 | 1.00x | 1.00x |
| | 256 | 671.1M | 51.0 | 1.00x | 1.00x | 2.51 | 1.00x | 1.00x | 0.0098 | 1.00x | 1.00x | 1.256 | 1.00x | 1.00x |
| BD-LoRA | 32 | 36.2M | 121.5 | 1.05x | N/A | 1.05 | 1.05x | N/A | 0.0069 | 1.03x | N/A | 0.174 | 1.12x | N/A |
| | 64 | 72.4M | 113.8 | 1.05x | **0.98x** | 1.13 | 1.05x | **0.98x** | 0.0070 | 1.02x | 1.02x | 0.233 | 1.17x | **0.83x** |
| | 128 | 144.7M | 106.2 | 1.12x | **0.98x** | 1.21 | 1.12x | **0.98x** | 0.0068 | 1.10x | 1.04x | 0.331 | 1.19x | **0.83x** |
| | 256 | 289.4M | 88.4 | 1.18x | 0.94x | 1.45 | 1.18x | 0.94x | 0.0070 | 1.16x | 1.07x | 0.549 | **1.21x** | 0.72x |
| | 512 | 578.8M | 63.5 | **1.25x** | 0.85x | 2.02 | **1.25x** | 0.85x | 0.0076 | **1.29x** | 1.08x | 1.045 | 1.20x | 0.64x |
| S-LoRA | 16 | 41.9M | 94.5 | 1.00x | 1.00x | 1.36 | 1.00x | 1.00x | 0.0089 | 1.00x | 1.00x | 0.218 | 1.00x | 1.00x |
| | 32 | 83.9M | 89.1 | 1.00x | 1.00x | 1.44 | 1.00x | 1.00x | 0.0090 | 1.00x | 1.00x | 0.291 | 1.00x | 1.00x |
| | 64 | 167.8M | 81.2 | 1.00x | 1.00x | 1.58 | 1.00x | 1.00x | 0.0091 | 1.00x | 1.00x | 0.413 | 1.00x | 1.00x |
| | 128 | 335.5M | 67.8 | 1.00x | 1.00x | 1.89 | 1.00x | 1.00x | 0.0095 | 1.00x | 1.00x | 0.669 | 1.00x | 1.00x |
| | 256 | 671.1M | 49.6 | 1.00x | 1.00x | 2.58 | 1.00x | 1.00x | 0.0107 | 1.00x | 1.00x | 1.214 | 1.00x | 1.00x |
| BD-LoRA | 32 | 36.2M | 121.5 | 1.29x | N/A | 1.05 | 1.29x | N/A | 0.0069 | 1.29x | N/A | 0.174 | **1.25x** | N/A |
| | 64 | 72.4M | 113.8 | 1.28x | **1.20x** | 1.13 | 1.28x | **1.20x** | 0.0070 | 1.28x | 1.28x | 0.233 | **1.25x** | **0.93x** |
| | 128 | 144.7M | 106.2 | **1.31x** | 1.19x | 1.21 | **1.31x** | 1.19x | 0.0068 | 1.33x | **1.31x** | 0.331 | **1.25x** | 0.88x |
| | 256 | 289.4M | 88.4 | **1.30x** | 1.09x | 1.45 | **1.30x** | 1.09x | 0.0070 | 1.36x | 1.29x | 0.549 | 1.22x | 0.75x |
| | 512 | 578.8M | 63.5 | 1.28x | 0.94x | 2.02 | 1.28x | 0.94x | 0.0076 | **1.41x** | 1.26x | 1.045 | 1.16x | 0.64x |

Table 25: Speedup of throughput, end-to-end (E2E) latency, decoding latency, and prefill latency of **Llama-3.1-70B** using **multi-adapters** with an input Token (**IT**) length of **1024**, an output Token (**OT**) length of **128**, and batch sizes (**BS**) of **1**. **S.-0.87x** denotes the speedup when BD-LoRA has 0.87x the number of trainable parameters compared to S-LoRA or NFS-LoRA. **S.-1.74x** denotes the speedup when BD-LoRA has 1.74x the number of trainable parameters compared to S-LoRA or NFS-LoRA.

| Method | Rank | # Trainable Parameters | Throughput ↑ | | | E2E Latency ↓ | | | Decoding Latency ↓ | | | Prefill Latency ↓ | | |
|---|---|---|---|---|---|---|---|---|---|---|---|---|---|---|
| | | | Token/s | S.-0.87x | S.-1.74x | Time | S.-0.87x | S.-1.74x | Time | S.-0.87x | S.-1.74x | Time | S.-0.87x | S.-1.74x |
| NFS-LoRA | 16 | 207.1M | 34.6 | 1.00x | 1.00x | 3.70 | 1.00x | 1.00x | 0.0243 | 1.00x | 1.00x | 0.593 | 1.00x | 1.00x |
| | 32 | 414.2M | 30.8 | 1.00x | 1.00x | 4.16 | 1.00x | 1.00x | 0.0248 | 1.00x | 1.00x | 0.975 | 1.00x | 1.00x |
| | 64 | 828.4M | 26.0 | 1.00x | 1.00x | 4.93 | 1.00x | 1.00x | 0.0258 | 1.00x | 1.00x | 1.623 | 1.00x | 1.00x |
| | 128 | 1656.8M | 19.4 | 1.00x | 1.00x | 6.59 | 1.00x | 1.00x | 0.0275 | 1.00x | 1.00x | 3.066 | 1.00x | 1.00x |
| | 256 | 3313.5M | 12.8 | 1.00x | 1.00x | 10.01 | 1.00x | 1.00x | 0.0322 | 1.00x | 1.00x | 5.883 | 1.00x | 1.00x |
| BD-LoRA | 32 | 180.2M | 34.6 | 1.00x | N/A | 3.70 | 1.00x | N/A | 0.0240 | 1.01x | N/A | 0.622 | 0.95x | N/A |
| | 64 | 360.4M | 32.7 | 1.06x | **0.94x** | 3.91 | 1.06x | **0.94x** | 0.0240 | 1.03x | 1.01x | 0.836 | 1.17x | **0.71x** |
| | 128 | 720.9M | 28.8 | 1.11x | 0.93x | 4.45 | 1.11x | 0.93x | 0.0241 | 1.07x | 1.03x | 1.367 | 1.19x | **0.71x** |
| | 256 | 1441.8M | 22.5 | **1.16x** | 0.87x | 5.68 | **1.16x** | 0.87x | 0.0247 | 1.11x | 1.04x | 2.517 | **1.22x** | 0.64x |
| | 512 | 2883.6M | 14.5 | 1.14x | 0.75x | 8.81 | 1.14x | 0.75x | 0.0259 | **1.25x** | **1.06x** | 5.497 | 1.07x | 0.56x |
| S-LoRA | 16 | 207.1M | 29.0 | 1.00x | 1.00x | 4.41 | 1.00x | 1.00x | 0.0295 | 1.00x | 1.00x | 0.627 | 1.00x | 1.00x |
| | 32 | 414.2M | 26.5 | 1.00x | 1.00x | 4.83 | 1.00x | 1.00x | 0.0298 | 1.00x | 1.00x | 1.018 | 1.00x | 1.00x |
| | 64 | 828.4M | 23.3 | 1.00x | 1.00x | 5.48 | 1.00x | 1.00x | 0.0302 | 1.00x | 1.00x | 1.614 | 1.00x | 1.00x |
| | 128 | 1656.8M | 18.4 | 1.00x | 1.00x | 6.97 | 1.00x | 1.00x | 0.0315 | 1.00x | 1.00x | 2.945 | 1.00x | 1.00x |
| | 256 | 3313.5M | 12.6 | 1.00x | 1.00x | 10.18 | 1.00x | 1.00x | 0.0345 | 1.00x | 1.00x | 5.767 | 1.00x | 1.00x |
| BD-LoRA | 32 | 180.2M | 34.6 | 1.19x | N/A | 3.70 | 1.19x | N/A | 0.0240 | 1.23x | N/A | 0.622 | 1.01x | N/A |
| | 64 | 360.4M | 32.7 | **1.23x** | **1.13x** | 3.91 | **1.23x** | **1.13x** | 0.0240 | 1.24x | 1.23x | 0.836 | **1.22x** | **0.75x** |
| | 128 | 720.9M | 28.8 | **1.23x** | 1.09x | 4.45 | **1.23x** | 1.09x | 0.0241 | 1.26x | **1.24x** | 1.367 | 1.18x | 0.74x |
| | 256 | 1441.8M | 22.5 | **1.23x** | 0.96x | 5.68 | **1.23x** | 0.96x | 0.0247 | 1.27x | 1.22x | 2.517 | 1.17x | 0.64x |
| | 512 | 2883.6M | 14.5 | 1.16x | 0.79x | 8.81 | 1.16x | 0.79x | 0.0259 | **1.33x** | 1.22x | 5.497 | 1.05x | 0.54x |

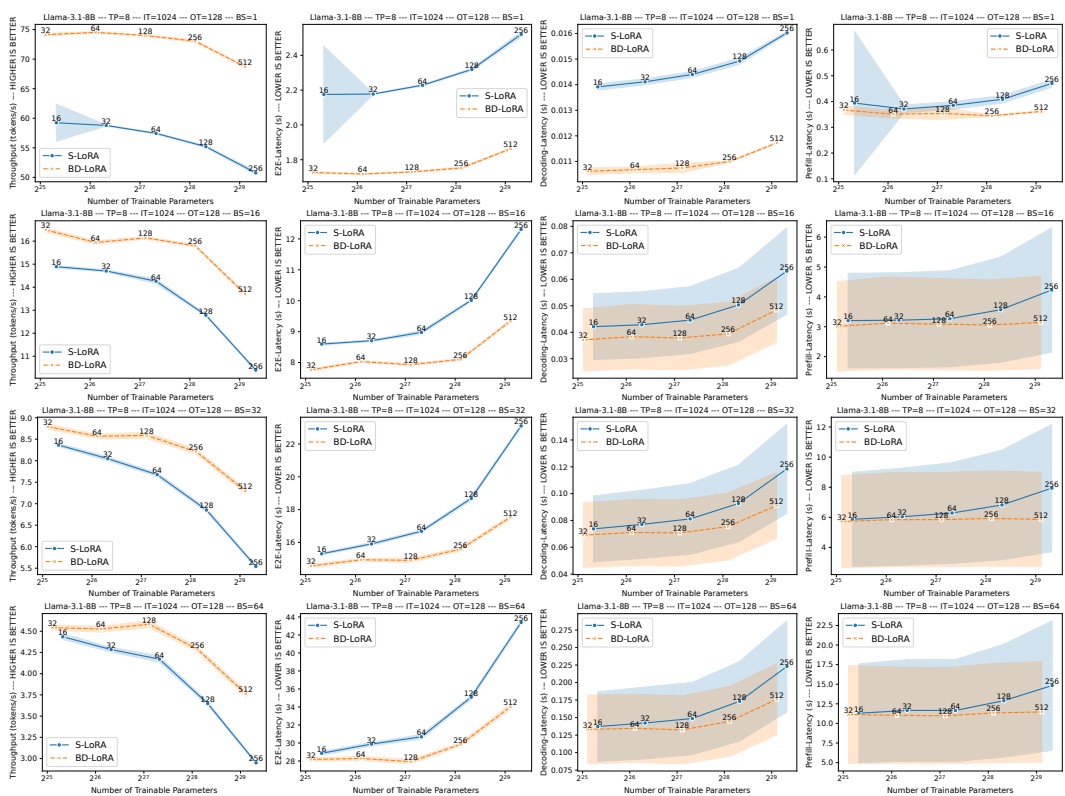

Figure 24: Throughput (1st column—higher is better), end-to-end (E2E) latency (2nd column—lower is better), decoding latency (3rd column—lower is better), and prefill latency (4th column—lower is better) of **Llama-3.1-8B** on **NVIDIA A10G** with an input token (**IT**) length of **1024**, an output token (**OT**) length of **128**, and batch sizes (**BS**) of **1, 16, 32 and 64** (1st to 4th row).

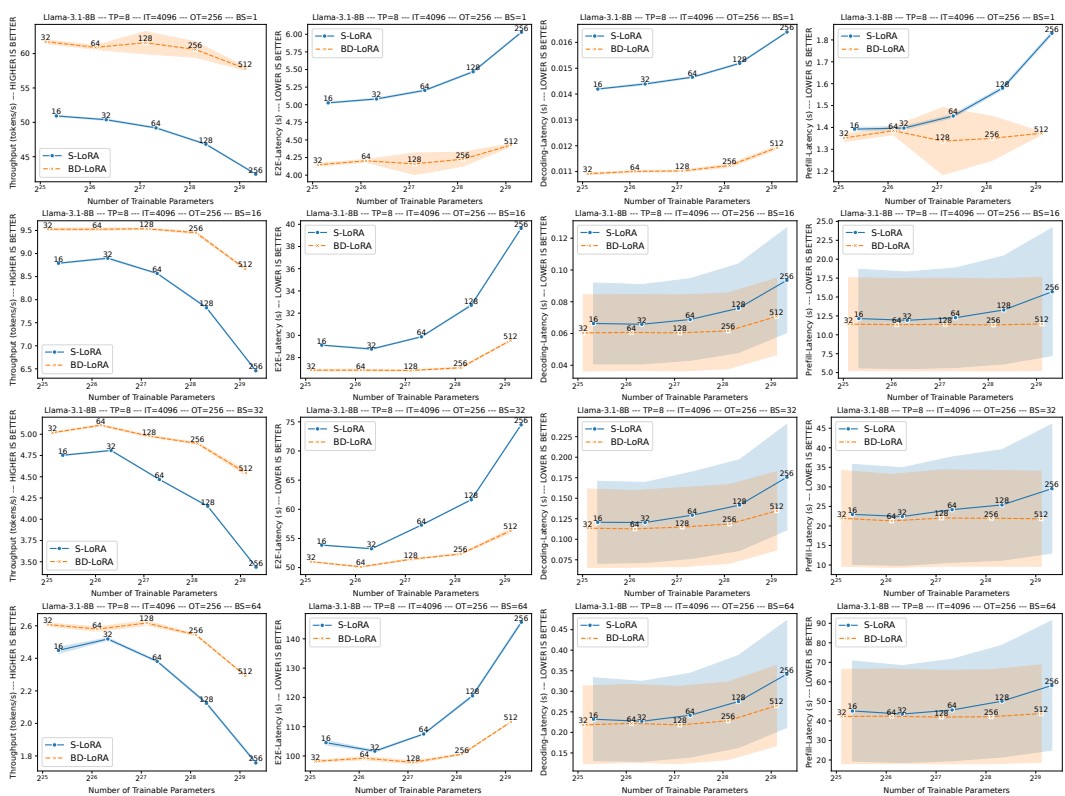

Figure 25: Throughput (1st column—higher is better), end-to-end (E2E) latency (2nd column—lower is better), decoding latency (3rd column—lower is better), and prefill latency (4th column—lower is better) of **Llama-3.1-8B** on **NVIDIA A10G** with an input token (**IT**) length of **4096**, an output token (**OT**) length of **256**, and batch sizes (**BS**) of **1, 16, 32 and 64** (1st to 4th row).

Table 26: Speedup of throughput, end-to-end (E2E) latency, decoding latency, and prefill latency of **Llama-3.1-8B** on **NVIDIA A10G** GPUs with an input token (**IT**) length of **1024**, an output token (**OT**) length of **128**, and batch sizes (**BS**) of **1, 16, 32, and 64**. **S.-0.86x** denotes the speedup when BD-LoRA has 0.86x the number of trainable parameters compared to S-LoRA or NFS-LoRA. **S.-1.73x** denotes the speedup when BD-LoRA has 1.73x the number of trainable parameters compared to S-LoRA or NFS-LoRA.

| Method | Rank | # Trainable Parameters | Throughput ↑ | | | E2E Latency ↓ | | | Decoding Latency ↓ | | | Prefill Latency ↓ | | |
|---|---|---|---|---|---|---|---|---|---|---|---|---|---|---|
| | | | Token/s | S.-0.86x | S.-1.73x | Time | S.-0.86x | S.-1.73x | Time | S.-0.86x | S.-1.73x | Time | S.-0.86x | S.-1.73x |
| **Llama-3.1-8B BS=1** | | | | | | | | | | | | | | |
| S-LoRA | 16 | 41.9M | 59.2 | 1.00x | 1.00x | 2.18 | 1.00x | 1.00x | 0.0139 | 1.00x | 1.00x | 0.394 | 1.00x | 1.00x |
| | 32 | 83.9M | 58.8 | 1.00x | 1.00x | 2.18 | 1.00x | 1.00x | 0.0141 | 1.00x | 1.00x | 0.371 | 1.00x | 1.00x |
| | 64 | 167.8M | 57.4 | 1.00x | 1.00x | 2.23 | 1.00x | 1.00x | 0.0144 | 1.00x | 1.00x | 0.385 | 1.00x | 1.00x |
| | 128 | 335.5M | 55.2 | 1.00x | 1.00x | 2.32 | 1.00x | 1.00x | 0.0149 | 1.00x | 1.00x | 0.409 | 1.00x | 1.00x |
| | 256 | 671.1M | 50.8 | 1.00x | 1.00x | 2.52 | 1.00x | 1.00x | 0.0160 | 1.00x | 1.00x | 0.470 | 1.00x | 1.00x |
| BD-LoRA | 32 | 36.2M | 74.1 | 1.25x | N/A | 1.73 | 1.26x | N/A | 0.0106 | 1.31x | N/A | 0.367 | 1.07x | N/A |
| | 64 | 72.4M | 74.5 | 1.27x | 1.26x | 1.72 | 1.27x | **1.27x** | 0.0107 | 1.32x | 1.30x | 0.351 | 1.06x | 1.12x |
| | 128 | 144.7M | 74.0 | 1.29x | 1.26x | 1.73 | 1.29x | 1.26x | 0.0107 | 1.34x | **1.31x** | 0.354 | 1.09x | 1.05x |
| | 256 | 289.4M | 73.0 | 1.32x | **1.27x** | 1.75 | 1.32x | 1.27x | 0.0110 | 1.36x | **1.31x** | 0.345 | 1.19x | 1.12x |
| | 512 | 578.8M | 68.7 | **1.35x** | 1.25x | 1.86 | **1.35x** | 1.25x | 0.0117 | **1.37x** | 1.27x | 0.362 | 1.30x | 1.13x |
| **Llama-3.1-8B BS=16** | | | | | | | | | | | | | | |
| S-LoRA | 16 | 41.9M | 14.9 | 1.00x | 1.00x | 8.60 | 1.00x | 1.00x | 0.0421 | 1.00x | 1.00x | 3.205 | 1.00x | 1.00x |
| | 32 | 83.9M | 14.7 | 1.00x | 1.00x | 8.70 | 1.00x | 1.00x | 0.0428 | 1.00x | 1.00x | 3.220 | 1.00x | 1.00x |
| | 64 | 167.8M | 14.3 | 1.00x | 1.00x | 8.98 | 1.00x | 1.00x | 0.0446 | 1.00x | 1.00x | 3.267 | 1.00x | 1.00x |
| | 128 | 335.5M | 12.8 | 1.00x | 1.00x | 10.01 | 1.00x | 1.00x | 0.0503 | 1.00x | 1.00x | 3.571 | 1.00x | 1.00x |
| | 256 | 671.1M | 10.4 | 1.00x | 1.00x | 12.31 | 1.00x | 1.00x | 0.0631 | 1.00x | 1.00x | 4.229 | 1.00x | 1.00x |
| BD-LoRA | 32 | 36.2M | 16.5 | 1.11x | N/A | 7.76 | 1.11x | N/A | 0.0371 | 1.13x | N/A | 3.014 | 1.06x | N/A |
| | 64 | 72.4M | 15.9 | 1.08x | 1.07x | 8.03 | 1.08x | 1.07x | 0.0383 | 1.12x | 1.10x | 3.120 | 1.03x | 1.03x |
| | 128 | 144.7M | 16.1 | 1.13x | 1.10x | 7.93 | 1.13x | 1.10x | 0.0378 | 1.18x | 1.13x | 3.087 | 1.06x | 1.04x |
| | 256 | 289.4M | 15.8 | 1.24x | 1.11x | 8.10 | 1.24x | 1.11x | 0.0394 | 1.28x | 1.13x | 3.057 | 1.17x | 1.07x |
| | 512 | 578.8M | 13.7 | 1.32x | 1.07x | 9.34 | 1.32x | 1.07x | 0.0484 | 1.30x | 1.04x | 3.146 | 1.34x | 1.14x |
| **Llama-3.1-8B BS=32** | | | | | | | | | | | | | | |
| S-LoRA | 16 | 41.9M | 8.4 | 1.00x | 1.00x | 15.30 | 1.00x | 1.00x | 0.0737 | 1.00x | 1.00x | 5.869 | 1.00x | 1.00x |
| | 32 | 83.9M | 8.0 | 1.00x | 1.00x | 15.91 | 1.00x | 1.00x | 0.0771 | 1.00x | 1.00x | 6.039 | 1.00x | 1.00x |
| | 64 | 167.8M | 7.7 | 1.00x | 1.00x | 16.68 | 1.00x | 1.00x | 0.0811 | 1.00x | 1.00x | 6.288 | 1.00x | 1.00x |
| | 128 | 335.5M | 6.9 | 1.00x | 1.00x | 18.68 | 1.00x | 1.00x | 0.0926 | 1.00x | 1.00x | 6.821 | 1.00x | 1.00x |
| | 256 | 671.1M | 5.5 | 1.00x | 1.00x | 23.10 | 1.00x | 1.00x | 0.1184 | 1.00x | 1.00x | 7.942 | 1.00x | 1.00x |
| BD-LoRA | 32 | 36.2M | 8.8 | 1.05x | N/A | 14.56 | 1.05x | N/A | 0.0690 | 1.07x | N/A | 5.724 | 1.03x | N/A |
| | 64 | 72.4M | 8.6 | 1.06x | 1.02x | 14.94 | 1.06x | 1.02x | 0.0710 | 1.08x | 1.04x | 5.843 | 1.03x | 1.00x |
| | 128 | 144.7M | 8.6 | 1.12x | 1.07x | 14.90 | 1.12x | 1.07x | 0.0707 | 1.15x | 1.09x | 5.850 | 1.07x | 1.03x |
| | 256 | 289.4M | 8.2 | 1.20x | 1.07x | 15.59 | 1.20x | 1.07x | 0.0754 | 1.23x | 1.08x | 5.925 | 1.15x | 1.06x |
| | 512 | 578.8M | 7.3 | 1.32x | 1.07x | 17.54 | 1.32x | 1.07x | 0.0912 | 1.30x | 1.02x | 5.866 | **1.35x** | **1.16x** |
| **Llama-3.1-8B BS=64** | | | | | | | | | | | | | | |
| S-LoRA | 16 | 41.9M | 4.4 | 1.00x | 1.00x | 28.86 | 1.00x | 1.00x | 0.1371 | 1.00x | 1.00x | 11.302 | 1.00x | 1.00x |
| | 32 | 83.9M | 4.3 | 1.00x | 1.00x | 29.88 | 1.00x | 1.00x | 0.1423 | 1.00x | 1.00x | 11.660 | 1.00x | 1.00x |
| | 64 | 167.8M | 4.2 | 1.00x | 1.00x | 30.69 | 1.00x | 1.00x | 0.1487 | 1.00x | 1.00x | 11.652 | 1.00x | 1.00x |
| | 128 | 335.5M | 3.6 | 1.00x | 1.00x | 35.09 | 1.00x | 1.00x | 0.1733 | 1.00x | 1.00x | 12.905 | 1.00x | 1.00x |
| | 256 | 671.1M | 2.9 | 1.00x | 1.00x | 43.39 | 1.00x | 1.00x | 0.2231 | 1.00x | 1.00x | 14.833 | 1.00x | 1.00x |
| BD-LoRA | 32 | 36.2M | 4.5 | 1.02x | N/A | 28.18 | 1.02x | N/A | 0.1330 | 1.03x | N/A | 11.142 | 1.01x | N/A |
| | 64 | 72.4M | 4.5 | 1.06x | 1.02x | 28.28 | 1.06x | 1.02x | 0.1347 | 1.06x | 1.02x | 11.033 | 1.06x | 1.02x |
| | 128 | 144.7M | 4.6 | 1.10x | 1.07x | 27.93 | 1.10x | 1.07x | 0.1324 | 1.12x | 1.07x | 10.979 | 1.06x | 1.06x |
| | 256 | 289.4M | 4.3 | 1.18x | 1.03x | 29.85 | 1.18x | 1.03x | 0.1446 | 1.20x | 1.03x | 11.341 | 1.14x | 1.03x |
| | 512 | 578.8M | 3.8 | 1.28x | 1.03x | 34.01 | 1.28x | 1.03x | 0.1762 | 1.27x | 0.98x | 11.460 | 1.29x | 1.13x |

Table 27: Speedup of throughput, end-to-end (E2E) latency, decoding latency, and prefill latency of **Llama-3.1-8B** on **NVIDIA A10G** GPUs with an input token (**IT**) length of **4096**, an output token (**OT**) length of **256**, and batch sizes (**BS**) of **1, 16, 32, and 64**. **S.-0.86x** denotes the speedup when BD-LoRA has 0.86x the number of trainable parameters compared to S-LoRA or NFS-LoRA. **S.-1.73x** denotes the speedup when BD-LoRA has 1.73x the number of trainable parameters compared to S-LoRA or NFS-LoRA.

| Method | Rank | # Trainable Parameters | Throughput ↑ | | | E2E Latency ↓ | | | Decoding Latency ↓ | | | Prefill Latency ↓ | | |
|---|---|---|---|---|---|---|---|---|---|---|---|---|---|---|
| | | | Token/s | S.-0.86x | S.-1.73x | Time | S.-0.86x | S.-1.73x | Time | S.-0.86x | S.-1.73x | Time | S.-0.86x | S.-1.73x |
| | | | | | | **Llama-3.1-8B BS=1** | | | | | | | | |
| S-LoRA | 16 | 41.9M | 50.9 | 1.00x | 1.00x | 5.03 | 1.00x | 1.00x | 0.0142 | 1.00x | 1.00x | 1.392 | 1.00x | 1.00x |
| | 32 | 83.9M | 50.4 | 1.00x | 1.00x | 5.08 | 1.00x | 1.00x | 0.0144 | 1.00x | 1.00x | 1.397 | 1.00x | 1.00x |
| | 64 | 167.8M | 49.2 | 1.00x | 1.00x | 5.20 | 1.00x | 1.00x | 0.0147 | 1.00x | 1.00x | 1.452 | 1.00x | 1.00x |
| | 128 | 335.5M | 46.8 | 1.00x | 1.00x | 5.47 | 1.00x | 1.00x | 0.0152 | 1.00x | 1.00x | 1.580 | 1.00x | 1.00x |
| | 256 | 671.1M | 42.5 | 1.00x | 1.00x | 6.03 | 1.00x | 1.00x | 0.0164 | 1.00x | 1.00x | 1.830 | 1.00x | 1.00x |
| BD-LoRA | 32 | 36.2M | 61.7 | 1.21x | N/A | 4.15 | 1.21x | N/A | 0.0109 | 1.30x | N/A | 1.352 | 1.03x | N/A |
| | 64 | 72.4M | 60.9 | 1.21x | 1.20x | 4.20 | 1.21x | 1.20x | 0.0110 | 1.31x | 1.29x | 1.385 | 1.01x | 1.01x |
| | 128 | 144.7M | 61.6 | 1.25x | 1.22x | 4.16 | 1.25x | 1.22x | 0.0110 | 1.33x | **1.31x** | 1.338 | 1.09x | 1.04x |
| | 256 | 289.4M | 60.6 | 1.29x | 1.23x | 4.23 | 1.29x | 1.23x | 0.0112 | 1.35x | 1.30x | 1.350 | 1.17x | 1.08x |
| | 512 | 578.8M | 57.8 | **1.36x** | **1.24x** | 4.43 | **1.36x** | **1.24x** | 0.0119 | **1.38x** | 1.27x | 1.373 | 1.33x | 1.15x |
| | | | | | | **Llama-3.1-8B BS=16** | | | | | | | | |
| S-LoRA | 16 | 41.9M | 8.8 | 1.00x | 1.00x | 29.12 | 1.00x | 1.00x | 0.0663 | 1.00x | 1.00x | 12.146 | 1.00x | 1.00x |
| | 32 | 83.9M | 8.9 | 1.00x | 1.00x | 28.77 | 1.00x | 1.00x | 0.0658 | 1.00x | 1.00x | 11.918 | 1.00x | 1.00x |
| | 64 | 167.8M | 8.6 | 1.00x | 1.00x | 29.88 | 1.00x | 1.00x | 0.0688 | 1.00x | 1.00x | 12.260 | 1.00x | 1.00x |
| | 128 | 335.5M | 7.8 | 1.00x | 1.00x | 32.72 | 1.00x | 1.00x | 0.0759 | 1.00x | 1.00x | 13.290 | 1.00x | 1.00x |
| | 256 | 671.1M | 6.5 | 1.00x | 1.00x | 39.64 | 1.00x | 1.00x | 0.0935 | 1.00x | 1.00x | 15.698 | 1.00x | 1.00x |
| BD-LoRA | 32 | 36.2M | 9.5 | 1.08x | N/A | 26.87 | 1.08x | N/A | 0.0604 | 1.10x | N/A | 11.410 | 1.06x | N/A |
| | 64 | 72.4M | 9.5 | 1.07x | 1.08x | 26.87 | 1.07x | 1.08x | 0.0607 | 1.08x | 1.09x | 11.337 | 1.05x | 1.07x |
| | 128 | 144.7M | 9.5 | 1.11x | 1.07x | 26.84 | 1.11x | 1.07x | 0.0604 | 1.14x | 1.09x | 11.376 | 1.08x | 1.05x |
| | 256 | 289.4M | 9.5 | 1.21x | 1.10x | 27.08 | 1.21x | 1.10x | 0.0616 | 1.23x | 1.12x | 11.301 | 1.18x | 1.08x |
| | 512 | 578.8M | 8.7 | 1.34x | 1.11x | 29.54 | 1.34x | 1.11x | 0.0707 | 1.32x | 1.07x | 11.438 | **1.37x** | **1.16x** |
| | | | | | | **Llama-3.1-8B BS=32** | | | | | | | | |
| S-LoRA | 16 | 41.9M | 4.8 | 1.00x | 1.00x | 53.86 | 1.00x | 1.00x | 0.1207 | 1.00x | 1.00x | 22.958 | 1.00x | 1.00x |
| | 32 | 83.9M | 4.8 | 1.00x | 1.00x | 53.24 | 1.00x | 1.00x | 0.1204 | 1.00x | 1.00x | 22.402 | 1.00x | 1.00x |
| | 64 | 167.8M | 4.5 | 1.00x | 1.00x | 57.29 | 1.00x | 1.00x | 0.1294 | 1.00x | 1.00x | 24.169 | 1.00x | 1.00x |
| | 128 | 335.5M | 4.2 | 1.00x | 1.00x | 61.64 | 1.00x | 1.00x | 0.1417 | 1.00x | 1.00x | 25.370 | 1.00x | 1.00x |
| | 256 | 671.1M | 3.4 | 1.00x | 1.00x | 74.49 | 1.00x | 1.00x | 0.1756 | 1.00x | 1.00x | 29.522 | 1.00x | 1.00x |
| BD-LoRA | 32 | 36.2M | 5.0 | 1.06x | N/A | 51.02 | 1.06x | N/A | 0.1135 | 1.06x | N/A | 21.946 | 1.05x | N/A |
| | 64 | 72.4M | 5.1 | 1.06x | 1.07x | 50.14 | 1.06x | 1.07x | 0.1127 | 1.07x | 1.07x | 21.285 | 1.05x | 1.08x |
| | 128 | 144.7M | 5.0 | 1.11x | 1.04x | 51.43 | 1.11x | 1.04x | 0.1149 | 1.13x | 1.05x | 22.019 | 1.10x | 1.02x |
| | 256 | 289.4M | 4.9 | 1.18x | 1.09x | 52.33 | 1.18x | 1.09x | 0.1186 | 1.19x | 1.09x | 21.961 | 1.16x | 1.10x |
| | 512 | 578.8M | 4.5 | 1.32x | 1.09x | 56.32 | 1.32x | 1.09x | 0.1347 | 1.30x | 1.05x | 21.841 | 1.35x | **1.16x** |
| | | | | | | **Llama-3.1-8B BS=64** | | | | | | | | |
| S-LoRA | 16 | 41.9M | 2.4 | 1.00x | 1.00x | 104.55 | 1.00x | 1.00x | 0.2321 | 1.00x | 1.00x | 45.112 | 1.00x | 1.00x |
| | 32 | 83.9M | 2.5 | 1.00x | 1.00x | 101.64 | 1.00x | 1.00x | 0.2270 | 1.00x | 1.00x | 43.525 | 1.00x | 1.00x |
| | 64 | 167.8M | 2.4 | 1.00x | 1.00x | 107.50 | 1.00x | 1.00x | 0.2418 | 1.00x | 1.00x | 45.603 | 1.00x | 1.00x |
| | 128 | 335.5M | 2.1 | 1.00x | 1.00x | 120.55 | 1.00x | 1.00x | 0.2751 | 1.00x | 1.00x | 50.120 | 1.00x | 1.00x |
| | 256 | 671.1M | 1.8 | 1.00x | 1.00x | 145.68 | 1.00x | 1.00x | 0.3418 | 1.00x | 1.00x | 58.161 | 1.00x | 1.00x |
| BD-LoRA | 32 | 36.2M | 2.6 | 1.06x | N/A | 98.18 | 1.06x | N/A | 0.2182 | 1.06x | N/A | 42.308 | 1.07x | N/A |
| | 64 | 72.4M | 2.6 | 1.02x | 1.05x | 99.24 | 1.02x | 1.05x | 0.2220 | 1.02x | 1.05x | 42.407 | 1.03x | 1.06x |
| | 128 | 144.7M | 2.6 | 1.10x | 1.04x | 97.83 | 1.10x | 1.04x | 0.2182 | 1.11x | 1.04x | 41.972 | 1.09x | 1.04x |
| | 256 | 289.4M | 2.5 | 1.20x | 1.07x | 100.63 | 1.20x | 1.07x | 0.2283 | 1.20x | 1.06x | 42.175 | 1.19x | 1.08x |
| | 512 | 578.8M | 2.3 | 1.30x | 1.08x | 111.63 | 1.30x | 1.08x | 0.2651 | 1.29x | 1.04x | 43.763 | 1.33x | 1.15x |

Table 28: Speedup of throughput, end-to-end (E2E) latency, decoding latency, and prefill latency with a **quantized Llama-3.1-8B** model (`https://huggingface.co/RedHatAI/Meta-Llama-3.1-8B-Instruct-quantized.w4a16`) with an input token (**IT**) length of **1024**, an output token (**OT**) length of **128**, and batch size (**BS**) of **16**. Rank is chosen such that BD-LoRA has better accuracies and thus speedups are meaningful.

| Method | Rank | # Trainable Parameters | OpenOrca ↓ | GLUE ↑ | Throughput [token/s] | E2E Latency [s] | Decoding Latency [s] | Prefill Latency [s] |
|---|---|---|---|---|---|---|---|---|
| **BD-LoRA** | 128 | 144.7M | 2.303 | 76.17 | 56.00 | 2.29 | 0.01 | 0.49 |
| **S-LoRA** | 64 | 167.8M | 2.303 | 76.01 | 44.30 | 2.89 | 0.02 | 0.50 |
| Speedup over S-LoRA | - | - | - | - | 1.26x | 1.26x | 1.26x | 1.03x |
| **NFS-LoRA** | 64 | 167.8M | 2.303 | 76.01 | 47.80 | 2.68 | 0.02 | 0.61 |
| Speedup over NFS-LoRA | - | - | - | - | 1.17x | 1.17x | 1.21x | 1.24x |

