# OpenReview forum: "Block-Diagonal LoRA for Eliminating Communication Overhead in Tensor Parallel LoRA Serving"
_NeurIPS.cc/2025/Conference — NeurIPS 2025 poster_

### Official Review · Reviewer_h1j1 · 2025-06-15

**Clarity:** 2
**Significance:** 2
**Originality:** 3
**Rating:** 4
**Confidence:** 3

**Summary:**

This paper proposed a method named block-diagonal LoRA (BD-LoRA) to solve the communication costs of the previous S-Lora designed for serving thousands of different LoRA with the same base LLM. It first shows that S-Lora suffers from additional all-gather and one all-reduce operations. Motivated by this, BD-Lora adopts a block-wise weight matrix which enables the add operation within the GPU, thereby reducing the all-gather and one all-reduce operations. Lots of experiments show that BD-Lora enjoys lower time costs while having comparable performance.

**Questions:**

BD-LoRA’s core assumption is that weight shards on each GPU can co-exist with all required LoRA shards. For the "hundreds or thousands of LoRA adapters" scenario claimed to be supported by the paper, this assumption may not be met in actual deployments with limited memory.

**Ethical Concerns:**

["NO or VERY MINOR ethics concerns only"]

**Final Justification:**

I consider this paper to be simple but effective. For my current rating, I recommand accept (Borderline accept).

**Limitations:**

Yes.

**Paper Formatting Concerns:**

No concerns.

**Quality:**

3

**Strengths And Weaknesses:**

Strengths:

1. BD-Lora is a simple but effective method. The motivation is straightforward, and the performance is expected as it is.

2. This paper shows comprehensive experiments. These results fully demonstrate that the DB-Lora not only reduces time costs but also has a comparable performance with the normal Lora.

3. Detailed implementation section, which makes the reproduction easier.

Weaknesses:

1. The writing is not good. I highly suggest the author rewrite this paper to make it more concise and simple. The current version somehow makes the reader lose track since many sentences are redundant. A simple but concise description is the basic requirement for a top-tier conference.

2. Figures are too small. I highly suggest the author reorganize their figures to make these clearer.

3. Although multiple LoRA services were mentioned, most of the experiments were conducted with a single adapter or with the same adapter within a batch. Is it possible to add a simulation of real scenarios (such as 100 users with different adapters each, mixed across batches)?

---

> ### Author Rebuttal · Authors · 2025-07-30
>
> We thank the reviewer for their positive reception of our work. We will try to further improve the presentation in the final version of the paper and also increase figure sizes (the additionally allowed page in the camera-ready version will help us to do these). As for the single adapter vs multiple adapter comment and question, please note that our experiments mainly focus on the single adapter case because it best demonstrates (i.e., isolates) our innovation over S-LoRA (cf. Lines 270-280). But we
> do provide experiments with different adapters for every request in Appendix D.4 and a summary of our findings in the main part of the paper (Lines 324 - 330). In particular, in the latter scenario adapters are not held in GPU, but loaded from disk, and we still see significant speed-up of BD-LoRA, showing that it is not true that “BD-LoRA’s core assumption is that weight shards on each GPU can co-exist with all required LoRA shards.”

---

### Official Review · Reviewer_QwNc · 2025-07-01

**Clarity:** 3
**Significance:** 2
**Originality:** 2
**Rating:** 4
**Confidence:** 5

**Summary:**

- This work builds upon S-LoRA, targeting the task of serving multiple LoRA adapters simultaneously. The problem of serving multiple adapters is that the adapters cannot be merged to the base model as the adapter swapping would create computation overhead and the use of different adapters could not be batched.
- The author proposes to constrain LoRA factors to be block-diagonal, enabling better sharding of the adapter weights, without the need for any additional communication for the LoRA computations.
- The author justifies their method by demonstrating that on the one-hand their method does not degrade the parameter-efficiency (by increasing the rank), on the other hand has significant end-to-end speed-up over S-LoRA.

**Questions:**

- One thing I am a little bid confused is the block-diagonal structure of LoRA, in section three you say to let A or B to be block-diagonal, but both A and B or not symmetric matrices, how do you enforce the block-diagonal structure?
- I think another big benefit of this line of work (increasing the inference speed without merging the adapters) is to serve QLoRA-finetuned models, since the base model is quantized to lower precision, while the adapter weights are kept in higher precision, merging trained adapter weights to the base model will inevitably worsen the performance. To achieve the same performance, one should keep the adapters separate. But this will lead to slow inference speed. I would be interested to see how your method will benefit the serving of QLoRA-finetuned methods. Can you provide insight or experiments about this?

**Ethical Concerns:**

["NO or VERY MINOR ethics concerns only"]

**Final Justification:**

The author has addressed all my concerns and provided additional experiments for clarification, thus I will increase my rating to borderline accept.

**Limitations:**

In general, I find this work well-written and the idea intuitive, but I feel like there are some concerns open. I am willing to raise my ratings when the mentioned questions and concerns are being addressed

**Paper Formatting Concerns:**

I did not notice any significant formatting concerns.

**Quality:**

3

**Strengths And Weaknesses:**

Strengths
- The method builds upon S-LoRA while being strictly Pareto-dominant over S-LoRA, without degrading the performance
- The speed up is achieved by getting rid of communication, thus by principle faster than S-LoRA
- The author verifies the effectiveness through comprehensive studies

Weakness:
- Missing related work/baseline: using block-diagonal structure in parameter-efficient finetuning has been studied extensively within the orthogonal finetuning line of work, the block-diagonality structure has also been exploited for the task of serving multiple orthogonal adapters ("In Defense of Structural Sparse Adapters for Concurrent LLM Serving.") I would like to see the authors to compare against these previous works, a small-scale ablation experiment would be sufficient. Also, a small discussion of the benefits of BD-LoRA over the previous works should be included in the related work section.

---

> ### Author Rebuttal · Authors · 2025-07-30
>
> We thank the reviewer for their initial assessment and appreciate their view on our work. We hope to address some of the questions and proclaimed weaknesses below.
>
> >One thing I am a little bid confused is the block-diagonal structure of LoRA,
> in section three you say to let A or B to be block-diagonal, but both A and B or not
> symmetric matrices, how do you enforce the block-diagonal structure?
>
> Thanks for raising this question and making us aware that the definition of
> "block-diagonal" is not consistent in the literature, e.g., the standard text book
> "Matrix Computations" by Golub & Van Loan defines "block-diagonal" for general $m\times n$-
> matrices in the same way as we do at the beginning of Section 3 in our paper,
> but Wikipedia defines "block-diagonal" only for square matrices (we assume that by
> symmetric you mean square?). We will be more explicit in the final version of the paper.
>
>
> >Missing related work/baseline: using block-diagonal structure in parameter-efficient
>  finetuning has been studied extensively within the orthogonal finetuning
>  line of work, the block-diagonality structure has also been exploited for
>  the task of serving multiple orthogonal adapters
>  ("In Defense of Structural Sparse Adapters for Concurrent LLM Serving.")
>  I would like to see the authors to compare against these previous works,
>  a small-scale ablation experiment would be sufficient. Also, a small discussion of
>  the benefits of BD-LoRA over the previous works should be included in the related work section.
>
>
> Thanks for bringing this work to our attention--we will
> certainly include a discussion of it in the final version of our paper. However,
> that line of work is very different from our proposed approach. Concretely, the paper
> you mentioned (SpartanServe) (i) buils on orthogonal fine-tuning rather than LoRA as we do,
> (ii) is not concerned with the communication overhead of S-LoRA at all and all
> evaluations are performed on a single GPU whereas our focus is entirely on multi-GPU
> serving and communication reduction, and (iii) our proposed block-structure has
> nothing to do with the block-diagonality structure considered there (sorry again
> if there was confusion about the meaning of block-diagonal, see above). Note that
> SpartanServe does not discuss multi-GPU serving at all and it is unclear how one would
> do it, and that a comparison of our approach to SpartanServe on a single GPU would completely
> miss our point and would be the same as comparing S-LoRA with SpartanServe, which has
> already been done in the SpartanServe paper. As such, we kindly ask you to reconsider
> your request to run experiments with SpartanServe as a baseline (besides, we could not
> find any open Source implementation of SpartanServe).
>
>
> >I would be interested to see how your method will benefit the serving of
> QLoRA-finetuned methods. Can you provide insight or experiments about this?
>
> Thank you for this suggestion, this is indeed a nice and straightforward generalization of our work. In fact our vLLM implementation of BD-LoRA works out of the box also with quantized models. To illustrate this we ran some new benchmark where we are using a quantized variant of Llama-3.1 8B Instruct (https://huggingface.co/RedHatAI/Meta-Llama-3.1-8B-Instruct-quantized.w4a16) as the backbone model.
> For BD-LoRA we use TP=8 and rank 128. For vanilla LoRA we use rank 64. In this setting BD-LoRA achieves same or better perplexity/accuracy than LoRA on OpenOrca and Glue. We benchmark with batch size 16, 1024 input tokens and 128 output tokens. We find  that serving BD-LoRA on top of the quantized model gives E2E speedups of 1.26x over S-LoRA and 1.17x over NFS-LoRA. We will include those results in the paper and discuss the applicability to finetuning and serving with quantized models.
>
> |                       | Rank | # Trainable Parameters | OpenOrca Perplexity (lower better) | Glue avg. Score (higher better) | Throughput [token/s] | E2E latency [s] | Decoding Latency [s] | Prefill Latency [s] |
> |-----------------------|------|------------------------|-------------------------|----------------------|----------------------|-----------------|----------------------|---------------------|
> | BD-LoRA               |  128 | 144.7M                 |                   2.303 |                76.17 |                56.00 |            2.29 |                 0.01 |                0.49 |
> | S-Lora                |   64 | 167.8M                 |                   2.303 |                76.01 |                44.30 |            2.89 |                 0.02 |                0.50 |
> | Speedup over S-LoRA   |      |                        |                         |                      |                 1.26x |            1.26x |                 1.26x |                1.03x |
> | NFS-LoRA              |   64 | 167.8M                 |                   2.303 |                76.01 |                47.80 |            2.68 |                 0.02 |                0.61 |
> | Speedup over NFS-LoRA |      |                        |                         |                      |                 1.17x |            1.17x |                 1.21x |                1.24x |

---

> > ### Comment · Reviewer_QwNc · 2025-08-05
> >
> > Thanks for the response. I have no further concerns. Best,

---

### Official Review · Reviewer_PcKT · 2025-07-02

**Clarity:** 2
**Significance:** 3
**Originality:** 3
**Rating:** 5
**Confidence:** 2

**Summary:**

The paper presents a technique called Block-diagonal LoRA for the efficient serving of LLMs with many LoRA adapters. The paper proposes constraining part of LoRA factors to be block-diagonal, reducing the communication costs.

The first part of the evaluation shows that standard LoRA and BD-LoRA achieve similar performance on the downstream tasks. The rest of the evaluation shows that BD-LoRA provides significant speedups compared to baseline S-LoRA.

**Questions:**

How does BD-LoRA affect the prefill stage vs the decode stage of the inference?

**Ethical Concerns:**

["NO or VERY MINOR ethics concerns only"]

**Final Justification:**

Authors have addressed my remaining concerns. I keep my positive score.

**Limitations:**

Yes

**Quality:**

3

**Strengths And Weaknesses:**

**Strengths:**

1. The problem of serving LLMs with multiple adapters is crucial, and the paper makes a strong contribution in this direction.
2. The evaluation is comprehensive and shows the effectiveness of BD-LoRA compared to state-of-the-art baseline S-LoRA.

**Weaknesses:**

1. BD-LoRA requires having information on the TP degree N before the training of the LoRA adapter. This limits the transferability of these adapters to a different setup.

2. The description in the paper uses examples from Figures 2 and 3 nicely to explain the idea. However, the writing does not summarize how this idea is used in the general case and mainly lacks a high-level pseudocode/algorithm on applying the technique given any DNN. The algorithm should clearly explain how, given any arbitrary model architecture for the purpose of fine-tuning, the approach identifies all adapter components that need to be block-diagonal.

**Minor:**

- Line 7: Cite S-LoRA in abstract

- Line 82: define rank r here again - Line 167:  The alternate interpretation of simply training a separate adapter of rank r/N on N devices sounds more intuitive to understand. This may come earlier in the paper for better readability.

- Line 186: I would recommend moving Section 4 with the additional related work to the end of the paper.

- I would also recommend that authors include how BD-LoRA affects the prefill stage vs decode stage in Section 3 of the paper.

- Figure 4: The font size for labels in the plots is too small.

---

> ### Author Rebuttal · Authors · 2025-07-30
>
> We thank the reviewer for evaluating that we are making a strong contribution to a crucial problem. Below we address two questions and kindly invite the reviewer to read our rebuttal to reviewer LsQR as well, where we provide additional discussion on the limitation of knowing the TP degree upfront.
>
> >The description in the paper uses examples from Figures 2 and 3 nicely to explain the idea. However, the writing does not summarize how this idea is used in the general case and mainly lacks a high-level pseudocode/algorithm on applying the technique given any DNN. The algorithm should clearly explain how, given any arbitrary model architecture for the purpose of fine-tuning, the approach identifies all adapter components that need to be block-diagonal.
>
> This is a great suggestion and we will enrich the final paper with a discussion, illustration and pseudocode for the general Matrix Multiplication in an arbitrary DNN. We emphasize that the structure of the adapters should always be based on the parallelism design of the serving configuration. For a general matrix multiplication, let us assume the setting that all inputs and outputs should be replicated across devices and that the weights are column sharded (sharded along the output dimension). Then the backbone distributed matrix multiplication would require an all-gather after the sharded matrix multiplication. In this case BD-LoRA would have the `B` matrix block diagonal, i.e., the constraint is the same as for the first matrix multiplication in the Megatron style approach. The `A` Matrix is dense regularly column-sharded. This approach would hence not introduce any additional communication beyond the all-gather required by the backbone. We illustrate this for N shards in below pseudocode:
>
> ```
> Input: X ∈ ℝ^(B×d_in) (replicated), W^(i) ∈ ℝ^(d_in×d_out/N) (base weights, shard i)
>       A^(i) ∈ ℝ^(d_in×r/N) (LoRA A matrix, shard i), B^(i) ∈ ℝ^(r/N×d_out/N) (LoRA B block i)
> Output: Y ∈ ℝ^(B×d_out) (complete output)
>
> 1: for each device i ∈ {1, 2, ..., N} do
> 2:    // Compute base model contribution
> 3:    Y_base^(i) ← X @ W^(i)
> 4:
> 5:    // Compute adapter contribution (no communication)
> 6:    Z^(i) ← X @ A^(i)
> 7:    Y_adapter^(i) ← Z^(i) @ B^(i)
> 8:
> 9:    // Combine local contributions
> 10:   Y_local^(i) ← Y_base^(i) + Y_adapter^(i)
> 11: end for
> 12:
> 13: // Aggregate results across devices
> 14: Y ← AllGather({Y_local^(1), Y_local^(2), ..., Y_local^(N)})
> 15: return Y
> ```
>
> > How does BD-LoRA affect the prefill stage vs the decode stage of the inference?
>
> In Figure 5 and Section 5.2 we illustrate and discuss how BD-LoRA affects prefill and decoding stage. `The speed-up of BD-LoRA mainly comes from a speed-up in decoding (and hence is largest in generation-heavy tasks with small IT and large OT; cf. Appendix D). BD-LoRA usually also has the lowest prefill latency, but for the prefill latency the confidence intervals are typically very large and heavily overlap so that we cannot consider any method to be superior.`
> We will expand this paragraph with some additional explanation. Namely, during decoding the data size that needs to be communicated is very small, hence the communication kernels' latency is basically a fixed overhead. This overhead plays a large role during decoding. During prefill the communication data sizes are larger, but so are the compute latencies of the matrix multiplications. Overall, during prefill the communication overhead is not as significant as during decoding.
>
> ---
> Thank you for the additional suggestions on how to improve the presentation of our work. We will incorporate those for the final version. We hope that our rebuttal helps to increase your confidence in our paper.

---

### Official Review · Reviewer_LsQR · 2025-07-02

**Clarity:** 3
**Significance:** 4
**Originality:** 3
**Rating:** 4
**Confidence:** 2

**Summary:**

This paper presents Block-Diagonal LoRA (BD-LoRA) which is a variant of LoRA (Low-Rank Adaptation) targeted towards eliminating communication bottlenecks in LLM fine tuning. Existing methods shard the low rank factors in a similar way as the parameter matrices, but this incurs communications overhead in distributed training on multiple GPUs. The authors propose using a block-diagonal approximation which avoids the entire communication across shards. Experimental evaluations show that BD-LoRA can achieve similar performance as standard LoRA for a similar number of parameters. Llama models on vLLM show significant latency and throughput gains (up to 1.79x speedup) over S-LoRA, by removing communication costs.

**Questions:**

1. As mentioned in Section 3, increasing rank r can increase expressivity in low rank factors. However, wouldn't increasing the rank in S-LoRA increase the computation and thereby allow better hiding of the communication overhead (e.g. compute each rank in a pipelined fashion.). If so, then is there any additional benefit of BD-LoRA if the comms overhead can be almost completely hidden?

2. Can the N-dependence be alleviated by using smaller blocks than dictated by the sharding? How small can the blocks be while still maintaining parameter-efficiency?

**Ethical Concerns:**

["NO or VERY MINOR ethics concerns only"]

**Limitations:**

yes

**Quality:**

3

**Strengths And Weaknesses:**

*Strengths*
- Core idea is simple and elegant, which combines system optimization with insight about the mathematical structure of the approximation. The method avoids communication entirely.
- The authors show that the rank can be increased in the block diagonal approximation, matching the number of parameters as the standard LoRA setups, while still maintaining parameter-efficient downstream quality. This shows that even in terms of absolute quality, BD-LoRA can match standard LoRA.
- Extensive evaluations show that BD-LoRA is parameter-efficient while achieving significant speedups on practical models and training/serving setups.

*Weaknesses*
- The quality of fine-tuning becomes highly dependent on the sharding strategy used. This may be undesirable and hurt reproducibilty and flexibility.
- The number of devices (N or TP degree) must be known at training time. The authors already state this limitation. To add another point to this limitation, this significantly hinders training on a different number of devices (e.g. to speed up training) than the serving number of devices.

---

> ### Author Rebuttal · Authors · 2025-07-30
>
> We thank the reviewer for their positive assessment of our work and their
> additional comments and questions, which we hope to address below:
>
> > The number of devices (N or TP degree) must be known at training time.
> The authors already state this limitation. To add another point to this limitation,
> this significantly hinders training on a different number of devices
> (e.g. to speed up training) than the serving number of devices.
>
> The motivation for our work comes purely from *serving* the models more efficiently
> rather than *training* more efficiently. The additional communication introduced by
> LoRA adapters only occurs when using Tensor Parallelism.
> Tensor parallelism is prevalent in serving, because usually there is no pipelining
> across layers involved, and hence TP is the simplest way to utilize all the
> devices for every layer. In fact, the training with the PEFT library is based on FSDP,
> and we do not employ tensor parallelism in training. This means that BD-LoRA does
> not reduce the training cost, but the training cost remains the same. But it also
> means that the number of devices to train can be freely chosen,
> irrespectively of the target serving TP degree (same as for vanilla LoRA).
> However, as we discuss at the end of Section 3 (as acknowledged by the reviewer),
> the serving-TP degree has to be defined upfront as it defines the
> architecture and parameter counts of the BD-LoRA adapters---but note that for each
> base LLM and serving hardware configuration there is usually a typical serving-TP degree
> (e.g., TP = 8 when serving Llama-X-70B on NVIDIA DGX A100 servers).
>
> > As mentioned in Section 3, increasing rank r can increase expressivity in low rank
> factors. However, wouldn't increasing the rank in S-LoRA increase the computation
> and thereby allow better hiding of the communication overhead (e.g. compute each
> rank in a pipelined fashion.). If so, then is there any additional benefit of BD-LoRA
>  if the comms overhead can be almost completely hidden?
>
> As discussed above we are not targeting communication reduction for training, where
> such overlapping consideration could potentially be relevant if training with tensor parallelism (which is usually not the case for LoRA). During inference, hiding the communication is much harder, and it is not even possible
> to fully hide the communication overhead in the base model (without Lora) within vLLM. The base model (without Lora) already has large shapes, which --in theory-- should allow hiding of
> communication. But even there it proves hard. Furthermore, irrespective of the
> communication overheads, the adapters should still be kept small, otherwise the
> actual matrix multiplications will take too much time. Especially in the decoding
> phase the communication amounts are also very tiny. The communication latency is
> thus basically a startup time constant, and tiling the communication would make that
> even worse. Overall, in practice, we see that eliminating communication calls is the
> more promising approach for serving LLMs whenever it can be done without negative
> accuracy impact.
>
> > Can the N-dependence be alleviated by using smaller blocks than dictated by
> the sharding? How small can the blocks be while still maintaining parameter-efficiency?
>
> Thanks for pushing us on this point, as it can be perceived as a main limitation of BD-LoRA
> --- but please note again that for each base LLM and serving hardware configuration
> there is usually a typical serving-TP degree
> (e.g., TP = 8 when serving Llama-X-70B on NVIDIA DGX A100 servers).
> As we discussed in line 175 to 185, a BD-LoRA adapter trained for TP=8 is also a
> valid BD-LoRA adapter for TP=4 etc. However, implementing such "downward compatible" version efficiently,
> while conceptually being easy, would require quite some additional work.
>
>
> While TP=8 seems the most relevant use case (as most nodes have up to 8 devices), it is an interesting question whether BD-LoRA maintains a favorable accuracy vs. parameter count tradeoff when (hypothetically) increasing the TP degree even further. Since only one of each two adapters is block-diagonal, keep in mind that increasing the TP degree (at a fixed rank) does not imply the parameter count goes to zero. We will run an ablation where we start BD-LoRA with rank 64 and TP=1, which is thus equivalent to vanilla LoRA. For BD-LoRA we will then keep the rank fixed and swipe the TP degree over 1, 2, 4, 8, 16, 32, 64. We will confront it with vanilla LoRA at smaller rank.

---

> > ### Comment · Reviewer_LsQR · 2025-08-08
> >
> > Thank you for answering my questions. I will keep my score at this time.

---

### Decision · Program_Chairs · 2025-09-17

**Decision:**

Accept (poster)

**Comment:**

The authors propose to add an additional block diagonal structure to low rank adaptors (LoRA) in order to reduce communication bottlenecks in LLM fine tuning. All reviewers praise the idea to be simple but effective, as corroborated by comprehensive experiments. On the other hand, it is recommended to improve the writing by removing redundant sentences.